# Can machine learning extract the mechanisms controlling phytoplankton growth from large-scale observations? – A proof of concept study

Christopher Holder[1], Anand Gnanadesikan[1]

[1] Morton K. Blaustein Department of Earth and Planetary Sciences, Johns Hopkins University, Baltimore, MD 21218, United States of America

*Correspondence to:* Christopher Holder (cholder2@jh.edu)

**Abstract**

A key challenge for biological oceanography is relating the physiological mechanisms controlling phytoplankton growth to the spatial distribution of those phytoplankton. Physiological mechanisms are often isolated by varying one driver of growth, such as nutrient or light, in a controlled laboratory setting producing what we call "intrinsic relationships". We contrast these with the "apparent relationships" which emerge in the environment in

climatological data. Although previous studies have found machine learning (ML) can find apparent relationships, there has yet to be a systematic study examining when and why these apparent relationships diverge from the underlying intrinsic relationships found in the lab, and how and why this may depend on the method applied. Here we conduct a proof-of-concept study with three scenarios in which biomass is by construction a function of time-averaged phytoplankton growth rate. In the first scenario, the inputs and outputs of the intrinsic and apparent

relationships vary over the same monthly timescales. In the second, the intrinsic relationships relate averages of drivers that vary on hourly timescales to biomass, but the apparent relationships are sought between monthly averages of these inputs and monthly averaged output. In the third scenario we apply ML to the output of an actual Earth System Model (ESM). Our results demonstrated that when intrinsic and apparent relationships operate on the same spatial and temporal timescale, Neural Network Ensembles (NNEs) were able to extract the intrinsic

relationships when only provided information about the apparent relationships, while co-limitation and its inability to extrapolate, resulted in Random Forests (RF) diverging from the true response. When intrinsic and apparent relationships operated on different timescales (as little separation as hourly versus daily), NNEs fed with apparent relationships in time-averaged data produced responses with the right shape but underestimated the biomass. This was because when the intrinsic relationship was nonlinear, the response to a time-averaged input differed

systematically from the time-averaged response. Although the limitations found by NNEs were overestimated, they were able to produce more realistic shapes of the actual relationships compared to Multiple Linear Regression. Additionally, NNEs were able to model the interactions between predictors and their effects on biomass, allowing for a qualitative assessment of the co-limitation patterns and the nutrient causing the most limitation. Future research may be able to use this type of analysis for observational datasets and other ESMs to identify apparent relationships

between biogeochemical variables (rather than spatiotemporal distributions only) and identify interactions and co-limitations without having to perform (or at least performing fewer) growth experiments in a lab. From our study, it appears that ML can extract useful information from ESM output and could likely do so for observational datasets, as well.

## 1 Introduction

Phytoplankton growth can be limited by multiple environmental factors (Moore et al., 2013) such as macronutrients, micronutrients, and light. Limiting macronutrients include nitrogen (Eppley et al., 1973; Ryther and Dunstan, 1971; Vince and Valiela, 1973), phosphorus (Downing et al., 1999), and silicate (Brzezinski and Nelson, 1995; Dugdale et al., 1995; Egge and Aksnes, 1992; Ku et al., 1995; Wong and Matear, 1999). Limiting micronutrients can include iron (Boyd et al., 2007; Martin, 1990; Martin and Fitzwater, 1988), zinc, and cobalt (Hassler et al., 2012).

Additionally, limitations can interact with one another to produce co-limitations (Saito et al., 2008). Examples of this include the possible interactions between the micronutrients iron, zinc, and cobalt (Hassler et al., 2012) and the interaction between nitrogen and iron (Schoffman et al., 2016) such that local sources of nitrogen can have a strong influence on the amount of iron needed by phytoplankton (Maldonado and Price, 1996; Price et al., 1991; Wang and Dei, 2001). Spatial and temporal variations, such as mixed layer depth and temperature, affect such limitations, and

have been related to phytoplankton biomass using different functional relationships (Longhurst et al., 1995).

Limitations on phytoplankton growth are usually characterized in two ways – which we term intrinsic and apparent. Intrinsic relationships are those where the effect of one driver (nutrient/light) at a time is observed, while all others are held constant (often at levels where they are not limiting). An example of such intrinsic relationships is the

Michaels-Menten growth rate curves that emerge from laboratory experiments (Eppley and Thomas, 1969). Apparent relationships are those which emerge in the observed environment. An example of apparent relationships are those that emerge from satellite observations, which provide spatial distributions of phytoplankton on timescales (say a month) much longer than the phytoplankton doubling time, which can be compared against monthly distributions of nutrients. A significant challenge that remains is determining how intrinsic relationships found in the

laboratory scale up to the apparent relationships observed at the ecosystem scale (i.e., scaling the small to the large). Differences may arise between the two because apparent relationships reflect both intrinsic growth and loss rates, which are near balance over the long monthly timescales usually considered in climatological analyses. Biomass concentrations may thus not reflect growth rates. Differences may also arise because different limitation factors may not vary independently.


Earth System Models (ESMs) have proved valuable in linking intrinsic and apparent relationships. The intrinsic relationships are programmed into ESMs as equations that are run forward in time, and the output is typically provided as monthly averaged fields. The output of these ESMs is then compared against observed fields, such as

chlorophyll and nutrients, and can be analyzed to find apparent relationships between the two. If the ESM output is
close to the observations we find in nature, we say that the ESM is performing well. However, as recently pointed
out by Löptien and Dietze (2019), ESMs can trade-off biases in physical parameters with biases in biogeochemical
parameters (i.e., they can arrive at the same answer for different reasons). Using two versions of the UVic 2.9 ESM,
they showed that they could increase mixing (thus bringing more nutrients to the surface) while simultaneously
allowing for this nutrient to be more efficiently cycled – producing similar distributions of surface properties.
However, the carbon uptake and oxygen concentrations predicted by the two models diverged under climate change.
Similarly, Sarmiento et al. (2004) showed that physical climate models would be expected to produce different
spatial distributions of physical biomes due to differences in patterns of upwelling and downwelling, as well as the
annual cycle of sea ice. These differences would then be expected to be reflected in differences in biogeochemical
cycling, independent of differences in the biological models. These studies highlight the importance of constraining
not just individual biogeochemical fields, but also their relationships with each other.

To help with constraining these fields, some researchers have turned to machine learning (ML) to help in uncovering
the dynamics of ESMs. ML techniques are capable of fitting a model to a dataset without any prior knowledge of the
system and without any of the biases that may come from researchers about what processes are most important. As
applied to ESMs, ML has mostly been used to constrain physics parameterizations, such as longwave radiation
(Belochitski et al., 2011; Chevallier et al., 1998) and atmospheric convection (Brenowitz and Bretherton, 2018;
Gentine et al., 2018; Krasnopolsky et al., 2010, 2013; O'Gorman and Dwyer, 2018; Rasp et al., 2018).

With regard to phytoplankton, ML has not been explicitly applied within ESMs but has been used on phytoplankton
observations (Bourel et al., 2017; Flombaum et al., 2020; Kruk and Segura, 2012; Mattei et al., 2018; Olden, 2000;
Rivero-Calle et al., 2015; Scardi, 1996, 2001; Scardi and Harding, 1999) and has used ESM output as input for a ML
model trained on phytoplankton observations (Flombaum et al., 2020). Rivero-Calle et al. (2015) used random forest
(RF) to identify the drivers of coccolithophore abundance in the North Atlantic through feature importance measures
and partial dependence plots. The authors were able to find an apparent relationship between coccolithophore
abundance and environmental levels of $CO_2$, which was consistent with intrinsic relationships between
coccolithophore growth rates and ambient $CO_2$ reported from 41 laboratory studies. They also found consistency
between the apparent and intrinsic relationships between coccolithophores and temperature. While they were able to
find links between particular apparent relationships found with the RFs and intrinsic relationships between
laboratory studies, it remains unclear when and why this link breaks.


ML has been used to examine apparent relationships of phytoplankton in the environment (Flombaum et al., 2020;
Rivero-Calle et al., 2015; Scardi, 1996, 2001) and it is reasonable to assume that ML could find intrinsic

relationships when provided a new independent dataset from laboratory growth experiments. However, it has yet to be determined under what circumstances the apparent relationships captured by ML have significantly different functional forms to the intrinsic relationships that actually control phytoplankton growth.

To investigate when and why the link between intrinsic and apparent relationships break, we try to answer two main questions in this paper:

1. Can ML techniques find the correct underlying intrinsic relationships and, if so, what methods are most skillful in finding them?
2. How do you interpret the apparent relationships that emerge when they diverge from the intrinsic relationships we expect?

In addressing the first question, we first needed to demonstrate that we had a ML method that would correctly extract intrinsic relationships from apparent relationships. We constructed a simple model in which the biomass is directly proportional to the time-smoothed growth rate. In this scenario, intrinsic and apparent relationships operated on the same time and spatial scale and were only separated by a scaling factor, but the environmental drivers of phytoplankton growth had realistic inter-relationships. Having a better handle on the results from the first question, we were able to move onto the second question where we looked at where the link between intrinsic and apparent relationships diverged. We modified the first scenario so that the apparent relationships use a time-averaged input (similar to what would be used in observations), but the intrinsic relationships operate by smoothing growth rates derived from hourly input. Finally, we conduct a proof-of-concept study with real output from the ESM used to generate the inputs for scenarios 1 and 2, in which the biomass is a nonlinear function of the time-smoothed growth rate.

## 2 Methods

The main points of each Scenario are summarized in Table 1 including information on the predictors, target variable, equations used to calculate biomass, source file, and scenario description. For each of the three scenarios, three ML methods were used (Multiple Linear Regression [MLR], Random Forests [RF], and Neural Network Ensembles [NNE]).

### 2.1 Scenario 1: Closely related intrinsic and apparent relationships on the same timescale

In the first scenario, we wanted to determine how well different ML methods could extract intrinsic relationships when only provided information on the apparent relationships and when the intrinsic and apparent relationships were

operating on the same timescale. In this scenario, the apparent relationships between predictors and biomass were simply the result of multiplying the intrinsic relationships between predictors and growth rate by a scaling constant.

We designed a simple phytoplankton system in which biomass was a function of micronutrient, macronutrient, and light limitations based on realistic inter-relationships between limitations (Eq. 1):

$$B = S_* \times \min(L_{micro}, L_{macro}) \times L_{Irr} \tag{1}$$

where B is the value for biomass (mol kg$^{-1}$), S$_*$ is a scaling factor, and L$_{micro,macro,irr}$ are the limitation terms for micronutrient (micro), dissolved macronutrient (macro), and light (irradiance; irr), respectively. The scaling factor (1.9x10$^{-6}$ mol kg$^{-1}$) was used, so the resulting biomass calculation was in units of mol kg$^{-1}$. While simplistic, this is actually the steady-state solution of a simple phytoplankton-zooplankton system when grazing scales as the product of phytoplankton and zooplankton concentrations, and zooplankton mortality is quadratic in the zooplankton

concentration.

Each of the nutrient limitation terms ($L_{micro,macro}$ in Eq. 1) were functions of Michaelis-Menten growth curves (Eq. 2):

$$L_N = \frac{N}{K_N + N} \tag{2}$$

where L$_N$ is the limitation term for the respective factor, N is the concentration of the nutrient, and K$_N$ is the half-

saturation constant specific to each limitation. The light limitation was given by (Eq. 3):

$$L_{Irr} = 1 - e^{-\left(\frac{Irr}{K_{Irr}}\right)} \tag{3}$$

where L$_{Irr}$ is the light limitation term, Irr is the light intensity, and K$_{Irr}$ is the light limitation constant. In terms of our nomenclature, Eq. 1 defines the apparent relationship between nutrients, light, and biomass, such as might be found in the environment, while Eq. 2 and 3 are the intrinsic relationships between nutrients/light and growth rate, such as might be found in the laboratory or coded in an ESM.


For the concentrations of each factor (*N* in Eq. 2), we took the monthly averaged value for every lat/lon pair (i.e., 12 monthly values for each lat/lon pair) from the Earth System Model ESM2Mc (Galbraith et al., 2011). ESM2Mc is a fully coupled atmosphere, ocean, sea ice model into which is embedded an ocean biogeochemical cycling module. Known as BLING (Biogeochemistry with Light, Iron, Nutrients, and Gases; Galbraith et al., 2010), this module

carries a macronutrient, a micronutrient, and light as predictive variables and uses them to predict biomass using a highly parameterized ecosystem (described in more detail below). The half-saturation coefficients (*K$_N$* in Eq. 2) for

the macronutrient and micronutrient were also borrowed from BLING with values of $1 \times 10^{-7}$ mol kg$^{-1}$ and $2 \times 10^{-10}$ mol kg$^{-1}$, respectively. The light-limitation coefficient $K_{Irr}$ was set at 34.3 W m$^{-2}$, which was the global mean for the light limitation factor in the ESM2Mc simulation used later in this paper.


The final dataset consisted of three input/predictor variables and one target term with a total of 77,328 observations. The input variables given to each of three ML methods (Multiple Linear Regression [MLR], Random Forests [RF], and Neural Network Ensembles [NNE], described in more detail below) were the concentrations (not the limitation terms) for the micronutrient, macronutrient, and light. The target variable was the biomass we calculated from Eq. 1-170 3. The same three ML methods were applied to all three Scenarios.

The dataset was then randomly split into training and testing datasets, with 60% of the observations going to the training dataset and the remainder going to the testing dataset. This provided a standard way to test the generalizability of each ML method by presenting them with new observations from the test dataset and ensuring the 175 models did not overfit the data. The input and output values for the training dataset were used to train a model for each ML method. Once each method was trained, we provided the trained models with the input values of the testing dataset to acquire their respective predictions. These predictions were then compared to the actual output values of the test dataset. To assess model performance, we calculated the coefficient of determination ($R^2$) and the root mean squared error (RMSE) between the ML predictions and the actual output values for the training and testing datasets.


Following this, a sensitivity analysis was performed on the trained ML models. We allowed one predictor to vary across its min-max range while holding the other two input variables at specific percentile values. This was repeated for each predictor. This allowed us to isolate the impact of each predictor on the biomass – creating "cross-sections" of the dataset where only one variable changed at a time. For comparison, these values were also run through Eq. 1-185 3 to calculate the true response of how the simple phytoplankton model would behave. This allowed us to view which of the models most closely reproduced the underlying intrinsic relationships of the simple phytoplankton model.

For our main sensitivity analyses, we chose to hold the predictors that were not being varied at their respective 25th, 190 50th, and 75th percentile values. We chose to use these particular percentile values for several reasons:

1. It allowed us to avoid the extreme percentiles (1st and 99th). As we approach these extremes, the uncertainty in the predictions grows quite rapidly because of the lack of training samples within that domain space of the dataset. For example, there are no observations which satisfy the conditions of being in the 99th percentile of two variables simultaneously. This extreme distance outside of the training domain generally

leads to standard deviations in predictions that are too large to provide a substantial level of certainty about the ML model's predictions.

2.    Similar to the idea that we can avoid the extremes, we also chose these values as they are quite typical values for the edges of box plots. Generally, values within the range of the 25th to 75th percentiles are not considered outliers. Along those lines, we wanted to examine the conditions in a domain space that are

likely to be found in actual observational datasets, with the reasoning that if there was high uncertainty in the ML predictions at these more moderate levels, there would be even higher uncertainty towards the extremes.

This method of sensitivity analysis contrasts with partial dependence plots (PDPs), which are commonly used in ML

visualization. PDPs show the marginal effect that predictors have on the outcome. They consider every combination of the values for a predictor of interest and all values of the other predictors, essentially covering all combinations of the predictors. The predictions of a model are then averaged and show the marginal effect of a predictor on the outcome – creating responses moderately comparable to averaged cross-sections. Because of this averaged response, PDPs may hide significant effects from subgroups within a dataset. A sensitivity analysis avoids this disadvantage

by allowing separate visualization of subgroup relationships. For example, if macronutrient is the primary limiter over half of the domain, but not limiting at all over the other half, PDPs of the biomass dependence on micronutrient will reflect this macronutrient limitation, while a sensitivity analysis at the 75th percentile of macronutrient will not.

Using the predictions produced from the sensitivity analyses, we also computed the half-saturation constants for each curve. A limitation of observational data is the frequency of sampling, which limits the ability to estimate half-saturation coefficients without performing growth experiments in a lab. Calculating the half-saturation constants from the sensitivity analysis predictions allowed us to investigate if ML methods could provide a quantitative estimate from the raw observational data. The half-saturation constants were determined by fitting a non-linear

regression model to each sensitivity analysis curve matching the form of a Michaelis-Menten curve (Eq. 4):

$$B = \frac{\alpha_1 N}{\alpha_2 + N} \tag{4}$$

where B corresponds to the biomass predictions from the sensitivity analyses, N represents the nutrient concentrations from the sensitivity analyses, and $\alpha_1$ and $\alpha_2$ are the constants that are being estimated by the non-linear regression model. The constant $\alpha_2$ was taken as the estimation of the half-saturation coefficient for each sensitivity analysis curve.


Since co-limitations can affect the calculation of half-saturation coefficients, we also created interaction plots. This is useful because trying to calculate the half-saturation constant based on a nutrient curve that is experiencing limitation by another nutrient could cause the calculation to be underestimated. The interaction plots are a form of sensitivity analysis where two predictor variables are varied across their min-max range, rather than one. This

produces a mesh of predictor pairs covering the range of possible combinations of two predictors. With these interaction plots, it was possible to visualize the interaction of two variables and their combined effect on the target variable. For each pair of predictors that were varying, we set the other predictor that was not varying to its 50th percentile (median) value. As with the sensitivity analysis for single predictors, these predictor values were run through Eq. 1-3 so a comparison could be made as to which method most closely reproduced the true variable

interactions.

### 2.2 Scenario 2: Distantly related intrinsic and apparent relationships on different timescales

In Scenario 1, the intrinsic relationships between environmental conditions and growth rate and apparent relationships between environmental conditions and biomass differed only by a scale factor and operated at the same

timescale. In reality, input variables (such as light) vary on hourly timescales so that growth rates vary on similar timescales. Biomass reflects the average of this growth rate over many hours to days, while satellite observations and ESM model output are often only available on monthly averaged timescales. So the reality is that even if a system is controlled by intrinsic relationships, the apparent relationships gained from climatological variables on long timescales will not reproduce these intrinsic relationships since the average light (irradiance) limitation is not

equal to the limitation given the averaged light value (Eq. 5).

$$\overline{L_{Irr}} = \overline{\left(1 - e^{-\left(\frac{Irr}{K_{Irr}}\right)}\right)} \neq 1 - e^{-\left(\frac{\overline{Irr}}{K_{Irr}}\right)} \tag{5}$$

where the overbar denotes a time-average, and Irr stands for irradiance (light). For Scenario 2, we wanted to investigate how such time averaging biased our estimation of the intrinsic relationships from the apparent ones; i.e., how does the link between the intrinsic and apparent relationships change with different amounts of averaging over time?


For the short timescale intrinsic relationships, we took daily inputs for the three predictor variables for one year from the ESM2Mc model. We further reduced the timescale from days to hours to introduce daily variability for the irradiance variable relative to the latitude, longitude, and time of year (Eq. 6).

$$Irr_{Int}(t) = \frac{12\pi Irr_{daily}}{T_{Day}} \sin\left(\frac{\pi(t - t_{Sunrise})}{T_{Day}}\right) \text{ when } 0 < t < T_{Day} \tag{6}$$

where Irr$_{Int}$ is the hourly interpolated value of irradiance, Irr$_{daily}$ is the **daily-mean** value of irradiance, t is the hour of the day being interpolated, t$_{Sunrise}$ is the hour of sunrise, and T$_{Day}$ is the total length of the day. The resulting curve preserves the day-to-day variation in the daily mean irradiance due to clouds and allows a realistic variation over the course of the day. The hourly values for the micronutrient and macronutrient were assigned using a standard interpolation between each of the daily values. Thus, light was the only predictor variable that varied hourly. These hourly interpolated values were then used to calculate an "hourly biomass" from Eq. 1-3. Note that we are not claiming real-world biomass would be zero at night but assume that on a long enough timescale, it should approach the average of the hourly biomass.

To simulate apparent relationships, we smoothed the hourly values for both biomass and the input variables into daily, weekly, and monthly averages for each lat/lon point. To reiterate, the intrinsic and apparent relationships in Scenario 2 differed in timescales, but not in spatial scales. Each dataset was then analyzed following steps similar to those outlined in Scenario 1; constructing training and testing datasets, using the same variables as inputs to predict the output (biomass), and using the same ML methods. To assess each method's performance, we calculated the R$^2$ value and the RMSE between the predictions and observations for the training and testing datasets. We also performed a sensitivity analysis, calculated half-saturation constants, and created interaction plots similar to those described above.

### 2.3 Scenario 3: BLING biogeochemical model

As a demonstration of their capabilities, the ML methods were also applied directly to monthly averaged output from the BLING model itself using the same predictors in Scenarios 1 and 2, but using the biomass calculated from the actual BLING model. As described in Galbraith et al. (2010), BLING is a biogeochemical model where biomass is diagnosed as a non-linear function of the growth rate smoothed in time. The growth rates, in turn, have the same functional form as in Scenarios 1 and 2, namely (Eq. 7):

$$\mu = \mu_0 * \exp(k * T) * \min\left(\frac{N_{micro}}{K_{micro} + N_{micro}}, \frac{N_{macro}}{K_{macro} + N_{macro}}\right) \times \left(1 - \exp\left(-\frac{Irr}{Irr_K}\right)\right) \tag{7}$$

where the first exponential parameterizes temperature-dependent growth following Eppley (1972), $N_{macro,micro}$ are the macronutrient and micronutrient concentrations, $K_{macro,micro}$ are the half-saturation coefficients for the macronutrient and micronutrient, Irr is the irradiance, and $Irr_k$ is a scaling for light limitation. An important difference (to which we will return later in the manuscript) is that the light limitation term is calculated using a variable Chl:C ratio following the theory of Geider et al. (1997). The variation of the Chl:C ratio would correspond to a $K_{Irr}$ in Scenarios 1 and 2 which adjusts in response to both changes in irradiance (if nutrient is low) or changes in nutrient (if irradiance is high), as well as changes in temperature. Given the resulting growth rate $\mu$ the total biomass then asymptotes towards (Eq. 8)

$$B = \left(\frac{\tilde{\mu}}{\lambda} + \frac{\tilde{\mu}^3}{\lambda^3}\right) S_* \tag{8}$$

where $\lambda = \lambda_0 \exp(k * T)$ is a grazing rate, the tilde denotes an average over a few days and $S_*$ is the biomass constant that we saw in the previous two scenarios. Note that because grazing and growth have the same temperature dependence, the biomass then ends up depending on the nutrients and light in a manner very similar to Scenarios 1 and 2. Growth rates and biomass are then combined to drive the uptake and water-column cycling of micronutrient

and macronutrient within a coarse-resolution version of the GFDL ESM2M fully coupled model (Galbraith et al., 2011), denoted as ESM2Mc.

As described in Galbraith et al. (2011) and Bahl et al. (2019), ESM2Mc produces relatively realistic spatial distributions of nutrients, oxygen, and radiocarbon. Although simpler in its configuration relative to models such as

TOPAZ (Tracers of Ocean Productivity with Allometric Zooplankton; Dunne et al., 2013), it has been demonstrated that in a higher-resolution physical model BLING produces simulations of mean nutrients, anthropogenic carbon uptake, and oceanic deoxygenation under global warming that are almost identical to such complicated models (Galbraith et al., 2015).

We chose to use BLING for three main reasons. The first is that we know it produces robust apparent relationships between nutrients, light, and biomass by construction – although these relationships can be relatively complicated – particularly insofar as iron and light co-limitation is involved (Galbraith et al., 2010). As such, it represents a reasonable challenge for a ML method to recover such non-linear relationships. The second is that we know how these relationships are determined by the underlying intrinsic relationships between limiting factors and growth.

Models with more complicated ecosystems (including explicit zooplankton and grazing interactions between functional groups) may exhibit more complicated time-dependence that would confuse such a straightforward linkage between phytoplankton growth limitation and biomass. The third is that despite its simplicity, the model has relatively realistic annual mean distributions of surface nutrients, iron, and chlorophyll, and under global warming, it simulates changes in oxygen and anthropogenic carbon uptake that are similar to much more complicated ESMs

(Galbraith et al., 2015).

**2.4 ML Algorithms**

We chose to use Random Forests (RFs) and Neural Network Ensembles (NNEs) in this manuscript. Although other ML methods exist, the list of possible choices is rather long. It was decided that the number of ML algorithms being

compared would be limited to RFs and NNEs, given their popularity in studying ecological systems. Additionally, we chose to compare the performance of the ML techniques to the performance of Multiple Linear Regression

(MLR), which allows us to quantify the importance of nonlinearity. It should be noted that we are not trying to suggest that MLR is always ineffective for studying ecological systems. MLR is a very useful and informative approach for studying linear relationships within marine ecological systems (Chase et al., 2007; Harding et al., 2015; 320   Kruk et al., 2011).

### 2.4.1 Random Forests

RFs are an ensemble ML method utilizing many decision trees to turn "weak learners" into a single "strong learner" by averaging multiple outputs (Breiman, 2001). In general, RFs work by sampling (with replacement) about two-325   thirds of a dataset and constructing a decision tree. This process is known as bootstrap aggregation. At each split, the random forest takes a random subset of the predictors and examines which variable can be used to split a given set of points into two maximally distinct groups. This use of random predictor subsets helps to ensure the model is not overfitting the data. The process of splitting the data is repeated until an optimal tree is constructed or until the stopping criteria are met, such as a set number of observations in every branch (then called a leaf / final node). The 330   process of constructing a tree is then repeated a specified number of times, which results in a group (i.e., "forest") of decision trees. Random forests can also be used to construct regression trees in which a new set of observations traverse each decision tree with its associated predictor values and the result from each tree is aggregated into an averaged value.

Here, we used the same parameters for RF in the three scenarios to allow for a direct comparison between the scenarios and to minimize the possible avenues for errors. Each RF scenario was implemented using the TreeBagger function in Matlab 2019b, where 500 decision trees were constructed with each terminal node resulting in a minimum of five observations per node. An optimization was performed to decide the number of decision trees that minimized the error while still having a relatively short runtime of only several minutes. For additional details about 340   the construction and training of the RFs, please see Appendix B.

### 2.4.2 Neural Network Ensembles

Neural networks (NNs) are another type of ML that has become increasingly popular in ecological applications (Flombaum et al., 2020; Franceschini et al., 2019; Guégan et al., 1998; Lek et al., 1996a, 1996b; Mattei et al., 2018; 345   Olden, 2000; Özesmi and Özesmi, 1999; Scardi, 1996, 2001; Scardi and Harding, 1999). Scardi (1996) used NNs to model phytoplankton primary production in the Chesapeake and Delaware Bays. Lek et al. (1996b) demonstrated the ability of NNs to explain trout abundance using several environmental variables through the use of the "profiling" method, a type of variable importance metric that averages the results of multiple sensitivity analyses to acquire the importance of each variable across its range of values.


Feed-forward NNs consist of nodes connected by weights and biases with one input layer, (usually) at least one hidden layer, and one output layer. The nodes of the input layer correspond to the input values of the predictor variables, and the hidden and output layer nodes each contain an activation function. Each node from one layer is connected to all other nodes before and after it. The values from the input layer are transformed by the weights and

biases connecting the input layer to the hidden layer, put through the activation function of the hidden layer, modified by the weights and biases connecting the hidden layer to the output layer, and finally entered into the final activation function of the output node.

The output (predictions) from this forward pass through the network is compared to the actual values, and the error

is calculated. This error is then used to update the weights with a backward pass through the network using backpropagation. The process is repeated a specified number of times or until some optimal stopping criteria are met, such as error minimization or validation checks where the error has increased a specified number of times. For a more in-depth discussion of NNs, see Schmidhuber (2015).

For this particular study, we use neural network ensembles (NNEs), which are a collection of NNs (each of which uses a subsample of the data) whose predictions are averaged into a single prediction. It has been demonstrated that NNEs can outperform single NNs and increase the performance of a model by reducing the generalization error (Hansen and Salamon, 1990).

To minimize the differences between scenarios, we used the same framework for the NNs in each scenario. Each NN consisted of three input nodes (one for each of the predictor variables), 25 nodes in the hidden layer, and one output node. The activation function within the hidden nodes was a hyperbolic tangent sigmoid function, and the activation function within the output node used a linear function. The stopping criteria for each NN was set as a validation check, such that the training stopped when the error between the predictions and observations increased

for six consecutive epochs. An optimization was performed to decide the number of nodes in the hidden layer that minimized the error while maintaining a short training time. A sensitivity analysis was also performed using different activation functions to ensure the choice of activation function had minimal effect on the outcome. Furthermore, another sensitivity analysis was performed to ensure additional hidden layers were not necessary. The details of the optimization and sensitivity analyses to determine the NN parameters can be found in Appendix B.


Each NNE consisted of ten individual NNs, and each NN was trained using the feedforwardnet function in Matlab 2019b.

Each variable was scaled between -1 and 1 based on its respective maximum and minimum (Eq. 9).

$$V_S = \frac{max_S - min_S}{max_U - min_U}(V_U - min_U) + min_S$$ (9)

where V is the value of the variable being scaled, S stands for the scaled value (min is -1 and max is 1), and U represents the unscaled value. This step ensures that no values are too close to the limits of the hyperbolic tangent sigmoid activation function, which would significantly increase the training time of each NN. Additionally, this normalization ensures that each predictor falls within a similar range, so more weight is not provided to variables with larger ranges. Although scaling is not necessary for RF and MLR, the scalings used for the NNE were still applied to each method for consistency. The results presented in this paper were then transformed back to their original scales to avoid confusion from scaling (Eq. 10).

$$V_U = \frac{max_U - min_U}{max_S - min_S}(V_S - min_S) + min_U$$ (10)

Where the letters represent the same values as in Eq. 9.

**3 Results and Discussion**

**3.1 Scenario 1: Closely related intrinsic and apparent relationships on the same timescale**

In the first scenario, our main objective was to determine if ML methods could extract intrinsic relationships when given information on the apparent relationships and reasonable spatiotemporal distributions of co-limitation when the intrinsic and apparent relationships were operating on the same timescale.

In Scenario 1, the RF and NNE both outperformed the MLR as demonstrated by higher $R^2$ values and lower RMSE (Table 2). The MLR captured just under half of the variance ($R^2 = 0.44$-$0.45$; Table 2), while the RF and NNE essentially captured all of it ($R^2 > 0.99$; Table 2). The decreased performance of the MLR is not inherently surprising, given the non-linearity of the underlying model, but it does demonstrate that the range of nutrients and light produced as inputs by ESM2Mc are capable of producing a non-linear response. Additionally, each method showed similar performances between the training and testing datasets suggesting adequate capture of the model dynamics in both datasets.

From the spatial distributions and error plots of the true response and the predictions from each method, it can be observed that the RF and NNE showed the closest agreement with the true response (Fig. 1). The NNE showed the

lowest error and closest agreement with the true response (Fig. 1 g), followed closely by the RF with slightly higher errors (Fig. 1 f). Additionally, the RF and NNE were able to reproduce the biomass patterns in the Equatorial Atlantic and Pacific, along with the low biomass concentrations at higher latitudes (Fig. 1 a, c, d). Although MLR was able to reproduce the general trend of the highest biomass in the low latitudes and low biomass in the high latitudes, it was not able to predict higher biomass values (Fig. 1 b) and it exhibited the highest errors of the three
methods (Fig. 1 e).

In addition to examining whether the different ML methods matched the correct response, we also interrogated these methods to look at how different predictors contributed to the answer, and whether these contributions matched the intrinsic relationships between the predictors and biomass as we had put into the model (Fig. 2). The MLR (red
dashed lines) showed very little response to changes in macronutrient (Fig. 2 a, d, g), an unrealistic negative response to increases in micronutrient (Fig. 2 b, e, h), and a reasonable (albeit linear) match to the light response (Fig. 2 c, f, i). By contrast, the response to any predictor for the NNE (green dashed lines) showed agreement with the true response of the model (black lines) in all circumstances, insofar as the true response was always within the standard deviation of the NNE predictions (Fig. 2).


The RF prediction of the response to a given predictor (blue dashed lines) showed agreement with the true response when the other predictors were fixed at the lower percentiles (Fig. 2 a-c), but began deviating in the higher percentiles (Fig. 2 d-i). This was likely due to the range of the training dataset and how RFs acquire their predictions. When presented with predictor information, RFs rely on the information contained within their training
data. If they are presented with predictor information that goes outside the range of the dataspace of the training set, RFs will provide a prediction based within the range of the training set. When performing the sensitivity analysis, the values of the predictors in the higher percentiles were outside the range of the training dataset. For example, RF deviates from the true response as the concentration of the macronutrient increases – actually decreasing as nutrient increases despite the fact that such a result is not programmed into the underlying model (Fig. 2 g). Although there
may be observations in the training dataset where the light and micronutrient are at their 75th percentile values when the macronutrient is low, there likely are not any observations where high levels of the macronutrient, micronutrient, and light are co-occurring. Without any observations meeting that criteria, the RF provided the highest prediction it could based on the training information.

In contrast to the RF's inability to extrapolate outside the training range, the NNE showed its capability to make predictions on observations on which it was not trained (Fig. 2). Note, however, that while we have programmed Michaelis-Menten intrinsic dependencies for individual limitations into our model, we did not get Michaelis-Menten type curves back for macro- and micronutrients when the other variables were set at low percentiles (Fig. 2 a-c). The

reason is that Liebig's law of the minimum applies to the two nutrient limitations. When the micronutrient is low, it
prevents the entire Michaelis-Menten curve for the macronutrient from being seen.

Although the NNEs captured the true intrinsic relationships, we could not interpret these curves without
remembering that multiple limitations affect biomass. For example, when we computed an estimated half-saturation
for the nutrient curves in the top row of Fig. 2, we calculated values for $K_N$ that were far lower than the actual ones
specified in the model (Table 3). The estimated half-saturation when other predictors were held at their 25th
percentile for the micro- and macronutrient were underestimated by one and two orders of magnitude, respectively.
When higher percentiles were used (Table 4), the estimated half-saturation was overestimated for some predictors
and underestimated for others. At the 99th percentile, the macronutrient half-saturation was underestimated by 49%
and micronutrient and light were overestimated by 77% and 36%, respectively (Table 4). It is possible that even at
the higher percentiles, micronutrient was still exerting some limitation on the macronutrient curve which would
explain why the estimate for the macronutrient half-saturation was underestimated. However, this does not explain
why the estimations for the micronutrient and light half-saturations were overestimated by so much. Although the
ability to calculate half-saturation coefficients from the sensitivity analysis curves seemed to be a way to quantify
the accuracy of the ML predictions, co-limitations lead to high uncertainties in the estimates. While mathematically
obvious, this result has implications for attempts to extract (and interpret) $K_N$ from observational datasets, such that
one would expect colimitation to produce a systematic underestimation of $K_N$.

In an effort to visualize the co-limitations and to investigate the extent to which any of the methods could reproduce
these interactions, we examined the interaction plots (Fig. 3). MLR expectedly predicted linear relationships in
which higher concentration pairs of irradiance/macronutrient and irradiance/micronutrient lead to higher biomass
(Fig. 3 h, i), but it incorrectly predicted the interaction between the micro- and macronutrient such that decreasing
concentrations of macronutrient lead to higher biomass (Fig. 3 g). Note that the x and y axes in Fig. 3g were
switched relative to the other subplot axes, which was necessary to visualize the interaction. RF incorrectly
predicted the highest concentrations of biomass at moderate levels of the micro- and macronutrient in their
interactions with irradiance (Fig. 3 k, l). RF again incorrectly predicted the greatest biomass in the
micro/macronutrient interaction occurring at low levels of micronutrient across most levels of macronutrient (Fig. 3
j). The NNE was the only method that was able to reproduce the interactions of the model (Fig. 3 d-f, m-o).
Although the NNE overestimated the biomass prediction when concentrations were high for both predictors in the
irradiance/micronutrient and irradiance/macronutrient interactions (Fig. 3 e, f, n, o), these were also the areas of the
dataspace without any observations to constrain the NNE (Fig. 3 b, c). Similar to the sensitivity analyses for single
predictors, the NNE was capable of extrapolating outside the range of the training dataset while RF was not.

The NNE interaction plots (Fig. 3 m-o) bear resemblance to the co-limitation plots seen in Fig. 2 of Saito et al. (2008) and allowed for a qualitative comparison of the type of co-limitation that two predictors have on the target variable. For example, the micro/macronutrient interaction in Fig. 3m shows the same type of response as would be expected in Liebig minimizing (Saito et al., 2008 Fig. 2C). This result is what we would expect given that the equations for Scenario 1 (Eq. 1-3) were Liebig minimizing by construction between the macro- and micronutrient. Additionally, Liebig minimizing can be seen in the pattern displayed in the interaction plot of the true expected response (Fig. 3 d).

The interactions of macronutrient/irradiance (Fig. 3 n) and micronutrient/irradiance (Fig. 3 o) mirrored the co-limitation pattern of Independent Multiplicative Nutrients (Saito et al., 2008 Fig. 2B) where neither predictor was limiting and the effects of the two predictors have a multiplicative effect on the target variable. This was again consistent with the equations that govern Scenario 1 (Eq. 1-3). In Eq. 1, the irradiance limitation was only multiplied by the lesser limitation of the macro- and micronutrient and did not show a pattern of Liebig minimizing. It was interesting that the macronutrient/irradiance interaction (Fig. 3 n) almost appeared to display a pattern of no co-limitation (Saito et al., 2008 Fig. 2A), but this stark increase in the biomass past low concentrations of the macronutrient can be partially explained by the contour plot of observations (Fig. 3 b; please see Fig. C1 in Appendix C for individual box plots of the predictor and target variables). The majority of observations where macronutrient concentrations were low had a correspondingly high value for irradiance. Additionally, when the macronutrient passed a certain concentration (which happened to be very low in these conditions), the micronutrient became the limiting nutrient, such that light was the only variable that then affected the biomass (data not shown).

With respect to our main objective for Scenario 1, it was evident that only the NNE was able to extract the intrinsic relationships from information on the apparent relationships. This was due in large part to its capability of extrapolating outside the range of the training dataset, whereas RFs were constrained by training data, and MLR was limited by its inherent linearity and simplicity. Furthermore, the attempts to quantify the half-saturation coefficients from the sensitivity analysis curves proved unreliable because of nutrient co-limitations. However, we were able to use interaction plots to qualitatively describe the type of co-limitation occurring between each pair of predictors and support the result from the single predictor sensitivity analyses that micronutrient was most limiting in many situations.

### 3.2 Scenario 2: Distantly related intrinsic and apparent relationships on different timescales

In Scenario 1, the intrinsic and apparent relationships were simply related by a scaling factor. In practice, the relationships are more difficult to connect to each other. For the second scenario, both the output biomass and predictors (light, macronutrient, and micronutrient) were averaged over daily, weekly, and monthly timescales. Our

main objective was to investigate how the link between intrinsic and apparent relationships changed when using climatologically averaged data – as is generally the case for observational studies.

As in Scenario 1, the RF and NNE outperformed the MLR based on the performance metrics for the daily, weekly, and monthly time-averaged scenarios (Table 2), with linear models only able to explain about 30% of the variance. The comparable performances between the training and testing datasets suggested a sufficient sampling of the data for each method to capture the dynamics of the underlying model.

Examining the monthly apparent relationships found for each method and comparing them to the true intrinsic relationships showed that none of the methods were able to reproduce the true intrinsic relationships – in general systematically underestimating biomass at high levels of light and nutrient (Fig. 4). The one exception was the 25$^{th}$ percentile plot of the micronutrient (Fig. 4b). The underestimation was consistent across the different timescales, and the sensitivity analysis showed little difference in the predicted relationships between the daily, weekly, and

monthly averaged timescales for the NNEs (Fig. 5). Because the NNEs showed the closest approximations to the correct shape and magnitude of the curves compared to RF and MLR (Fig. 4), the remaining analysis of Scenario 2 is mainly focused on NNEs.

   The underestimation was not entirely unexpected. The averaging of the hourly values into daily, weekly, and

monthly timescales quickly lead to a loss of variability (Fig. 6), especially for light (Fig. 6c). A large portion of the variability was lost in the irradiance variable going from hourly to daily (Fig. 6c). The loss of variability meant that the light limitation computed from the averaged light was systematically higher than the averaged light limitation. To match the observed biomass, the asymptotic biomass at high light would have to be systematically lower (see Appendix A for the mathematical proof). Differences were much smaller for macronutrient and micronutrient as

they varied much less over the course of a month in our dataset. Our results emphasize that when comparing apparent relationships in the environment to intrinsic relationships from the laboratory, it is essential to take into account which timescales of variability that averaging has removed. Insofar as most variability is at hourly time scales, daily-, weekly-, and monthly-averaged data will produce very similar apparent relationships (Fig. 5). But if there was a strong week-to-week variability in some predictor, this may not be the case.


   To understand how the apparent relationships were changing across different timescales, we averaged the hourly dataset over a range of hourly timespans. Specifically, we averaged over the timescales of 1-hour (original hourly set), 2, 3, 4, 6, 8, 12, 24, 48, 72, 168 (weekly), and 720 (monthly) hours. This new set of averaged timescales was then used to train NNEs with one NNE corresponding to each averaged timescale. We then performed sensitivity

analyses on each of the trained NNEs to see the apparent relationships for each averaged timescale and set the

percentile vales for the other variables at their 50th percentile (median). For more details about this method, please

see Appendix D. To visualize all the timescales at once, we plotted them on surface plots (Fig. 7). The greatest

changes in the apparent relationships occurred in the first 24 hours (Fig. 7 b, d, f). Furthermore, when focused on the

first 24 hours, the apparent relationships below 12 hours were relatively close to the hourly apparent relationships

(Fig. 7 a, c, e) suggesting that a large portion of the variability may have been lost between the 12- to 24-hour

averaged datasets. It may be possible to use this type of diagnostics test to find the sampling frequency which would

be needed to recover true relationships in other datasets or to see how relationships change over different timescales.

Although we only averaged time in Scenario 2, this diagnostics test could also be applied to datasets that are

averaged in space only or in space and time.


Even though in Scenario 1 we showed estimating the half-saturation coefficients from the sensitivity analysis curves

can be unreliable, we felt that it could be helpful to include them in this manuscript so other researchers who may

have a similar idea in the future can be cautioned against it. It was not surprising that the estimated half-saturation

coefficients for Scenario 2 were also incorrect (Tables 3 and 4). The inaccuracies in Scenario 2 though were likely

the result of co-limitations and averaging, whereas Scenario 1 only dealt with co-limitations. Furthermore, even

though the predicted curves for the daily, weekly, and monthly NNEs were relatively similar (Fig. 5), the estimated

half-saturations varied quite a bit between them (Table 3). This was even more pronounced for the half-saturation

estimates at the 97th, 98th, and 99th percentiles (Table 4). For example, the estimated half-saturation for light from the

daily-NNE at these upper percentiles was an entire order of magnitude higher than the actual value (Table 4).


As with Scenario 1, we visualized the variable interactions in Scenario 2 with interaction plots and compared these

to the colimitation plots in Fig. 2 of Saito et al. (2008). As we observed in Scenario 1, the interaction plots showed

that when the NNEs were tasked with making predictions outside the range of their dataset, their predictions could

be drastically over or underestimated (Fig. 8 d-l) because no observations existed in that space to constrain the

NNEs (Fig. 9). For example, in the irradiance/micronutrient plot (Fig. 8 l) when high irradiance coincided with high

micronutrient concentrations, the NNE predicted a rapid increase in the biomass prediction. From Fig. 9i, which

shows the density plot of the observations for irradiance and micronutrient, it can be seen that this same area was far

outside the range of the dataset where there were no observations to constrain the NNE.

Each of the NNEs for the daily, weekly, and monthly-averaged datasets showed similar co-limitation patterns (Fig. 8

d-l) which also agreed with the patterns of the true interactions (Fig. 8 a-c). The macronutrient/micronutrient

interaction plots (Fig. 8 d, g, j) exhibited a pattern of Liebig minimizing as shown in Fig. 2C of Saito et al. (2008).

The irradiance/macronutrient (Fig. 8 e, h, k) and irradiance/micronutrient (Fig. 8 f, i, l) interaction plots show a co-

limitation pattern consistent with Independent Multiplicative Nutrients (Saito et al., 2008 Fig. 2B). These interaction
patterns are the same interaction patterns observed in Scenario 1. Once again, these patterns would be expected
because the equations contain these patterns, by construction. Surprisingly, these patterns held across time-averaging
even as great as one month (720 hours). Although the monthly interaction underestimated the biomass, the general
pattern, non-linearity, and interaction of the variables remained consistent across the different timescales. This could
imply that the use of monthly-mean observations could still allow researchers to identify interactions that hold true
at timescales as small as one hour.

Regarding our main objective for Scenario 2 to understand how the link between intrinsic and apparent relationships
changed, only the NNEs were able to provide reliable information. The sensitivity analysis with individual
predictors showed that variability could be lost in the span of a single day when considering information on hourly
timescales. This caused an underestimation of the biomass values for timescales that were averaged over ranges
greater than and equal to 24 hours. However, it was possible to visualize how the relationships changed from the
hourly data to the 720-hour (monthly) data by training NNEs on different timescales of the data. Additionally, the
interaction patterns observed in Scenario 1 where the intrinsic and apparent relationships were closely related were
also observed in the interaction patterns of Scenario 2 where the intrinsic and apparent relationships were distantly
related. This suggested that it may be possible to capture variable interactions occurring at small timescales, even
when data is sampled at a frequency as infrequent as once per month.

### 3.3 Scenario 3: BLING biogeochemical model

When run in the full ESM, the BLING biogeochemistry does end up producing surface biomass which is a strong
function of the growth rate (Fig. 10a) with a non-linear relationship as in Eq. 8. As the growth rate, in turn, is given
by Eq. 7, we can also examine how the monthly mean limitation terms for nutrient and light compare with the means
given by computing the limitations with monthly mean values of nutrients, $Irr$, and $Irr_k$. As shown in Fig. 10b, the
nutrient limitation is relatively well captured using the monthly mean values, although there is a tendency for the
monthly means to underestimate moderate values of nutrient limitation. Further analysis shows that this is due to the
interaction between micro- and macronutrient limitation – with the average of the minimum limitation being
somewhat higher than the minimum of the average limitation. However, using the actual monthly mean values of
$Irr$, and $Irr_k$ (Fig. 10c) causes the light limitation to be systematically biased high.

To demonstrate their capabilities, NNEs were applied directly to the monthly averaged output of one of the BLING
simulations. The main purpose of the final scenario was to demonstrate the capabilities of NNEs when applied to
actual ESM output with the reasoning that if it was unable to provide useful information on BLING (in which, by
definition, the biomass and limitations are closely related), it would also fail on more complex models.

Scenario 3 showed similar results to those of Scenarios 1 and 2, with respect to the performance metrics of the
training and testing datasets (Table 2), the inaccuracy of the estimated half-saturation coefficients (Tables 3 and 4),
and deviations in the interaction plots where no observations occur (Fig. 12). The performance metrics for Scenario
3 showed performances between the training and testing datasets indicating sufficient sampling of the data (Table 2).
Additionally, the half-saturation coefficients were included here (Tables 3 and 4) for the same reasons as stated in
Section 3.2 for Scenario 2. The largest deviation in the interaction plots occurred in the macronutrient/irradiance plot
when both macronutrient and light concentrations were near their maximum (Fig. 12 e). However, this was not
surprising since no observations existed in that range to constrain the NNE (Fig. 12 b; please see Fig. C2 in
Appendix C for individual box plots of the predictor and target variables).

In the sensitivity analysis, the macronutrient and light plots (Fig. 11 a, c, d, f, g, i) exhibited curves consistent with
colimitation where the curves reached an asymptote at a relatively low concentration. Although this value increased
with the increasing percentiles, the asymptotic value was rather low when compared to the curves in the
micronutrient plots (Fig. 11 b, e, h). For example, the predicted curves for the macronutrient (Fig. 11 green line)
relative to the observations (Fig. 11 gray contours) showed that higher biomass values were possible even when
micronutrient and irradiance were at their 75[th] percentile values and increases in the macronutrient did not yield
higher biomass (Fig. 11 a, d, g). Since the light curves (Fig. 11 c, f, i) showed the same trend as the macronutrient,
this suggests that the micronutrient was limiting in those circumstances. This is supported by the micronutrient
curves in which the asymptotic values occurred at relatively higher concentrations of the micronutrient (Fig. 11 b, e,
h). The predicted biomass for the micronutrient curves exceeded the highest observation even in the 50[th] percentile
plot (Fig. 11 e). Furthermore, the interaction plots supported this where only interactions with increasing
micronutrient saw increases in biomass (Fig. 12 d and f), while the macronutrient/irradiance plot (in which
micronutrient was held fixed) quickly plateaued (Fig. 12 e). Conceptually this makes sense since the micronutrient
limitation in the BLING model hinders growth, but also limits the efficiency of light-harvesting (Galbraith et al.,
2010). This result of micronutrient limitation was consistent with the other Scenarios and was not unexpected. The
equations governing Scenarios 1 and 2 (Eq. 1-3) were similar to the equation governing BLING (Eq. 7). So,
micronutrient limitation being present across all three Scenarios was consistent with what would be expected.

The interaction plots for Scenario 3 (Fig. 12 d-f) all appear to show a co-limitation pattern consistent with
Independent Multiplicative Nutrients (Saito et al., 2008 Fig. 2B). This agrees with the patterns of the previous
Scenarios, except for the micro/macronutrient interaction. In Scenarios 1 and 2, the micro/macronutrient interaction
showed a pattern matching Liebig minimizing, while Scenario 3 suggested Independent Multiplicative Nutrients.
This result would not have been expected from simply looking at the structure of the equations but arises in part

from the coupling between the nutrient and light limitations.

Since the objective of Scenario 3 was to apply what we learned in Scenarios 1 and 2 to output from an actual biogeochemical model, we believe we have demonstrated the capabilities of the information one can extract. Although the quantitative method of estimating the half-saturation coefficients proved unreliable, the qualitative information was informative. This includes information on limitations and interactions between variables, along with the ability to understand the level of variability explained by a given set of predictors.

**4 Conclusions**

Although researchers have been able to find apparent relationships for phytoplankton in environmental datasets, it remained unclear why and when the environmental apparent relationships were no longer equal to the intrinsic relationships that control phytoplankton growth. Our main objective in this manuscript was to understand when and why the link between intrinsic and apparent relationships would break by answering two questions:

1.  Can ML techniques find the correct underlying intrinsic relationships and, if so, what methods are most skillful in finding them?
   2.  How do you interpret the apparent relationships that emerge when they diverge from the intrinsic relationships we expect?

In addressing the first question, we observed that NNEs were far superior to RFs and MLR at extracting the intrinsic relationships using information on the apparent relationships when the intrinsic and apparent relationships were closely related. RFs were unable to match the relationships because of their inherent inability to extrapolate outside the range of their training data. Additionally, even though NNEs matched the true relationships well, we were unable to quantify half-saturation coefficient estimates from the sensitivity analysis curves because of co-limitations

between the predictors. However, we were able to show that one can use interaction plots to qualitatively visualize the type of co-limitations occurring between two predictors and identify the variables causing limitations.

Regarding the second question, we demonstrated that time-averaging can lead to a loss of variability in the dataset which, in turn, can greatly affect the predicted relationships one can extract. For our particular system, we found

averaging over large timespans caused underestimation of the predicted relationships (as shown in Appendix A, this will generally be the case for relationships which are concave downward – the opposite will be true for relationships that are concave upward). However, we showed that it was possible to visualize how the relationships were changing from intrinsic to apparent relationships by training NNEs on different averaged timescales of the data.

Furthermore, we showed that the general trends, variable interactions, and nutrient limitations occurring when the intrinsic and apparent relationships were closely linked (as in Scenario 1) could propagate through to situations when the intrinsic and apparent relationships operated over different timescales (Scenario 2).

As a proof-of-concept, we also showed that it was possible to extract information from the output of a biogeochemical model (Scenario 3) using the information and techniques we employed in Scenarios 1 and 2.

This study suffers from two major limitations: the number of ML algorithms we investigated and the number of predictor variables included for each scenario. We limited the number of ML algorithms and predictors for simplicity and easier visualization of the sensitivity analyses. In the real world, phytoplankton may be limited by more physical and biological processes, making the visualization of the sensitivity analyses impractical due to the sheer number of possible interactions that would have to be considered. In cases such as those, it would be beneficial to perform some form of importance analysis or dimensionality reduction to remove insignificant predictor variables, after which sensitivity analyses could be done on the remaining predictors.

The results of this study have several potential applications for oceanographers, including marine ecologists and Earth System modelers. For example, using output from biogeochemical models or observations from environmental datasets, researchers may now be able to:

1. Identify important interactions and colimitations occurring between variables.
2. Discern the type of colimitation occurring between nutrients.
3. Find nutrient limitations without having to perform (or at least being able to conduct fewer) nutrient growth experiments in a lab.
4. Identify apparent relationships between biogeochemical variables, instead of using only spatiotemporal distributions.
5. Understand how variable relationships change over different spatial and temporal scales.

Some potential future applications relevant to the results we show here include:

1. Using these techniques to find and compare the apparent relationships of different ESMs. This would allow the researcher to more specifically identify why different ESMs produce different results.
2. Apply these methods to compare the apparent relationships in observational data and ESM output. This would allow for finer tuning of ESM parameters and relationships, instead of only matching ESM spatial distributions to those of observational distributions.

Preliminary work on both applications shows them to have promising results. We will report on these in future manuscripts.

## Appendix A

Illustration of why time variation causes underestimation of the dependence of biomass on a limiter

$$B = S_* * \left(1 - \exp\left(-\frac{Irr}{K_{Irr}}\right)\right) = S_* * \left(1 - \exp\left(-\frac{\overline{Irr} + Irr'}{K_{Irr}}\right)\right) \tag{A1}$$

where the overbar refers to a time-average and the prime to a variation from this time average. Insofar as the variations are small.

$$B \approx S_* \left(\frac{\overline{Irr} + Irr'}{K_{Irr}} - \frac{1}{2}\left(\frac{\overline{Irr} + Irr'}{K_{Irr}}\right)^2\right) = S_* \frac{\overline{Irr} + Irr'}{Irr_k} * \left(1 - \frac{1}{2} * \frac{\overline{Irr} + Irr'}{K_{Irr}}\right) \tag{A2}$$

Averaging yields

$$\bar{B} \approx S_* \left(\left\{\frac{\overline{Irr}}{K_{Irr}} * \left(1 - \frac{\overline{Irr}}{2K_{Irr}}\right)\right\} - \frac{\overline{Irr'^2}}{2K_{Irr}}\right) < S_* \left(1 - \exp\left(-\frac{\overline{Irr}}{K_{Irr}}\right)\right) \tag{A3}$$

so that if we are trying to fit a curve of the form

$$\bar{B} \approx S_*^{ave} \left\{1 - \exp\left(-\frac{\overline{Irr}}{K_{Irr}}\right)\right\} \tag{A4}$$

We would expect that $S_*^{ave} < S_*$.


**Appendix B**

This appendix provides additional details of the training and construction of the RFs and NNEs that may not have been included in the main text of the manuscript.

**Appendix B1: Random Forests**

The RFs were implemented in Matlab 2019b using the TreeBagger function. Each RF used three predictors: macronutrient, micronutrient, and irradiance. The target variable was phytoplankton biomass. At each split, one random predictor variable was chosen from which two maximally distinct groups were determined. The splits continued until each terminal node contained a minimum of 5 observations. For reproducible results, the random
number generator was set to "twister" with an integer of "123". A total of 500 decision trees were constructed for each RF. This number was chosen because we wanted a sufficient number of trees to minimize the error and still be able to run the training in a relatively short span of time on a standard computer/laptop. The Out-of-Bag (OOB) error for each trained RF can be seen in Fig. B1. Past about 100 trees, the OOB error reaches an asymptote, such that more trees do not decrease the error. We chose to keep the number of trees at 500 because this helped to ensure
generalization in the RF. Additionally, it did not significantly increase the training time and it allowed for the RF structure to be the same across all the Scenarios.

Each variable was scaled between -1 and 1 corresponding to each variable's respective minimum and maximum, respectively (Eq. 9). These scalings were applied for use specifically in the NNEs, but for consistency they were also
applied to the MLR and RF. The values of the variables and predictions of each method were unscaled for analysis (Eq. 10).

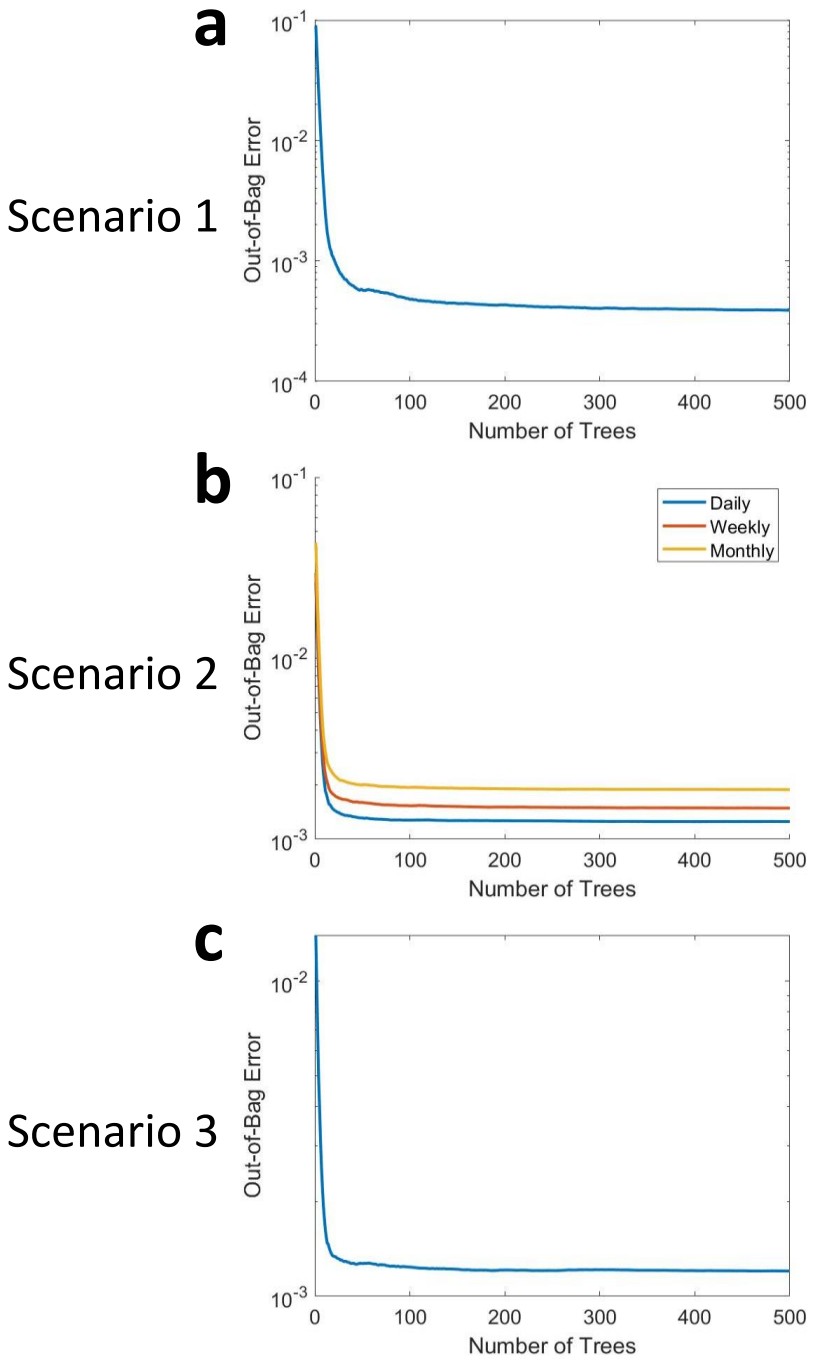

Figure B1: The Out-of-Bag (OOB) error for the trained RFs of each Scenario. The OOB error is shown as a function
of the number of trees for each RF (500 decision trees for each one). The y-axis for each plot is on a log scale.
Additionally, the plot for Scenario 2 shows the OOB error curves for each of the time-averaged datasets (daily,
weekly, monthly).


**Appendix B2: Neural Network Ensembles**

The NNEs consisted of ten individual NNs and each NN was trained using the feedforwardnet function in Matlab 2019b.

The framework of each NN had three input nodes, 25 nodes in a single hidden layer, and one output node. The activation function for the hidden nodes was a hyperbolic tangent sigmoid function and the output node activation

function was a simple linear function. The training dataset was used in the training of each NN, which consisted of 60% of the total observations in the entire dataset. For the training of each individual NN, Matlab further randomly partitioned the training dataset into its own training subset, validation subset, and testing subset. A total of 70% of the observations from the training dataset went to the training subset, 15% went to the validation subset, and 15% went to the testing subset. To ensure that each NN was trained on different observations, distinct combinations of

observations went into each subset for the training of each NN. This was done using a different number for the random number seed before the start of training for each NN. The random number seed ahead of each NN was set to the respective number of the NN. For example, the random number seed for the first NN was set to 1, the seed for the second NN was set to 2, etc. This random number seed ensured that the observations from the training dataset were being partitioned into different training, validation, and testing subsets for each individual NN. The stopping

criteria for each NN was a validation check, so training stopped when the error increased for six consecutive epochs.

The sensitivity analysis used to determine the optimal number of nodes in a single layer NNE for the daily, weekly, and monthly averaged datasets for Scenario 2 can be seen in Table B1. Separate NNEs were trained for each of the time-averaged datasets (daily, weekly, monthly) for each set of nodes. For example, separate NNEs were trained for

the daily-averaged dataset with 1 node, the weekly-averaged dataset with 1 node, and the monthly-averaged dataset with one node. Each NNE maintained the same construction as those specified in the manuscript (10 individual NNs) and kept the same training and stopping specifications outlined in the manuscript. The trained NNEs made predictions on the testing dataset and the $R^2$ values were calculated based on the comparison between those predictions and the actual values of the testing dataset. These values are recorded in Table B1. From the

performance metrics, it was decided that 25 nodes provided a sufficient level of performance while also maintaining a reasonable time for training.

The sensitivity analysis determining if an additional hidden layer increased the performance of the time-averaged datasets in Scenario 2 can be seen in Table B2. Each NNE consisted of ten individual NNs. The NNs were trained

according to the same criteria specified in the manuscript. The inclusion of an additional hidden layer did not significantly increase the performance of the NNEs, but it did significantly increase the time needed for training the

NNs. We decided to use only one hidden layer since the performance did not increase significantly and to keep the training time within a reasonable timeframe.

The sensitivity analysis assessing different activation functions in the nodes of the hidden layer for the time-averaged datasets of Scenario 2 can be seen in Table B3. Each NNE contained ten individual NNs. The NNs kept the same training criteria specified in the manuscript. We tested a total of seven activation functions: hyperbolic tangent (symmetric) sigmoid, logarithmic sigmoid, inverse, positive linear (ReLU), linear, soft max, and radial basis. The linear and inverse activation functions showed the poorest performance. The performance metrics were

comparable for the other activation functions. We decided to use the hyperbolic tangent (symmetric) sigmoid activation function for the nodes in the hidden layer.

Table B1: The $R^2$ values for the diagnostic test used to determine how the number of nodes in the hidden layer of a single layer neural network affected the performance of the time-averaged datasets of Scenario 2. The target variable was biomass (mol kg$^{-1}$). A separate NNE was trained for each of the time-averaged datasets (daily, weekly, monthly) for each set of nodes (ex. A unique NNE for the daily-averaged dataset with 1 node was trained, a unique NNE for the weekly averaged dataset with 1 node was trained, etc.). Each NNE contained 10 individual NNs and kept the same training and stopping specifications outlined in the manuscript. The trained NNEs made predictions on the testing dataset and the $R^2$ values were calculated based on the comparison between those predictions and the actual values of the testing dataset.

|  |  | $R^2$ Values | | |
| --- | --- | --- | --- | --- |
|  |  | Daily | Weekly | Monthly |
| | 1 | 0.5533 | 0.5472 | 0.5624 |
| | 2 | 0.7655 | 0.7705 | 0.7806 |
| | 5 | 0.9283 | 0.9248 | 0.9363 |
| | 10 | 0.9633 | 0.9628 | 0.9673 |
| Number of | 15 | 0.9676 | 0.9678 | 0.9713 |
| Nodes | 20 | 0.9693 | 0.9694 | 0.9727 |
| | 25 | 0.9700 | 0.9702 | 0.9732 |
| | 35 | 0.9709 | 0.9709 | 0.9737 |
| | 50 | 0.9716 | 0.9715 | 0.9743 |

Table B2: The $R^2$ values for the diagnostic test used to determine how the number of hidden layers and nodes within
individual neural networks affected the performance of the Scenario 2 time-averaged datasets. The target variable
was biomass (mol kg$^{-1}$). A separate NNE was trained for each of the time-averaged datasets (daily, weekly,
monthly) for each set of nodes (ex. A unique NNE for the daily-averaged dataset with 25 nodes was trained, a
unique NNE for the weekly averaged dataset with 25 nodes was trained, etc.). Each NNE contained 10 individual
neural networks and kept the same training and stopping specifications outlined in the manuscript. The trained
NNEs made predictions on the testing dataset and the $R^2$ values were calculated based on the comparison between
those predictions and the actual values of the testing dataset. The layers and number of nodes in the table are
specified as follows: # nodes in first layer - # nodes in second layer. If only one number is listed, this specifies the
number of nodes in the single hidden layer and that a second layer was not used.

|  |  | $R^2$ Values | | |
| --- | --- | --- | --- | --- |
|  |  | Daily | Weekly | Monthly |
| Layers and | 25 | 0.9700 | 0.9702 | 0.9732 |
| Number of | 25-10 | 0.9722 | 0.9724 | 0.9750 |
| Nodes | 25-25 | 0.9726 | 0.9727 | 0.9756 |

Table B3: The $R^2$ values for the diagnostic test used to assess how different activation functions in the hidden layer affected the performance of the Scenario 2 time-averaged datasets. The target variable was biomass (mol kg$^{-1}$). A separate NNE was trained for each of the time-averaged datasets (daily, weekly, monthly) for each activation function (ex. A unique NNE for the daily-averaged dataset with the logarithmic sigmoid activation function was trained, a unique NNE for the weekly averaged dataset with the logarithmic sigmoid activation function was trained, etc.). Each NNE contained 10 individual neural networks and kept the same training and stopping specifications outlined in the manuscript. The trained NNEs made predictions on the testing dataset and the $R^2$ values were calculated based on the comparison between those predictions and the actual values of the testing dataset.

|  |  | $R^2$ Values | | |
|---|---|---|---|---|
|  |  | Daily | Weekly | Monthly |
|  | Hyperbolic Tangent (Symmetric) Sigmoid | 0.9681 | 0.9688 | 0.9722 |
|  | Logarithmic Sigmoid | 0.9679 | 0.9691 | 0.9722 |
|  | Inverse | $1.01 \times 10^{-5}$ (0.7236)* | 0.7921 | 0.2455 |
| Activation Functions | Positive Linear (ReLU) | 0.9652 | 0.9671 | 0.9704 |
|  | Linear | 0.3104 | 0.3059 | 0.3125 |
|  | Soft Max | 0.9643 | 0.9649 | 0.9695 |
|  | Radial Basis | 0.9671 | 0.9688 | 0.9716 |

*The low $R^2$ value of the daily-averaged dataset for the Inverse activation function ($1.01 \times 10^{-5}$) was because the first neural network of that NNE stopped training after only 1 epoch due to the momentum parameter ("mu" in Matlab) reaching its maximum value. This significantly decreased the $R^2$ performance of that particular NNE. Removing the first neural network from that NNE increased the $R^2$ value to 0.7236.

**Appendix C**

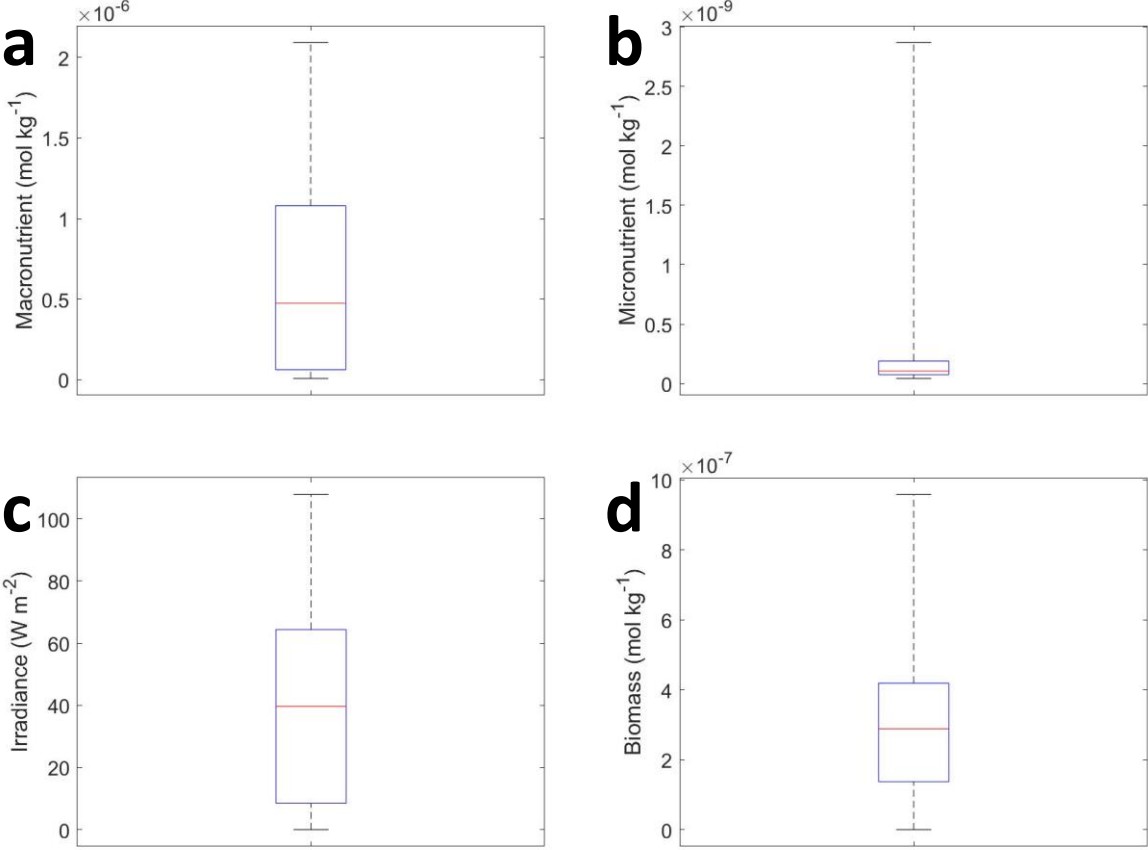

Figure C1: Boxplots showing the variability in the predictor and target variables of Scenario 1. The dataset consisted of monthly averaged variables. The predictor variables include (a) macronutrient, (b) micronutrient, and (c) irradiance. The target variable was phytoplankton (d) biomass. The red line corresponds to the median (50th percentile), the box edges are the 25th and 75th percentile values, and the whiskers are the minimum and maximum values.

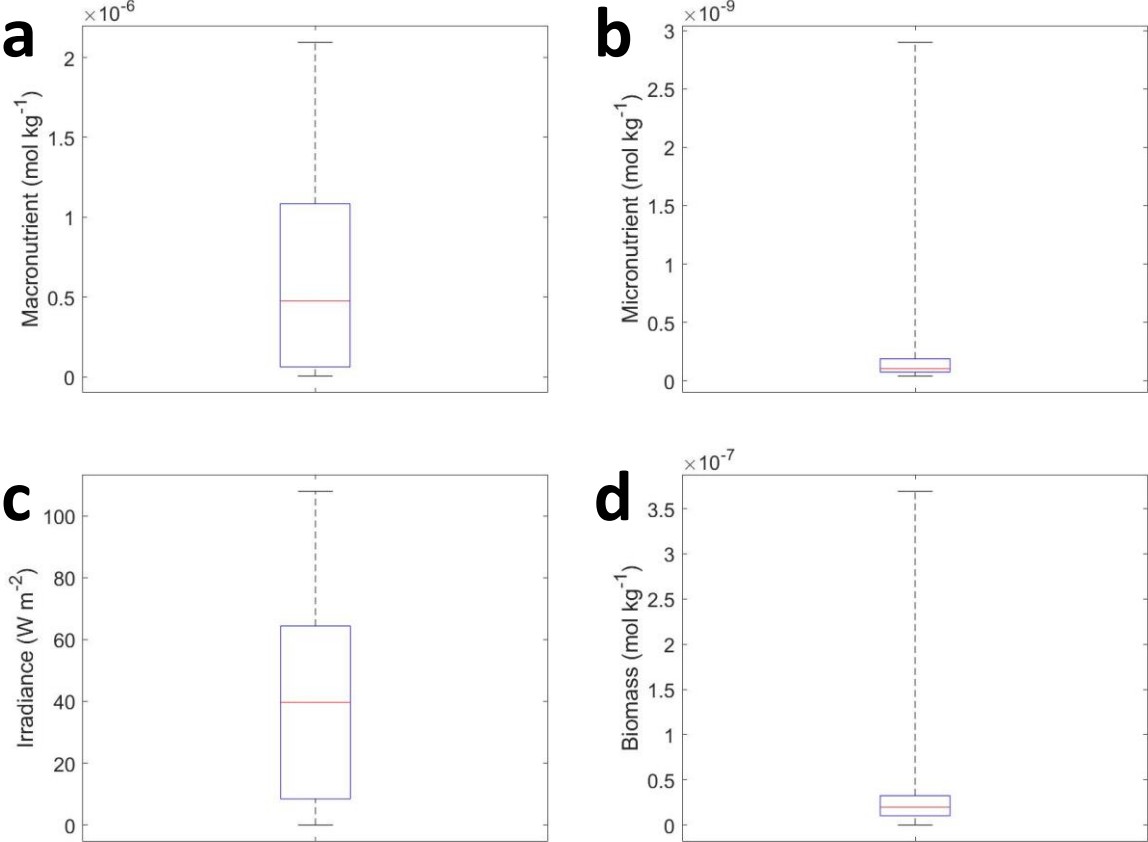

Figure C2: Boxplots showing the variability in the predictor and target variables of Scenario 3. The dataset consisted of monthly averaged variables. The predictor variables include (a) macronutrient, (b) micronutrient, and (c) irradiance. The target variable was phytoplankton (d) biomass. The red line corresponds to the median (50th percentile), the box edges are the 25th and 75th percentile values, and the whiskers are the minimum and maximum values.

**Appendix D**

This appendix provides details about the method used to visualize how the apparent relationships in Scenario 2 were changing from the hourly timescale through to the monthly averaged timescale.

To capture the apparent relationships ranging from the hourly to monthly averaged timescales, we averaged the

hourly dataset over a range of timespans. Specifically, we averaged over the timespans of 1-hour (original hourly dataset), 2, 3, 4, 6, 8, 12, 24, 48, 72, 168 (weekly), and 720 (monthly) hours. The timescales had to be multiples of, or divisible by, 24 hours. Hours that did not meet these criteria would mean that hours from one day would be averaged with hours from another day. For example, using a 7-hour timespan for averaging would have meant that the last three hours of Day 1 were being averaged with the first four hours of Day 2.


We trained one NNE for each of the averaged timescales. Each NNE contained ten individual NNs. The NNs kept the same training criteria specified in the manuscript.

After training the NNEs, we performed a sensitivity analysis on each of them to visualize the predicted apparent

relationships. The percentile values for variables that were not varying were set at their $50^{th}$ percentile (median) values. We then plotted all the predicted curves on a single surface plot so we could view the relationships of all the timescales at once. Additionally, because the greatest variability was lost in the first 24 hours, we also focused on the apparent relationships for the timespans that were less than or equal to 24 hours.

## Code and Data Availability

The Matlab scripts for the construction of the figures and tables, the scripts for training and testing the MLR, RF, and NNE algorithms, and the source files for each scenario are available in the Zenodo data repository (https://doi.org/10.5281/zenodo.3932387, Holder and Gnanadesikan, 2020).

## Author Contribution

CH implemented the ML algorithms, analyzed the results for each scenario, and wrote the majority of the manuscript. AG helped in developing the simple phytoplankton models for Scenarios 1 and 2, provided the biogeochemical model output used in Scenario 3, and helped in the analysis of the results.

## Competing Interest

The authors declare that they have no conflicts of interest.

## Acknowledgments

The authors would like to thank Eric Baumann and Dr. Nicole DeLuca for their comments and recommendations for this manuscript.

## Financial Support

This research was supported in part by the National Science Foundation (NSF) Integrative Graduate Education and Research Traineeship (IGERT) (Grant No. 1069213) and by the NSF Division of Ocean Sciences (OCE) (Grant No. 1756568).

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

1087

**Tables**

Table 1: Details for each Scenario that include the predictor variables, the target variable, the equations used to

calculate biomass, the type of source file used to acquire the values for the predictors, and a short description with

important details about each scenario.

| Scenario | Predictors | Target | Equations Used | Source File Description | Scenario Description |
|---|---|---|---|---|---|
| 1 | Macronutrient (mol kg$^{-1}$); Micronutrient (mol kg$^{-1}$); Irradiance (W m$^{-2}$) | Biomass (mol kg$^{-1}$) | 1, 2, 3 | Monthly Output from BLING | 1) Nutrient distributions (predictors) from BLING were run through Eq. 1, 2, and 3 to calculate the biomass (target) <br> 2) The true relationships were calculated by using the range of the values for the predictors and calculating the biomass based on Eq. 1, 2, and 3 |
| 2 | Macronutrient (mol kg$^{-1}$); Micronutrient (mol kg$^{-1}$); Irradiance (W m$^{-2}$) | Biomass (mol kg$^{-1}$) | 1, 2, 3, 6 | Daily Output from BLING | 1) Hourly values for the predictors were interpolated using the Daily Output of BLING <br> 1a) The macronutrient and micronutrient hourly values were calculated using a standard interpolation between the daily points. <br> 1b) The irradiance hourly values were calculated from Eq. 6 using the value of the BLING daily input, hour of day, time of year, and location. <br> 2) Hourly values of the predictors were fed to Eq. 1, 2, and 3 to calculate hourly values for the biomass (target) <br> 3) Daily-averaged values were calculated by averaging 24 hours for each location through one year <br> 4) Weekly-averaged values were calculated by averaging 168 hour blocks of time for each location through the year <br> 5) Monthly-averaged values were calculated by averaging the number of hours in each month (days per month * 24) for each location through the year <br> 6) The true relationships were calculated by using the range of the hourly values for the predictors and calculating the biomass based on Eq. 1, 2, and 3 |
| 3 | Macronutrient (mol kg$^{-1}$); Micronutrient (mol kg$^{-1}$); Irradiance (W m$^{-2}$) | Biomass (mol kg$^{-1}$) | 7, 8 (Equations within BLING used to determine the biomass) | Monthly Output from BLING | 1) Nutrient distributions from the BLING Output were used as the predictors; Biomass from the BLING Output itself was used as the target |

Table 2: Performance metrics (Coefficient of Determination [$R^2$] and Root Mean Squared Error [RMSE]) for the

training and testing datasets of each Scenario and the respective ML method (MLR – Multiple Linear Regression;

RF – Random Forest; NNE – Neural Network Ensemble). Scenario 2 had three time-averaged datasets (daily,

weekly, and monthly). The target variable for all Scenarios was phytoplankton biomass.

| | | | Training Data | | Testing Data | |
|---|---|---|---|---|---|---|
| | | | R-squared | RMSE | R-squared | RMSE |
| Scenario 1 | | MLR | 0.4528 | $1.32 \times 10^{-7}$ | 0.4471 | $1.33 \times 10^{-7}$ |
| | | RF | 0.9989 | $6.46 \times 10^{-9}$ | 0.9977 | $9.15 \times 10^{-9}$ |
| | | NNE | 0.9999 | $1.70 \times 10^{-9}$ | 0.9999 | $1.73 \times 10^{-9}$ |
| Scenario 2 | Daily | MLR | 0.3160 | $8.75 \times 10^{-8}$ | 0.3104 | $8.82 \times 10^{-8}$ |
| | | RF | 0.9841 | $1.35 \times 10^{-8}$ | 0.9684 | $1.90 \times 10^{-8}$ |
| | | NNE | 0.9686 | $1.88 \times 10^{-8}$ | 0.9681 | $1.90 \times 10^{-8}$ |
| | Weekly | MLR | 0.3054 | $8.35 \times 10^{-8}$ | 0.3059 | $8.31 \times 10^{-8}$ |
| | | RF | 0.9835 | $1.30 \times 10^{-8}$ | 0.9687 | $1.78 \times 10^{-8}$ |
| | | NNE | 0.9680 | $1.79 \times 10^{-8}$ | 0.9688 | $1.76 \times 10^{-8}$ |
| | Monthly | MLR | 0.3022 | $8.07 \times 10^{-8}$ | 0.3125 | $8.01 \times 10^{-8}$ |
| | | RF | 0.9859 | $1.16 \times 10^{-8}$ | 0.9729 | $1.60 \times 10^{-8}$ |
| | | NNE | 0.9722 | $1.61 \times 10^{-8}$ | 0.9722 | $1.61 \times 10^{-8}$ |
| Scenario 3 | | MLR | 0.0672 | $2.55 \times 10^{-8}$ | 0.0691 | $2.53 \times 10^{-8}$ |
| | | RF | 0.9727 | $4.49 \times 10^{-9}$ | 0.9445 | $6.26 \times 10^{-9}$ |
| | | NNE | 0.9417 | $6.38 \times 10^{-9}$ | 0.9386 | $6.50 \times 10^{-9}$ |

Table 3: The true value and estimated half-saturation coefficients for each Scenario and predictor (macronutrient, micronutrient, and light) based on the 25th, 50th, and 75th percentiles. The percentiles correspond to the values at which the other predictors were set (ex. For the 25th Percentile Macronutrient value, the macronutrient varied across its min-max range while micronutrient and light were set at their respective 25th percentile values). The coefficients were estimated using a non-linear regression function to fit a curve to the predictions in the sensitivity analyses of the form in Eq. 4, where $\alpha_2$ was the estimate for each half-saturation coefficient.

| | | | NNE | | |
| --- | --- | --- | --- | --- | --- |
| | | | Macronutrient | Micronutrient | Light |
| True Value | | | $1.00 \times 10^{-7}$ | $2.00 \times 10^{-10}$ | 34.30 |
| Scenario 1 | | 25th Percentile | $6.27 \times 10^{-9}$ | $1.29 \times 10^{-9}$ | 38.91 |
| | | 50th Percentile | $1.04 \times 10^{-8}$ | $1.44 \times 10^{-10}$ | 38.26 |
| | | 75th Percentile | $1.88 \times 10^{-8}$ | $2.86 \times 10^{-10}$ | 40.09 |
| Scenario 2 | Daily | 25th Percentile | $9.87 \times 10^{-9}$ | $-9.85 \times 10^{-11}$ | 22.04 |
| | | 50th Percentile | $3.22 \times 10^{-8}$ | $1.88 \times 10^{-10}$ | 23.20 |
| | | 75th Percentile | $4.89 \times 10^{-8}$ | $3.51 \times 10^{-10}$ | 20.09 |
| | Weekly | 25th Percentile | $1.08 \times 10^{-8}$ | $-6.48 \times 10^{-10}$ | 26.18 |
| | | 50th Percentile | $3.78 \times 10^{-8}$ | $1.92 \times 10^{-10}$ | 25.50 |
| | | 75th Percentile | $6.36 \times 10^{-8}$ | $1.11 \times 10^{-9}$ | 18.49 |
| | Monthly | 25th Percentile | $7.64 \times 10^{-9}$ | $-6.90 \times 10^{-10}$ | 23.13 |
| | | 50th Percentile | $3.26 \times 10^{-8}$ | $1.63 \times 10^{-10}$ | 19.37 |
| | | 75th Percentile | $1.38 \times 10^{-7}$ | $1.04 \times 10^{-9}$ | 21.89 |
| Scenario 3 | | 25th Percentile | $3.50 \times 10^{-8}$ | $6.84 \times 10^{2}$ | 1.85 |
| | | 50th Percentile | $8.89 \times 10^{-8}$ | $6.94 \times 10^{-10}$ | 5.80 |
| | | 75th Percentile | $1.64 \times 10^{-7}$ | $2.41 \times 10^{-9}$ | 7.78 |

Table 4: The true value and estimated half-saturation coefficients for each Scenario and predictor (macronutrient,
micronutrient, and light) based on the 97th, 98th, and 99th percentiles. The percentiles correspond to the values at
which the other predictors were set (ex. For the 97th Percentile Macronutrient value, the macronutrient varied across
its min-max range while micronutrient and light were set at their respective 97th percentile values). The coefficients
were estimated using a non-linear regression function to fit a curve to the predictions in the sensitivity analyses of
the form in Eq. 4, where $\alpha_2$ was the estimate for each half-saturation coefficient.

| | | | NNE | | |
| --- | --- | --- | --- | --- | --- |
| | | | Macronutrient | Micronutrient | Light |
| True Value | | | $1.00 \times 10^{-7}$ | $2.00 \times 10^{-10}$ | 34.30 |
| Scenario 1 | | 97th Percentile | $4.33 \times 10^{-8}$ | $4.73 \times 10^{-10}$ | 39.48 |
| | | 98th Percentile | $4.85 \times 10^{-8}$ | $4.68 \times 10^{-10}$ | 42.11 |
| | | 99th Percentile | $6.06 \times 10^{-8}$ | $4.49 \times 10^{-10}$ | 49.43 |
| Scenario 2 | Daily | 97th Percentile | $2.28 \times 10^{-7}$ | $4.10 \times 10^{-10}$ | 217.3 |
| | | 98th Percentile | $2.99 \times 10^{-7}$ | $4.02 \times 10^{-10}$ | 254.0 |
| | | 99th Percentile | $3.93 \times 10^{-7}$ | $3.90 \times 10^{-10}$ | 276.2 |
| | Weekly | 97th Percentile | $2.59 \times 10^{-7}$ | $7.23 \times 10^{-10}$ | 68.86 |
| | | 98th Percentile | $3.39 \times 10^{-7}$ | $6.33 \times 10^{-10}$ | 70.56 |
| | | 99th Percentile | $4.28 \times 10^{-7}$ | $5.19 \times 10^{-10}$ | 70.32 |
| | Monthly | 97th Percentile | $3.56 \times 10^{-7}$ | $9.04 \times 10^{-10}$ | 85.22 |
| | | 98th Percentile | $3.96 \times 10^{-7}$ | $9.16 \times 10^{-10}$ | 82.73 |
| | | 99th Percentile | $5.17 \times 10^{-7}$ | $9.55 \times 10^{-10}$ | 82.61 |
| Scenario 3 | | 97th Percentile | $5.19 \times 10^{-7}$ | $2.00 \times 10^{-9}$ | 54.00 |
| | | 98th Percentile | $7.02 \times 10^{-7}$ | $1.89 \times 10^{-9}$ | 76.48 |
| | | 99th Percentile | $1.01 \times 10^{-6}$ | $1.74 \times 10^{-9}$ | 86.21 |

**Figures**

Figure 1: The contour plots in the top row show the yearly-averaged biomass of Scenario 1 for the true response (a)

and the associated predictions from MLR (b), RF (c), and NNE (d). The biomass was measured in units of mol kg$^{-1}$.

The contour plots in the bottom row show the Log$_{10}$ Absolute Error between the true response and the predictions

from MLR (e), RF (f), and NNE (g).

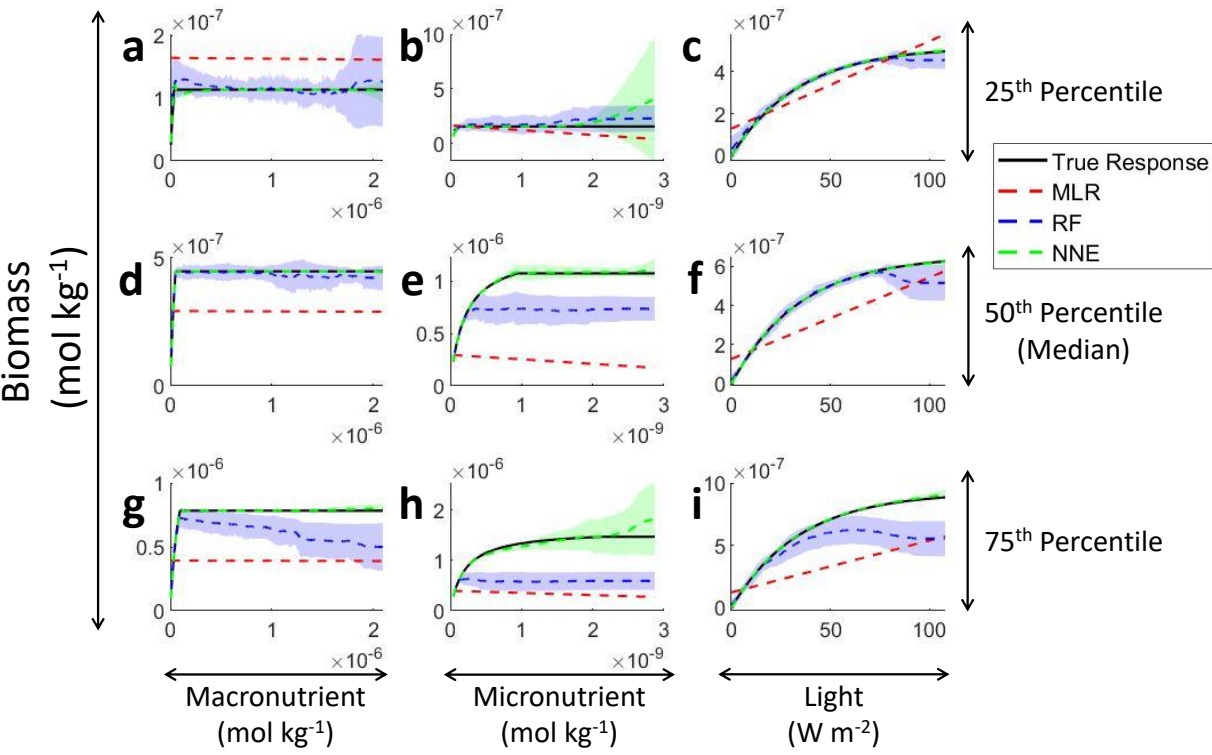

Figure 2: Sensitivity analysis for Scenario 1 showing the true and predicted relationships for each ML method. The columns correspond to the predictors and the rows correspond with the percentile value at which the other predictors were set (ex. Subplot **a** varies the macronutrient across its min-max range, while the micronutrient and light are held at their 25th percentile values, respectively). The black line shows the true intrinsic relationship calculated from Eq. 1-3. The dashed lines show the predicted apparent relationships for each method (MLR – red; RF – blue; NNE – green). The RF and NNE predicted relationships are the average of the individual predictions for each method. The colored regions around the RF and NNE dashed lines show one standard deviation in the predictions (ex. One standard deviation in the 10 individual NN predictions of the NNE).

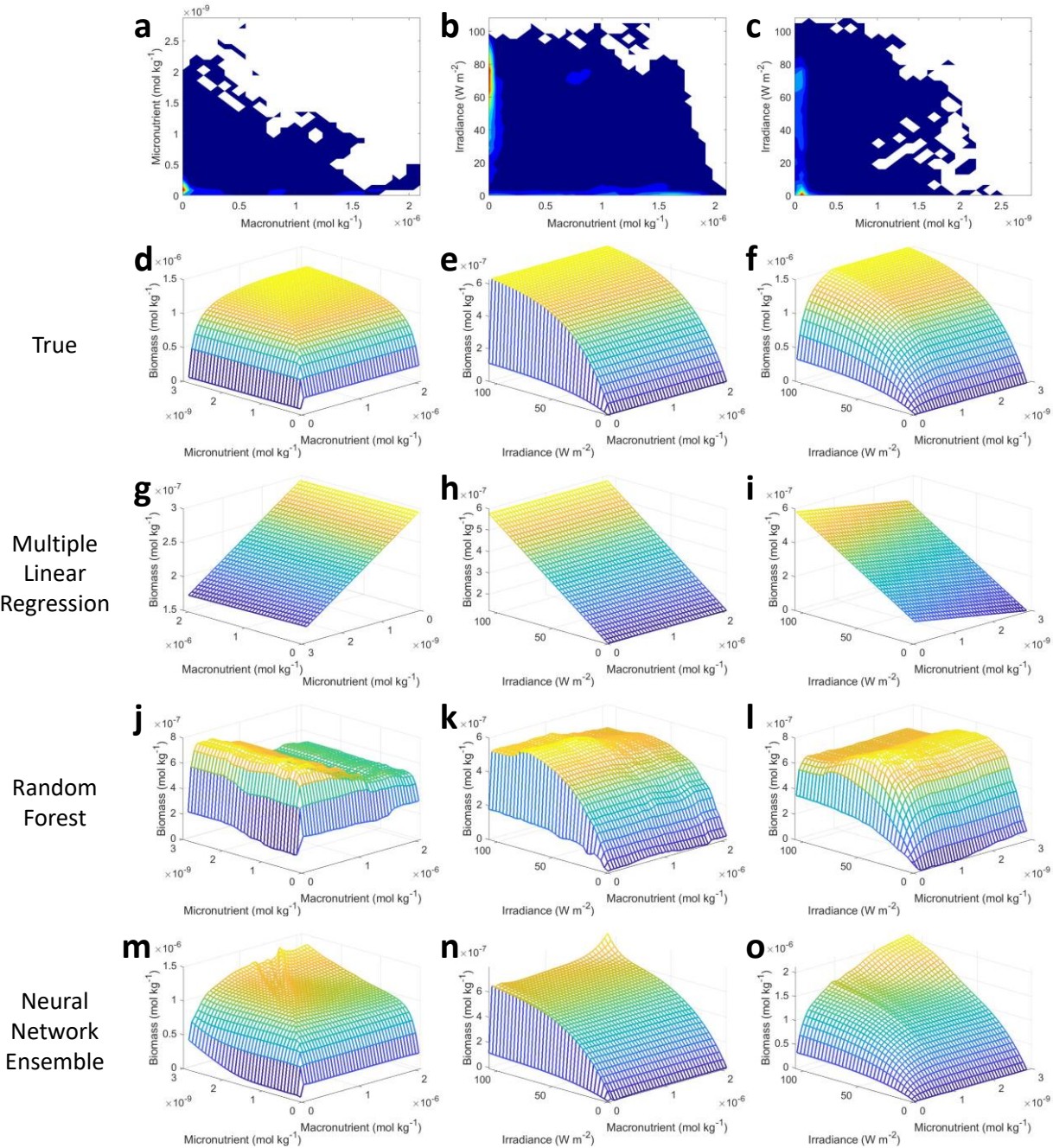

Figure 3: Contour and interaction plots for Scenario 1. The contour plots show the density of observations for each set of predictors (a-c) where blue signifies very few observations and colors moving up the spectrum to red indicate many observations. The interaction plots (d-o) show the biomass values for different combinations of the predictors on each x and y axis. The predictor that was not varying was set at its 50th percentile (median) value (ex. Subplot d allows the micro- and macronutrient to vary across their respective min-max ranges, while the irradiance is held fixed at its 50th percentile value). The top three interaction plots (d-f) show the true interactions calculated from Eq. 1-3. The remaining interaction plots show the predicted interactions for MLR (g-i), RF (j-l), and NNE (m-o). Note

that the x and y axes for subplot g were switched so that the interaction could be visualized. The RF and NNE

predicted relationships are the average of the individual predictions for each method.

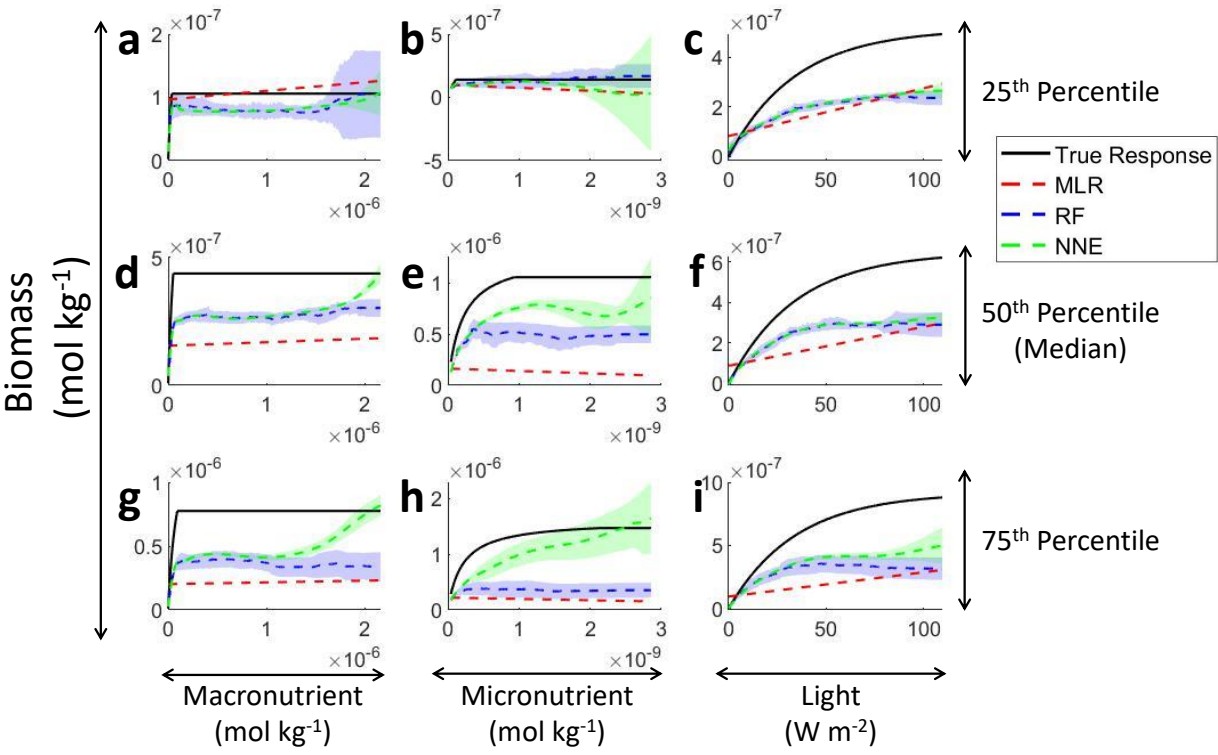

Figure 4: Sensitivity analysis for Scenario 2 showing the true and predicted relationships for each ML method. The columns correspond to the predictors and the rows correspond with the percentile value at which the other predictors were set (ex. Subplot **a** varies the macronutrient across its min-max range, while the micronutrient and light are held at their 25th percentile values, respectively). The black line shows the true intrinsic relationship calculated from Eq. 1-3. The dashed lines show the predicted **monthly** apparent relationships for each method (MLR – red; RF – blue; NNE – green). The RF and NNE predicted relationships are the average of the individual predictions for each method. The colored regions around the RF and NNE dashed lines show one standard deviation in the predictions (ex. One standard deviation in the 10 individual NN predictions of the NNE).

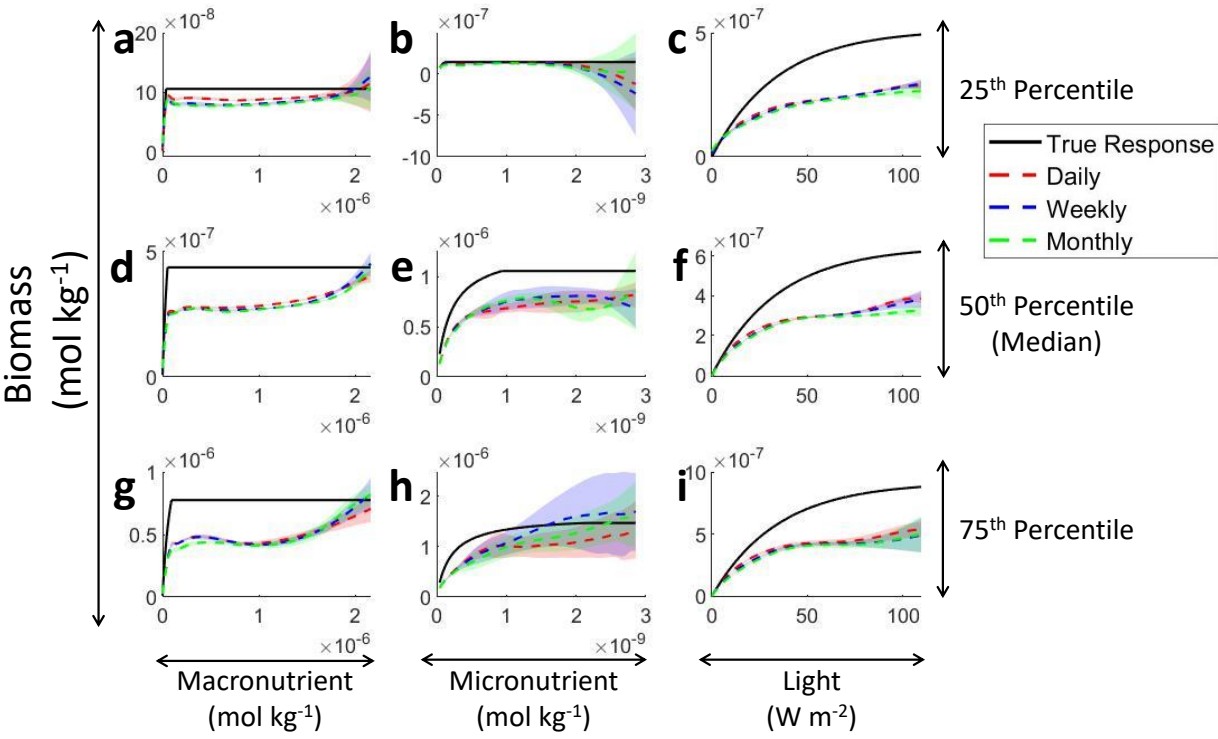

Figure 5: Sensitivity analysis for Scenario 2 showing the true and predicted NNE relationships for the different time-averaged datasets. The columns correspond to the predictors and the rows correspond with the percentile value at which the other predictors were set (ex. Subplot **a** varies the macronutrient across its min-max range, while the micronutrient and light are held at their 25th percentile values, respectively). The black line shows the true intrinsic relationship calculated from Eq. 1-3. The dashed lines show the predicted apparent relationships for each time-averaged dataset (Daily – red; Weekly – blue; Monthly – green). The conditions for the sensitivity analysis were based on the values from the monthly averaged dataset. It was necessary to give the same conditions to all the time-averaged datasets so that a direct comparison could be made between the predictions of the respective NNEs. The predicted relationships are the average of the individual predictions for each time-averaged NNE, respectively. The colored regions around the NNE dashed lines show one standard deviation in the predictions (ex. One standard deviation in the 10 individual NN predictions of each NNE).

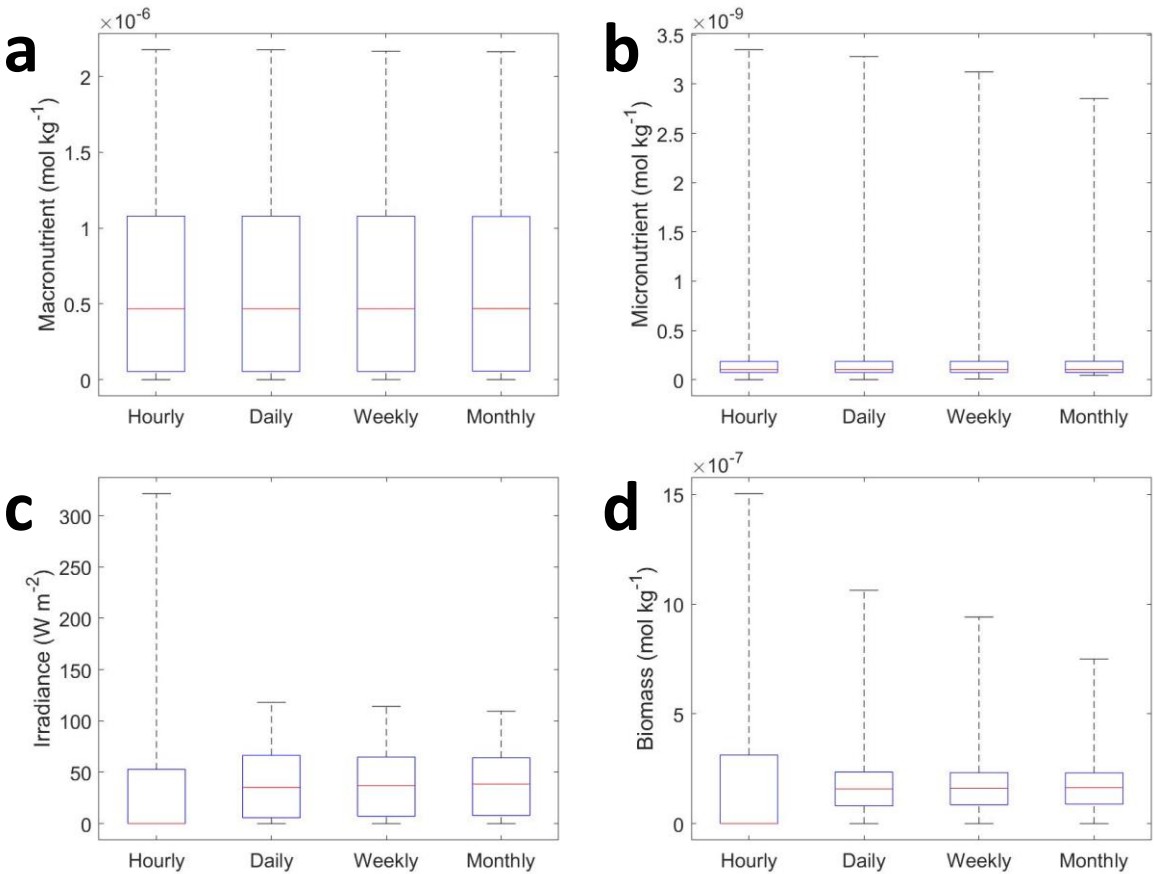

Figure 6: Boxplots showing the variability in the predictor and target variables of Scenario 2 for the various time-

averaged datasets. The predictor variables include (a) macronutrient, (b) micronutrient, and (c) irradiance. The target

variable was phytoplankton (d) biomass. The red line corresponds to the median (50th percentile), the box edges are

the 25th and 75th percentile values, and the whiskers are the minimum and maximum values.

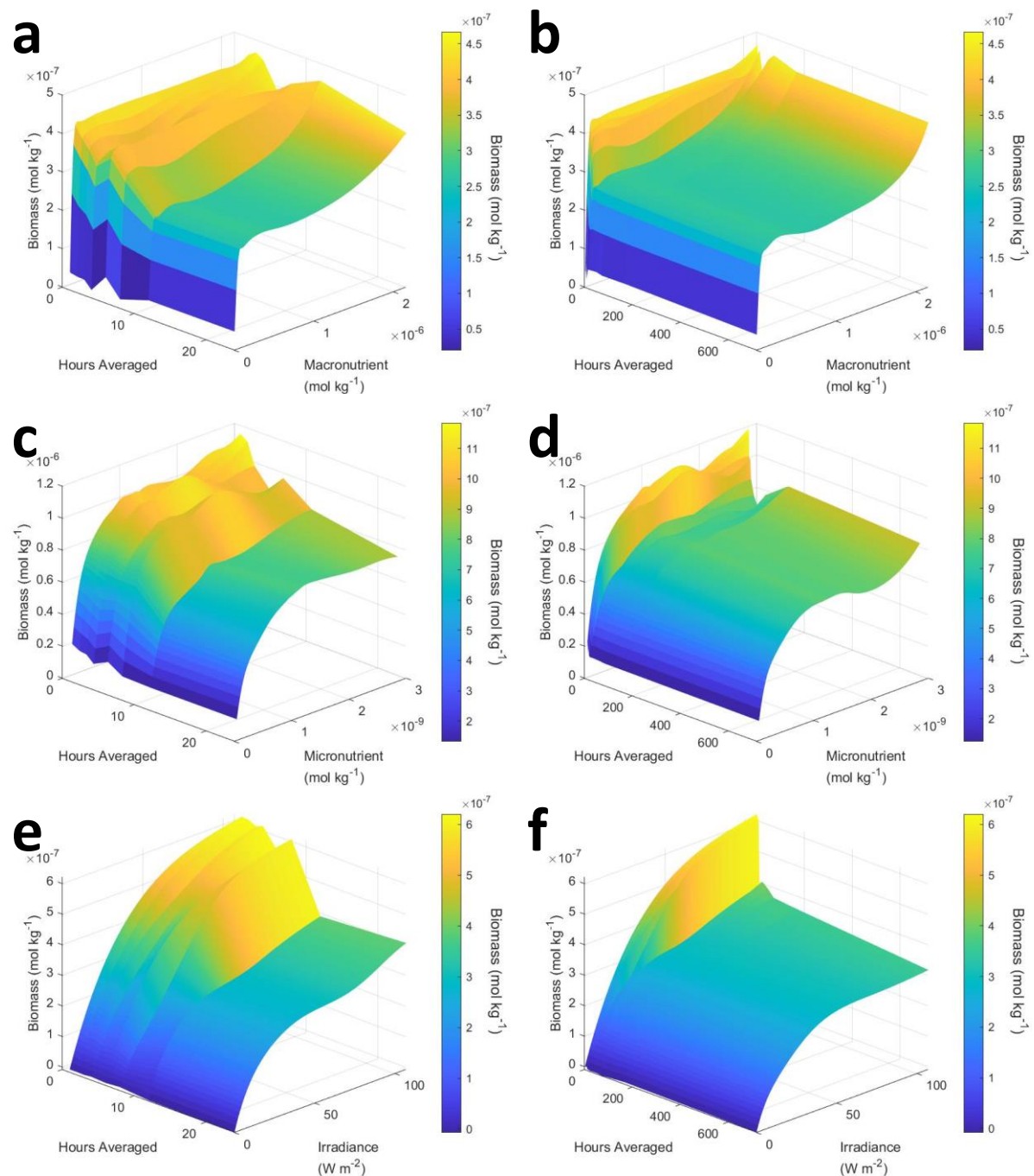

Figure 7: Surface plots showing the apparent relationships found across different averaged timescales for Scenario 2. The timescales range from 1 hour (original hourly set) up to 720 hours (monthly). The three plots on the right (b, d, f) show the relationships across the entire range of timescales (1 through 720 hours). The three plots on the left (a, c, e) show the timescales at and below 24 hours. The top plots show the relationships for the macronutrient (a, b), the middle plots show the relationships for the micronutrient (c, d), and the bottom plots show the relationships for irradiance (e, f). Variables not varying across their range were set at their 50th percentile (median) value. The

conditions of the sensitivity analyses were based on the conditions of the monthly averaged (720-hour) dataset. It

was necessary to give the same conditions to the all the time-averaged datasets so that a direct comparison could be

made between the predictions of the respective NNEs. The predicted relationships are the average of the individual

predictions for each time-averaged NNE.

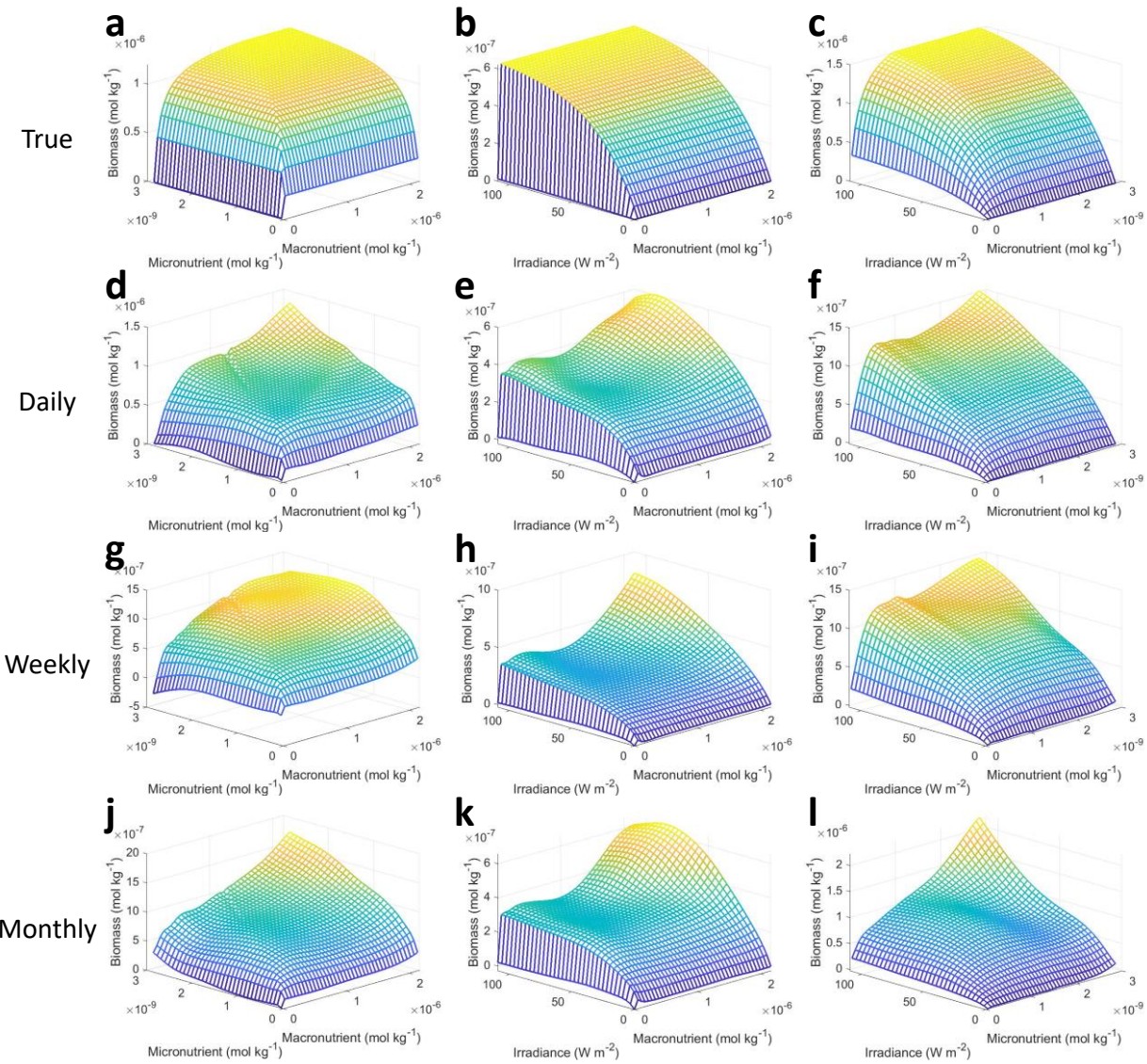

Figure 8: Interaction plots for Scenario 2. The interaction plots show the biomass values for different combinations

of the predictors on each x and y axis. The predictor that was not varying was set at its 50th percentile (median)

value (ex. Subplot d allows the micro- and macronutrient to vary across their respective min-max ranges, while the

irradiance is held fixed at its 50th percentile value). The top three interaction plots (a-c) show the true interactions

calculated from Eq. 1-3. The remaining interaction plots show the predicted interactions for the time-averaged

datasets: daily (d-f), weekly (g-i), and monthly (j-l). The conditions for the sensitivity analysis were based on the

values from the monthly averaged dataset. It was necessary to give the same conditions to all the time-averaged

datasets so that a direct comparison could be made between the predictions of the respective NNEs. The predicted

relationships are the average of the individual predictions for each time-averaged NNE.

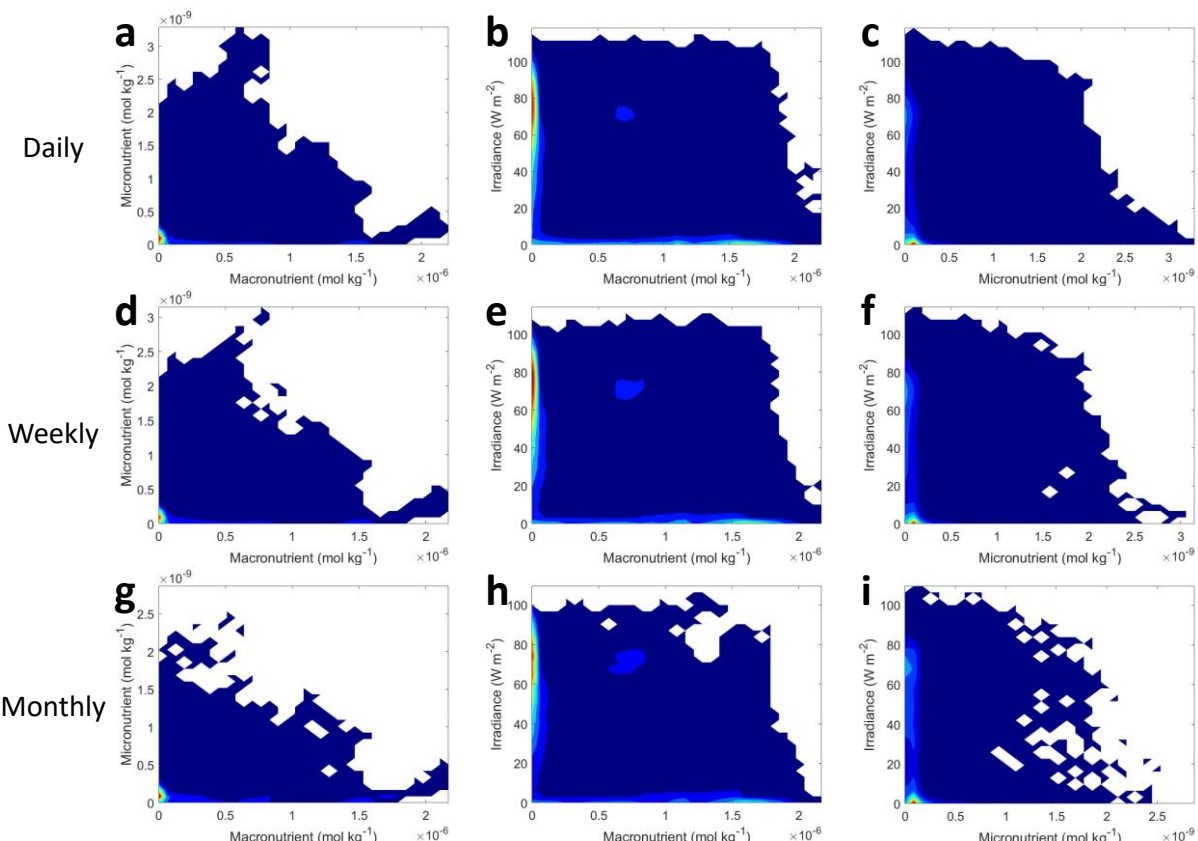

Figure 9: Contour plots of Scenario 2 for the time-averaged datasets: daily (a-c), weekly (d-f), and monthly (g-i).

The contour plots show the density of observations for each set of predictors where blue signifies very few

observations and colors moving up the spectrum to red indicate many observations.

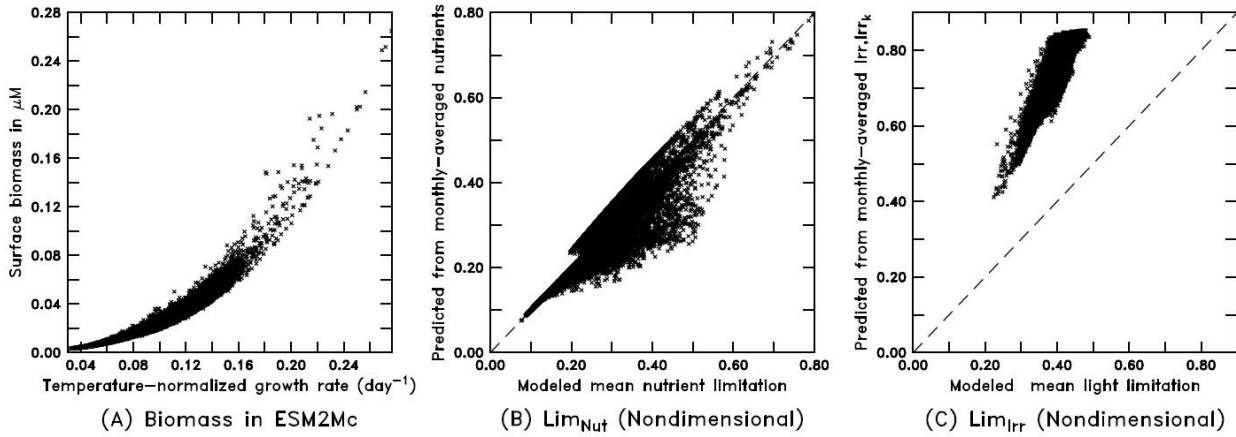

Figure 10: Scatter plots from the BLING model (a: surface biomass vs. temperature-normalized growth rate; b:

monthly-averaged nutrients vs. mean nutrient limitation; c: monthly-averaged Irr, $Irr_k$ vs. mean light limitation).

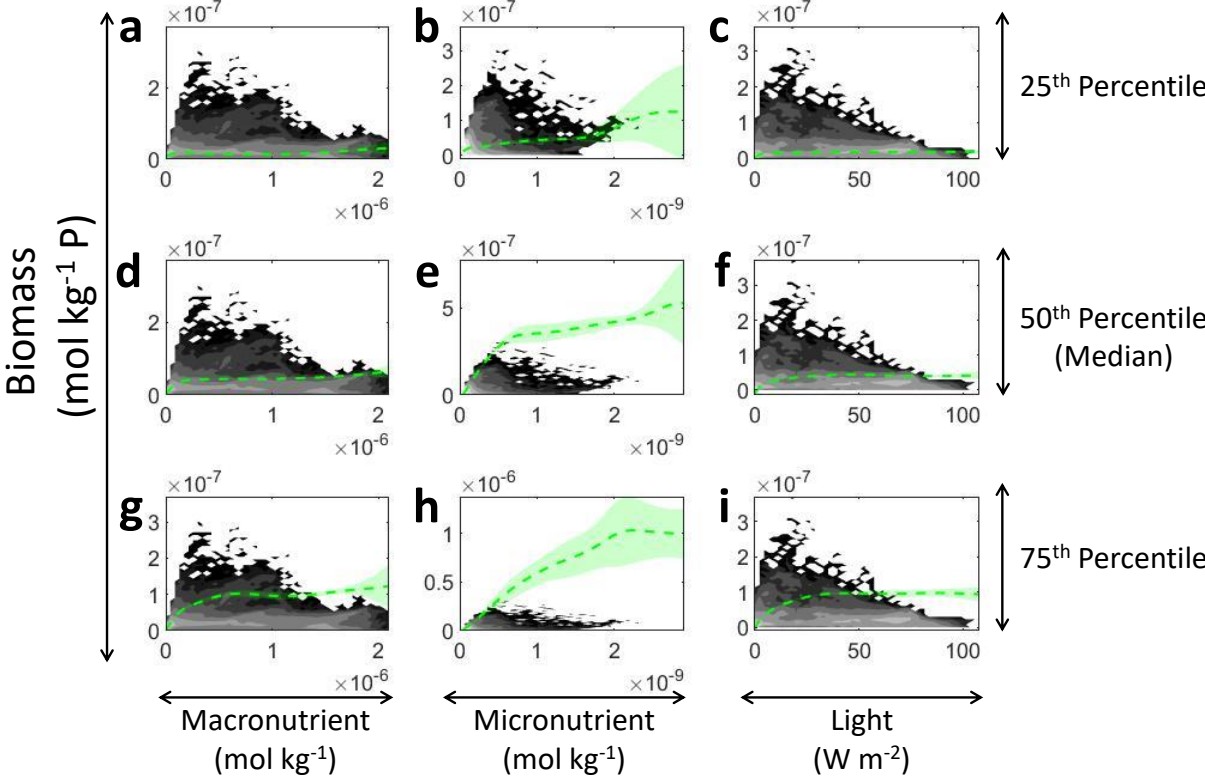

Figure 11: Sensitivity analysis for Scenario 3 showing the predicted relationships for the NNE. The columns

correspond to the predictors and the rows correspond with the percentile value at which the other predictors were set

(ex. Subplot **a** varies the macronutrient across its min-max range, while the micronutrient and light are held at their

25th percentile values, respectively). The green dashed line shows the apparent relationships predicted by the NNE.

The predicted relationships are the average of the individual predictions for each NN. The colored regions around

the NNE dashed lines show one standard deviation in the predictions (ex. One standard deviation in the 10

individual NN predictions of the NNE). The contour plot behind the predicted relationships show the observations

for each predictor against the biomass. Lighter colors signify a higher density of observations, while darker regions

correspond to fewer observations.

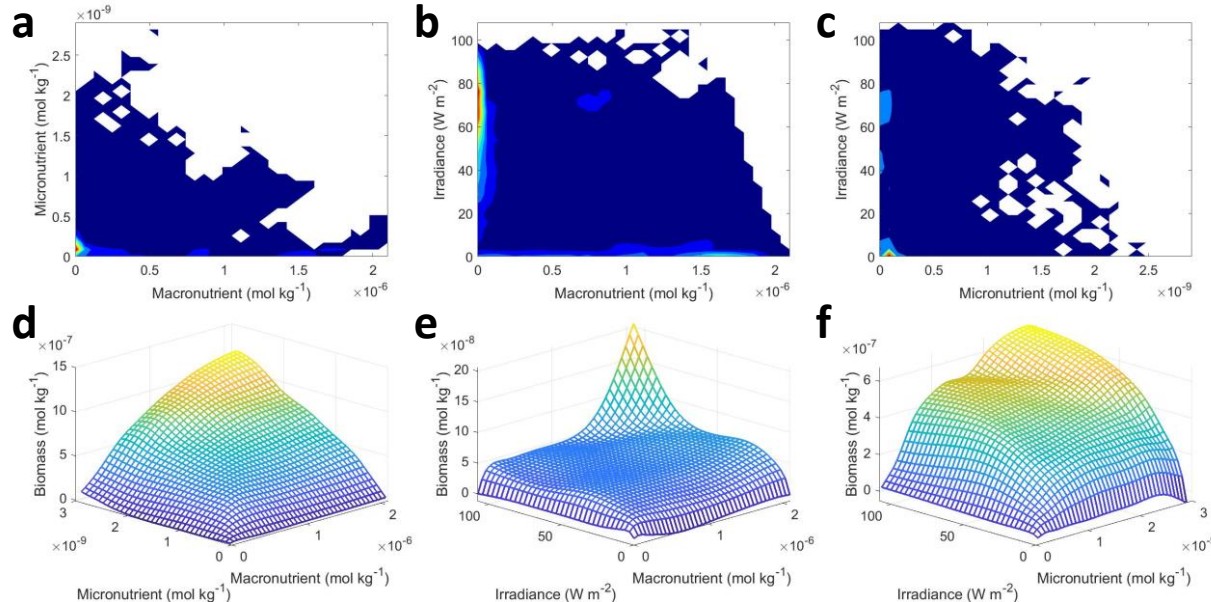

Figure 12: Contour and interaction plots for Scenario 3. The contour plots show the density of observations for each

set of predictors (a-c) where blue signifies very few observations and colors moving up the spectrum to red indicate

many observations. The interaction plots (d-f) show the biomass values for different combinations of the predictors

on each x and y axis. The predictor that was not varying was set at its 50th percentile (median) value (ex. Subplot d

allows the micro- and macronutrient to vary across their respective min-max ranges, while the irradiance is held

fixed at its 50th percentile value). The interaction plots show the predicted interactions based on the NNE. The

predicted relationships are the average of the individual predictions for each NN.