# Peer review of "Can machine learning extract the mechanisms controlling phytoplankton growth from large-scale observations? – A proof of concept study"

_Biogeosciences, 2020_

## Referee Comment (RC1) · Anonymous Referee #1 · 13 Aug 2020

The authors try to find a ML method that can establish and help to explain the link between intrinsic and apparent relationships of phytoplankton and environmental forcing. Three different methods were tested: Multiple Linear Regression (MLR), Random Forests (RFs) and Neural Network Ensembles (NNEs). The tests were provided on three different Scenarios. The authors found that the NNEs reproduce well the observed biomass (biomass estimated from intrinsic relationships) based on the knowledge of only apparent relationships when both relationships operate on the same spatial and temporal timescales. All methods fail when the intrinsic and apparent relation-

ships operated on different timescales. However, the main authors' conclusion is that ML methods still can give an information on shapes of intrinsic relationships. Using the Earth System Models (ESM) in their third Scenario they show that the ML methods can extract useful information from this model that can used for an examination of interaction between input variables.

The article arises an interesting subject. However, it misses a clear explanation how the trained data and data for validation were constructed; the explanation of the role of input data, especially the physical meaning of the choice to fix them at 25th, 50th and 75th percentile for a sensitive analysis. I also think that the authors did not use all possible ML capacities for example, test more hidden layers in NNs, or add few more input variables, environmental ones, maybe a temperature.

Also, the main results that the NNEs can reproduce the general shape of intrinsic relationships does not have a systematic character: it is clearly seen that in some cases the NNEs shapes reproduce a different behavior compare to observation data. If the authors want to keep this conclusion, they will have to provide an additional analysis of conditions under which their conclusion is persistent.

The article in its current state needs a serious major correction.

Specific comments:

It is hard to understand reading the introduction what is the main aim of the article. I have found three:

- "A significant challenge that remains is determining how intrinsic relationships found in the laboratory scale up to the apparent relationships observed at the ecosystem scale (i.e., scaling the small to the large)."

- "What is less clear is: 1. Can robust relationships be found? 2. If so, what methods are most skillful in finding them? 3. How do you interpret the apparent relationships that emerge when they diverge from the intrinsic relationships we expect?"

- "To investigate when and why the link between intrinsic and apparent relationships break..."

If the two first citations can be linked to the title "What are the challenges?", the last one, what the authors actually did, is not according to the title "What are the challenges?". I will advise to authors modify the structure of the introduction and emphasize what is the main idea of this article.

The conclusion also has to be rewritten in clear way: what are the challenges authors have found to link intrinsic and apparent relationships? The authors said: "Our main objective in this manuscript was to use ML to determine under what conditions intrinsic and apparent relationships between phytoplankton are no longer equal..." This objective was not clear in the introduction, and again, does not correspond well to the title of the article.

I advise you also to avoid the non-explained abbreviations in the abstract, like line 21: "ESM", line 28: "MLR".

It would be better not mentioning the results in the introduction (lines 100-103), and instead prepare your readers for the structure of the article.

In the section "2. Methods" I suggest you provide a scheme or formula what exactly you were using as input/predictors and output/target and how it links to your equations 1 and 2 etc. This will simplify the understanding. It is especially important for the description of the Scenario 2. It is unclear, are the values of biomass with which the authors compare their results calculated based on hourly values? Is the biomass for target in learning algorithm smoothed or calculated based on smoothed predictors? It will strongly affect the results.

The word "target" was not used in your article. It is common word in the ML domain and can also help to better understand the method especially for readers who only start to use ML techniques. Also, I would suggest to use the word "validation" instead
of "testing".

To perform a sensitive analysis the authors fixed two of three predictors at different percentiles. It misses the explanation why 25th, 50th and 75th percentiles were chosen, what is the physical meaning of this choice and how it influences the results? For example, does 75th percentiles represent the extremes and what does it mean? I think that it is important to explain it to better understand the results. Please, clarify that the sensitivity analysis was done already on trained ML model.

Did the authors try to increase the number of hidden layers in their NNs? It is known that the introduction of more hidden layers can improve the results. It would be interesting to see if there is any effect from the number of hidden layers in this particular problem.

The authors did not provide how they scaled their variables (lines 337-341). This procedure is known as normalization of variables. Normalization ensures that all predictors fall within a comparable range and avoids giving more weight to predictors with large variability ranges.

It is hard to agree that the NNEs and RFs represent well the behavior as for example on Figure 3 in left column the NNEs show a strong increase at the end of the macronutrient range that does not in agreement with the observation values; and there is a false decrease in the middle column at 25th percentile.

It is interesting to know if the authors have an idea about new parameter or variable that can bring back the information on hourly variability lost due to the time-averaging to improve the results? The article misses the total results for Scenarios 2 and 3 like on Figure 1.

In the Discussion of results for Scenario 3 the authors reasoned about BLING model behavior and did not mention their results. It would be interesting to know the authors' thoughts about NNEs behavior on Figure 7 middle column at 75th percentile.

I have also found that the authors wrote too much on things that they did not do: lines 171-177.

Lines 260 – 273: I suggest to rewrite these two paragraphs, there is a mention of results that have not yet been presented; and I do not see the necessity to mention that the authors "previously had little experience with ML."

Please use the figures captions like "a", "b" etc.

Please, expand the figure captions, for example on Figure 2 it would be good to add that the black line is estimated from Eq. 1.

Figure 5 was mentioned as the last but it is placed before Figure 6 and 7.

Line 149, 156, 168, 356: please avoid to use the sign " in scientific paper.

Line 149: sign " should be before the point .

Line 206: it feels that "but" should be replaced by "and".

Line 226, 235, 543: word "just" is unnecessary.

Line 250: "an ML" should be replaced by "a ML".

Line 356: Miss a figure indication.

---

## Referee Comment (RC2) · Luke Gregor (Referee) · 31 Aug 2020

**1   Overview**

The study by Holder and Gnanadesikan tries to assess if machine learning is able to extract the intrinsic relationship between phytoplankton growth and limiting nutrients and light from observed concentrations of nutrients and light intensity. This topic was investigated with three experiments of increasing complexity asking the following ques-

tions (with my brief understanding of the outcomes):

1. Are ML methods able to extract the relationship from observations at all at instantaneous time scales?
   *Yes, but NNE is better at extracting the relationship than RF despite both achieving fair results*

2. If time scales are averaged, can the relationships still be extracted?
   *Not very well. In most cases the estimated half-saturation is lower than it should be. I.e. even the better of the two ML methods, NNE, is not very accurate.*

3. Can the approach work in a more complex model setup where biomass losses are also accounted for?

While I appreciate the question the study is asking and think that this work is important, I found that the manuscript was not very easy to follow (my summaries of the results above might illustrate this). Part of the difficulty may be that the topic is not within my immediate field of expertise, but then I feel there are stylistic changes to be made that will improve the manuscript. I have overall comments in the document below and I linked a **PDF document with comments at the very end of this document** (I used Adobe Reader). I hope these comments help improve the flow of the manuscript.

**2  Title**

The title can be improved. To someone who is not familiar with the "intrinsic" and "apparent" terminology, the title is not informative. Something along the lines of : *Can machine learning extract the mechanisms of phytoplankton growth from large-scale observations?*

**3   Abstract**

The use of "intrinsic" and "apparent" relationships this early in the manuscript made it difficult to understand the study as I am not familiar with the terminology.

**4   Introduction**

I don't have major concerns with the introduction and it builds a good case for why this study is relevant.

The questions posed (L72-75) and ideas presented (L100-102) are useful in framing the study but are not carried clearly through the manuscript. It would be very useful for the reader to have these questions and ideas as a guide for why each experiment was performed. For example, L72-75 from the basis of experiment 1, but these questions are not explicitly answered in the discussion. And lines 100-102 form the basis of the design for experiment 2.

**5   Methods**

Regarding the structure of the methods section, I have the following suggestions:

- There is no overview of the methods. I think this would be useful in addition to an accompanying diagram outlining all the experiments and the use of the machine learning approaches used. It would help the reader understand the flow of the study.

- BLING is used throughout the study, albeit with different outputs from the model, but it may make sense to introduce the model before the experiment configurations are described.

- It would make sense to formalise the following structure for each experiment:

  – A brief introduction to the experiment
  – HEADING for data
  – HEADING for Machine learning parameterisation / application

From a methodological point of view

- In experiment 2, the authors create hourly data by simulating variability of light conditions. The data are then averaged again to create daily, weekly and monthly data. If I understand correctly, the hourly data is analogous to the data used in experiment 1 - i.e. there is no temporal averaging in the "apparent data". It would be much more methodologically consistent to use the hourly data in experiment 1 and easier for the reader to follow. Either, the authors should implement this, or should make this explicit and state the reason that a separate experiment is needed.

- Another question is regarding the model: what is the variability of the nutrients at a daily resolution (native model resolution), and the averaged resolutions (weekly, monthly). Show some violin/box plots for the normalised data.

- I still don't fully understand what the predictors and target variables are for each experiment and what is the role of the intrinsic? From what I understand, predictors are always the "apparent" data and biomass is the target. The intrinsic is what describes the relationship between the biomass and the "apparent data". Please make this more clear. Addressing the points above in the structure section will help with this.

**6  Results**

- The authors should only NNE results for experiment 2 (figure 4). Is there a reason for this? My presumption is that the intrinsic relationship estimated by RF for micronutrients is poor, thus only NNE is shown. This should be cleared up (unless I missed this).

- From what I understand, the half-saturation constants are the metric for whether the method is able to capture the intrinsic from the apparent. Make this much more clear - also in the abstract

**7  Discussion**

- The subheadings could be the questions posed in the introduction (see my previous comments on this section). This would help guide the reader

- I think the authors should make the point that given the simplicity of the definition of biomass, one would expect the ML methods to perfectly represent the Michaelis-Menten curves. The authors do correctly state that RF is less likely to estimate accurately as the method is not able to extrapolate. This then increases the importance of showing the distributions of the training and test data set distributions. A further comment: what is the envelope around the estimated curves and why is there a large variability for the NNE at larger values?

- The discussion around scenario/experiment 3 is not clear and I don't feel that there is a take-home message after reading this section.

**8   Tables and Figures**

The captions are not standalone for both figures and tables.

The reader needs to know what the target variable in each table is and there are no units.

What is the envelope around the dashed lines.

Please also note the supplement to this comment:
https://bg.copernicus.org/preprints/bg-2020-262/bg-2020-262-RC2-supplement.pdf

**Supplement:**

[revised manuscript text omitted]

---

## Author Comment (AC1) · 24 Sep 2020

Author responses to Anonymous Referee 1

In the following responses RC stands for Referee Comment and AR stands for Author Response. For sections where draft paragraphs for the revised manuscript are included, the beginning and end of the draft paragraphs are denoted with *BD* (Begin Draft) and *ED* (End Draft).

For detailed descriptions of the tables and figures included with this Author Response,

please see the Supplemental PDF included with this Author Response.

RC0: The authors try to find a ML method that can establish and help to explain the link between intrinsic and apparent relationships of phytoplankton and environmental forcing. Three different methods were tested: Multiple Linear Regression (MLR), Random Forests (RFs) and Neural Network Ensembles (NNEs). The tests were provided on three different Scenarios. The authors found that the NNEs reproduce well the observed biomass (biomass estimated from intrinsic relationships) based on the knowledge of only apparent relationships when both relationships operate on the same spatial and temporal timescales. All methods fail when the intrinsic and apparent relationships operated on different timescales. However, the main authors' conclusion is that ML methods still can give an information on shapes of intrinsic relationships. Using the Earth System Models (ESM) in their third Scenario they show that the ML methods can extract useful information from this model that can used for an examination of interaction between input variables.

The article arises an interesting subject. However, it misses a clear explanation how the trained data and data for validation were constructed; the explanation of the role of input data, especially the physical meaning of the choice to fix them at 25th, 50th and 75th percentile for a sensitive analysis. I also think that the authors did not use all possible ML capacities for example, test more hidden layers in NNs, or add few more input variables, environmental ones, maybe a temperature.

Also, the main results that the NNEs can reproduce the general shape of intrinsic relationships does not have a systematic character: it is clearly seen that in some cases the NNEs shapes reproduce a different behavior compare to observation data. If the authors want to keep this conclusion, they will have to provide an additional analysis of conditions under which their conclusion is persistent. The article in its current state needs a serious major correction.

AR0: We want to thank Anonymous Referee #1. We have found the comments and

suggestions they provided to be very helpful in restructuring this manuscript.

RC1: It is hard to understand reading the introduction what is the main aim of the article. I have found three:

AR1: We understand how some of these statements may have caused confusion as to the main purpose of the paper. We address these in the following author responses AR1.1, AR1.2, AR1.3, and AR1.4.

RC1.1: "A significant challenge that remains is determining how intrinsic relationships found in the laboratory scale up to the apparent relationships observed at the ecosystem scale (i.e., scaling the small to the large)."

AR1.1: It was our intent that this served as more of a big-picture gap that has yet to be filled/solved. With this serving as the big-picture statement, it serves as a leadup to rest of the introduction. In this paper, we aren't necessarily trying to answer the entirety of this big-picture statement; rather, we are attempting to answer part of it by examining some smaller aspects of it.

RC1.2: "What is less clear is: 1. Can robust relationships be found? 2. If so, what methods are most skillful in finding them? 3. How do you interpret the apparent relationships that emerge when they diverge from the intrinsic relationships we expect?"

AR1.2: This statement was intended to serve as a link between the two paragraphs. However, we can see the confusion, especially given that the questions are numbered which would typically signify importance. We have removed this statement in its original paragraph and have incorporated the questions into the third statement.

RC1.3: "To investigate when and why the link between intrinsic and apparent relationships break. . ."

AR1.3: We have incorporated the questions from the second statement into the third statement to highlight the main purposes of the paper. The modified structure of the introduction that will be included in the revised manuscript as indicated in the draft

below in AR1.4.

RC1.4: If the two first citations can be linked to the title "What are the challenges?", the last one, what the authors actually did, is not according to the title "What are the challenges?". I will advise to authors modify the structure of the introduction and emphasize what is the main idea of this article.

AR1.4: We have changed the title of the manuscript to be more reflective of the main points. The current draft title replacement is "Can machine learning extract the mechanisms controlling phytoplankton growth from large-scale observations? – A proof of concept study"

Additionally, a modified introduction will be included in the revised manuscript. The portion of the introduction that highlights the main ideas of the article is in draft form below for reference:

*BD* To investigate when and why the link between intrinsic and apparent relationships break, we try to answer two main questions in this paper: 1. Can ML techniques find the correct underlying intrinsic relationships and, if so, what methods are most skillful in finding them? 2. How do you interpret the apparent relationships that emerge when they diverge from the intrinsic relationships we expect?

In addressing the first question, we first needed to demonstrate that we had an ML method that would correctly extract intrinsic relationships from apparent relationships. We constructed a simple model in which the intrinsic and apparent relationships operated on the same time and spatial scale and were only separated by a scaling factor, but in which the environmental drivers had realistic inter-relationships. Having a better handle on the results from the first question, we were able to move onto the second question where we look at where the link between intrinsic and apparent relationships break. We modified the first scenario to allow the intrinsic and apparent relationships to operate on different timescales – allowing us to evaluate the impact of time-averaging on the retrieval of intrinsic relationships. Finally, we conduct a proof-of-concept study

with real output from an ESM. *ED*

RC2: The conclusion also has to be rewritten in clear way: what are the challenges authors have found to link intrinsic and apparent relationships? The authors said: "Our main objective in this manuscript was to use ML to determine under what conditions intrinsic and apparent relationships between phytoplankton are no longer equal. . ." This objective was not clear in the introduction, and again, does not correspond well to the title of the article.

AR2: The revised manuscript will include a modified Conclusion section to better highlight the main objectives and points of the paper. A draft version of the new title and objectives are listed in AR1.4.

RC3: I advise you also to avoid the non-explained abbreviations in the abstract, like line 21: "ESM", line 28: "MLR".

AR3: We have removed the non-explained abbreviations from the abstract. They have been replaced with their unabbreviated definitions. This will be corrected in the revised manuscript.

RC4: It would be better not mentioning the results in the introduction (lines 100-103), and instead prepare your readers for the structure of the article.

AR4: We agree with this comment. The lines mentioning the results in the introduction have been removed. This will be corrected in the revised manuscript.

RC5: In the section "2. Methods" I suggest you provide a scheme or formula what exactly you were using as input/predictors and output/target and how it links to your equations 1 and 2 etc. This will simplify the understanding. It is especially important for the description of the Scenario 2. It is unclear, are the values of biomass with which the authors compare their results calculated based on hourly values? Is the biomass for target in learning algorithm smoothed or calculated based on smoothed predictors? It will strongly affect the results.

AR5: A new table will be included in the revised manuscript. The draft version is included (Table 1) which provides the following details for each scenario: the predictor variables, the target variable, the equations used in each scenario, a description of the source file, and a description of the scenario, such as how the target biomass was calculated.

RC6: The word "target" was not used in your article. It is common word in the ML domain and can also help to better understand the method especially for readers who only start to use ML techniques. Also, I would suggest to use the word "validation" instead of "testing".

AR6: We agree with this comment and have replaced the term "response variable(s)" with the term "target variable(s)". Additionally, we have replaced the term "testing subset(s)" with the term "validation subset(s)". This correction will be reflected in the revised manuscript.

RC7: To perform a sensitive analysis the authors fixed two of three predictors at different percentiles. It misses the explanation why 25th, 50th and 75th percentiles were chosen, what is the physical meaning of this choice and how it influences the results? For example, does 75th percentiles represent the extremes and what does it mean? I think that it is important to explain it to better understand the results. Please, clarify that the sensitivity analysis was done already on trained ML model.

AR7: The main reason for choosing these particular percentile values was so that we could examine conditions in a domain space that may be found as an actual observation. Additionally, we wanted to avoid extreme percentiles (1st and 99th) because the standard deviation in the predictions of the trained ML models is very large at these extremes.

We have also clarified that the sensitivity analysis was performed on the trained ML models. The revised version of the manuscript will include these changes, along with more details about how the sensitivity analysis was performed and explain the reasons

for why we chose particular percentile values.

A draft rewrite of the portion of the paper describing the sensitivity analyses is listed below:

*BD* Following this, a sensitivity analysis was performed on the trained ML models. We allowed one predictor to vary across its min-max range while holding the other two input variables at specific percentile values. This was repeated for each predictor. This allowed us to isolate the impact of each predictor on the biomass – creating "cross-sections" of the dataset where only one variable changes. For comparison, these values were also run through Eq. 1 and 2 to calculate the true intrinsic response of how the simple phytoplankton model would behave. This allowed us to view which of the models most closely reproduced the underlying intrinsic relationships of the simple phytoplankton model.

For the sensitivity analysis, we chose to hold the predictors that weren't being varied at their respective 25th, 50th, and 75th percentile values. We chose to use these percentile values for several reasons:

1. It allows us to avoid the extreme percentiles (1st and 99th). As we approach these extremes, the uncertainty in the predictions grows quite rapidly because of the lack of training samples within that domain space of the dataset. For example, there are no observations which satisfy the conditions of being in the 99th percentile of two variables simultaneously. This extreme distance outside of the training domain leads to standard deviations in predictions that are too large to provide a substantial level of certainty about the ML model's predictions.

2. Similar to the idea that we can avoid the extremes, we also chose these values as they are quite typical values for the edges of box plots. Generally, values within the range of the 25th to 75th percentiles are not considered outliers. Along those lines, we wanted to examine the conditions in a domain space that might be found in an actual observation, with the reasoning that if there was high uncertainty in the ML predictions
at these more moderate levels, there would be even higher uncertainty towards the extremes. *ED*

RC8: Did the authors try to increase the number of hidden layers in their NNs? It is known that the introduction of more hidden layers can improve the results. It would be interesting to see if there is any effect from the number of hidden layers in this particular problem.

AR8: In responding to this particular comment, we are assuming the Referee is referring to the outcomes of Scenario 2 since the performance in Scenario 1 was already robust.

For our manuscript we chose to use single hidden layer neural networks with 25 nodes in the hidden layer for several reasons:

1. We ran a diagnostic test to determine the ideal number of nodes for each neural network in Scenario 2 (Table 2). From the R2 values in Table 2, it can be observed that the performance of each NNE begins to plateau around 10-15 nodes. However, the inclusion of additional nodes up to about 25 nodes did not cause the training times to increase drastically. We chose to include these extra nodes due to their moderate training times and robust performance.

2. All three Scenarios showed ideal performances in the 10-20 node range, but similar to the first point, we chose to include extra nodes due to their moderate training times and good performance. Additionally, this allowed us to be consistent in the architecture of the neural networks across all the Scenarios and helped to minimize the differences between them.

3. We considered adding more hidden layers to the neural networks (Table 3), but the slight increase in performance did not seem worth while due to significantly increased training times. As can be observed in Table 3, the performance did not significantly increase from a single hidden layer to when a second layer with additional nodes was

included (0.9700 vs 0.9722-0.9726).

4. Although some applications may require additional hidden layers, it is generally accepted that one hidden layer can approximate most functions provided there are sufficient nodes in the hidden layer and the nodes utilize squashing functions, such as sigmoid functions (Hornik et al., 1989).

Hornik, K., Stinchcombe, M. and White, H.: Multilayer feedforward networks are universal approximators, Neural Networks, 2(5), 359–366, doi:10.1016/0893-6080(89)90020-8, 1989.

RC9: The authors did not provide how they scaled their variables (lines 337-341). This procedure is known as normalization of variables. Normalization ensures that all predictors fall within a comparable range and avoids giving more weight to predictors with large variability ranges.

AR9: We have included the equations that we used to scale and unscale the values of each variable (Eq. 1 and 2; attached to this Author Response). The revised version of the manuscript will include these equations, along with the details pointed out by Referee 1 in RC9. A draft version for the revised manuscript is below:

*BD* Each variable was scaled between -1 and 1 based on its respective maximum and minimum (Eq. 1).

(See Eq. 1 below)

Where V is the value of the variable being scaled, S stands for the scaled value, and U represents the unscaled value. This step ensures that no values are too close to the limits of the hyperbolic tangent sigmoid activation function, which would significantly increase the training time of each NN. Additionally, this normalization ensures that each predictor falls within a similar range so more weight is not provided to variables with larger ranges. These scalings were also applied to the RF and MLR methods for consistency between methods which did not affect the results of either method (results

not shown). The results presented in this paper were then transformed back to their original scales to avoid confusion from scaling (Eq. 2).

(See Eq. 2 below)

Where the letters represent the same values as those in Eq. 1. *ED*

RC10: It is hard to agree that the NNEs and RFs represent well the behavior as for example on Figure 3 in left column the NNEs show a strong increase at the end of the macronutrient range that does not in agreement with the observation values; and there is a false decrease in the middle column at 25th percentile.

AR10: These behaviors likely have several causes:

1. In the areas of the dataspace where there are few or no observations, the NNEs are less certain about the predictions. This can be noted as the large gray areas in the middle column for the 25th percentile in Fig. 3. Additionally, the uncertainties in the previously mentioned subplot may seem large due to the magnitude of the predictions for those conditions. For example, the y-axis of the middle column for the 25th percentile of Fig. 3, shows the biomass in the range of 10-7. The other percentile plots in the middle column of Fig. 3 show biomass in the range of 10-6, a full order of magnitude larger than the biomass in the 25th percentile of the same column.

2. Since the lines in the sensitivity analyses for the NNEs are the average response of ten individual neural networks, it's possible that the NNE line in the sensitivity analyses could be pulled higher or lower in areas of higher uncertainty due to extreme outlying predictions. This might explain the behavior of the increase in biomass at the end of the macronutrient range in the 75th percentile subplot of the left column in Fig. 3. Additionally, since the random forest is an average of 500 trees it may be less susceptible to extreme outlying predictions, which would help to explain why that behavior is not seen in the random forest predictions for the previously mentioned macronutrient subplot.

We have only listed some of the potential causes, so others may exist to explain the

deviations in behavior. We will address these in more detail in the revised manuscript.

RC11: It is interesting to know if the authors have an idea about new parameter or variable that can bring back the information on hourly variability lost due to the time-averaging to improve the results?

AR11: We have considered a couple of ideas to attempt to bring back the variability lost due to the time-averaging:

1. We implemented a method that allowed us to visualize how the apparent relationships change from the hourly timescale through to the monthly averaged timescale.

To capture the apparent relationships at various timescales between the hourly and monthly timescales, we averaged the hourly dataset over a range of hourly timespans. Specifically, we averaged over timescales including 1 hour (original hourly set), 2, 3, 4, 6, 8, 12, 24, 48, 72, 168 (weekly), and 720 (monthly) hours. It was necessary to average over timespans that were multiples of, or divisible by, 24. Sets of hours that did not meet this criteria meant that hours from one day were ultimately being averaged with hours from another day. For example, using a 7 hour timespan for averaging would mean that the last three hours of Day 1 were then being averaged with the first four hours of Day 2.

This new set of averaged timescales was then used to train NNEs with one NNE corresponding to each averaged timescale. To be consistent, the NNEs were trained according to the same specifications listed in the original manuscript (10 individual NNs in each NNE, 25 nodes in hidden layer, one hidden layer, three predictors, stopping criteria, etc.).

We then performed sensitivity analyses on each of the trained NNEs to see the apparent relationships for each averaged timescale and set the percentile values for the other variables at their 50th percentile (median). To visualize all the timescales at once, we plotted them on contour plots (Fig. 1). The left three plots (Fig. 1 a, c, e) show the

full span on the timescales we tested. The greatest changes in the apparent relationships were in the first 24 hours, which is difficult to see because of the difference in magnitude of the hourly and monthly averaged timescales. Since there is little change in the apparent relationships past the 24-hour averaged dataset, we decided to examine the relationships of the timescales at and below 24 hours (Fig. 1 b, d, f). Again, what we are seeing is that the more the dataset is averaged, the more variability is lost, and the greater the underestimation of the relationships. However, what we also see is that many of the apparent relationships below 12 hours are fairly close to the 1-hour apparent relationships. It may be possible that one may be able to use this type of diagnostics test to know learn the necessary frequencies to recover the true relationships.

2. We included additional variables and tested different percentile values.

We included the length of day (the number of hours in a day that a particular location received sunlight above 0 W m-2) as a predictor for each of the time-averaged datasets of Scenario 2 and trained a new set of NNEs. With the exception of the number of predictors, the NNEs followed the same specifications listed in the original manuscript (10 individual NNs in each NNE, 25 nodes in hidden layer, one hidden layer, stopping criteria, etc.).

We knew from previously testing the higher percentile values (>90th percentile) for the sensitivity analyses that the higher percentiles led to greater uncertainty in the predictions (Fig. 2). Although the true relationships were captured in the 100th percentile plots, insofar as they were within the gray standard deviations of predictions, the uncertainty in the predictions had a range equal to or greater than the response (Fig. 2 g, h, i).

When we included length of day as a predictor and used the high percentile values of the other variables, we found that the 100th percentile plots captured the true relationship and the gray standard deviation ranges decreased (Fig. 3 i, j, k, l). Furthermore,

all of three of the time-averaged datasets were able to capture the correct relationships.

We will include this new result, with some discussion, in the revised manuscript.

RC12: The article misses the total results for Scenarios 2 and 3 like on Figure 1.

AR12: The contour plots of Figure 1 were mostly intended to provide a visual representation of the difference in performance of the ML methods compared to multiple linear regression. Table 1 serves the same purpose Figure 1, but with performance metrics instead visual representation. Additionally, Table 1 provides quantifiable measures of each method's performance, whereas Figure 1 only allows for a visual qualitative measure. Since Table 1 serves as a quantifiable metric of performance we chose to use only tables for Scenarios 2 and 3, instead of contour plots like those in Figure 1.

RC13: In the Discussion of results for Scenario 3 the authors reasoned about BLING model behavior and did not mention their results. It would be interesting to know the authors' thoughts about NNEs behavior on Figure 7 middle column at 75th percentile.

AR13: From our best estimation, we assume that Referee 1 is indicating that we do not mention the specific results of any one particular ML method. For example, we didn't differentiate between the responses found by RFs and NNEs.

In the Discussion section of Scenario 1, we mention that RF does not possess the same extrapolation capabilities as NNEs. Because of RF's inability to extrapolate, we focused our discussion of Scenario 3 on the relationships found by NNEs. However, it is now clear to us that we should either specify that point in the Discussion of Scenario 3 or discuss the relationships found by each method. The revised version of the manuscript will include the previously mentioned specification or will discuss the relationships found by each individual method.

RC14: I have also found that the authors wrote too much on things that they did not do: lines 171-177.

AR14: We originally included this section because partial dependence plots (PDP)

are a common visualization technique used in ML, especially for random forest (RF) analysis. In previous versions of this manuscript, initial comments (before submission to any journal) typically cited the use of PDPs vs sensitivity analyses as being an important distinction. So we felt it necessary to provide a brief explanation of how the use of sensitivity analyses differs from PDPs.

RC15: Lines 260 – 273: I suggest to rewrite these two paragraphs, there is a mention of results that have not yet been presented; and I do not see the necessity to mention that the authors "previously had little experience with ML."

AR15: We understand the confusion with these paragraphs. In past submissions, we were asked several questions:

1. Why had we chosen to use Random Forests and Neural Network Ensembles, instead of other machine learning algorithms? 2. Why did we choose to compare the performance of the machine learning algorithms to Multiple Linear Regression?

We tried to address these questions with these two paragraphs (lines 260-273), but we acknowledge that we may have included additional details that may not be needed. To address these two questions and still include details we find necessary, we have shortened the first of the two paragraphs mentioned in the Referee Comment. This condensed version will be included in the revised manuscript. A draft of the condensed paragraph is included below:

*BD* We chose to use Random Forests (RFs) and Neural Network Ensembles (NNEs) in this manuscript. Although other ML methods exist, the list of possible choices is rather long. It was decided that the number of ML algorithms being compared would be limited to RFs and NNEs given their popularity in studying ecological systems. Additionally, we chose to compare the performance of the ML techniques to the performance of Multiple Linear Regression which allows for a comparison of linear and non-linear methods. *ED*

Regarding the second paragraph, comments by other Referees have suggested the second paragraph (lines 270 to 273) remain unaltered. It provides an explicit statement by us that we are not trying to invalidate results or discourage the use of MLR in marine ecological systems.

RC16: Please use the figures captions like "a", "b" etc.

AR16: All figures that have subplots now have letter captions for each of the subplots. Additionally, the figure descriptions will refer to the letter captions for subplots instead of their location within each figure. For example when referring to a subplot, it is cited as "Box a" instead of the "top-left subplot in Fig. XX". These corrections will be reflected in the revised manuscript.

RC17: Please, expand the figure captions, for example on Figure 2 it would be good to add that the black line is estimated from Eq. 1.

AR17: The figure and table captions will be expanded in the revised version of the manuscript to include more specific details. A draft version for Figure 2 is included below as an example.

*BD* Figure 2: Sensitivity analysis for Scenario 1 showing the true and predicted relationships for how each predictor affects the biomass when the other predictors are set at specific percentiles. The columns correspond to the predictors and the rows correspond with the percentile value at which the other predictors were set. The black line shows the true intrinsic relationship calculated from Eq. 1 and 2. The dashed lines show the predicted apparent relationships for each method (MLR – red; RF – blue; NNE – green). The gray region around the RF and NNE dashed lines shows the standard deviation of the predictions. *ED*

RC18: Figure 5 was mentioned as the last but it is placed before Figure 6 and 7.

AR18: The order of the figures will be corrected in the revised manuscript so each figure is numbered based upon when it is first mentioned in the manuscript.

RC19: Line 149, 156, 168, 356: please avoid to use the sign " in scientific paper.

AR19: We removed the quotations around the words in the specified lines. Additionally, we have reexamined our usage of other quotation marks throughout the paper and have removed quotations we deemed unnecessary. This will be reflected in the revised manuscript.

RC20: Line 149: sign " should be before the point .

AR20: We removed the quotations around that particular term. This will be reflected in the revised manuscript.

RC21: Line 206: it feels that "but" should be replaced by "and".

AR21: We replaced the transition word "but" with the transition word "and". This change will be reflected in the revised manuscript.

RC22: Line 226, 235, 543: word "just" is unnecessary.

AR22: We removed the word "just" from the suggested lines. This will be reflected in the revised manuscript.

RC23: Line 250: "an ML" should be replaced by "a ML".

AR23: We replaced all instances of "an ML" with "a ML". This will be reflected in the revised manuscript.

RC24: Line 356: Miss a figure indication.

AR24: We filled in the missing figure indication. This change will be reflected in the revised manuscript.

Please also note the supplement to this comment:
https://bg.copernicus.org/preprints/bg-2020-262/bg-2020-262-AC1-supplement.pdf

$$V_S = \frac{max_S - min_S}{max_U - min_U} \left(V_U - min_U\right) + min_S \tag{1}$$

**Fig. 1.** Equation 1. Equation used to scale each variable between -1 and 1.

$$V_U = \frac{max_U - min_U}{max_S - min_S}\ (V_S - min_S) + min_U \qquad (2)$$

**Fig. 2.** Equation 2. Equation used to scale each variable back to its original values.

| Scenario | Predictors | Target | Equations Used | Source File Description | Scenario Description |
|---|---|---|---|---|---|
| 1 | Macronutrient (mol kg$^{-1}$); Micronutrient (mol kg$^{-1}$); Irradiance (W m$^{-2}$) | Biomass (mol kg$^{-1}$) | 1, 2 | Monthly Output from BLING | Nutrient distributions (predictors) from BLING were fed to Eq. 1 and 2 to calculate the biomass (target) |
| 2 | Macronutrient (mol kg$^{-1}$); Micronutrient (mol kg$^{-1}$); Irradiance (W m$^{-2}$) | Biomass (mol kg$^{-1}$) | 1, 2, 5 | Daily Output from BLING | 1) Hourly values for the predictors were interpolated using the Daily Output of BLING

1a) The macronutrient and micronutrient hourly values were calculated using a standard interpolation between the daily points.

1b) The irradiance hourly values were calculated from Eq. 5 using the value of the BLING daily input, hour of day, time of year, and location.

2) Hourly values of the predictors were fed to Eq. 1 and 2 to calculate hourly values for the biomass (target)

3) Daily-averaged values were calculated by averaging 24 hours for each location through one year

4) Weekly-averaged values were calculated by averaging 168 hour blocks of time for each location through the year

5) Monthly-averaged values were calculated by averaging the number of hours in each month (days per month * 24) for each location through the year

6) The true relationships were calculated by using the range of the hourly values for the predictors and calculating the biomass based on Eq. 1 and 2. |
| 3 | Macronutrient (mol kg$^{-1}$); Micronutrient (mol kg$^{-1}$); Irradiance (W m$^{-2}$) | Biomass (mol kg$^{-1}$) | 6, 7 (Equations within BLING used to determine the biomass) | Monthly Output from BLING | Nutrient distributions from the BLING Output were used as the predictors; Biomass from the BLING Output itself was used as the target |

**Fig. 3.** Table 1: Details for each Scenario.

|  | R² Values | | |
|---|---|---|---|
| Number of Nodes | Daily | Weekly | Monthly |
| 1 | 0.5533 | 0.5472 | 0.5624 |
| 2 | 0.7655 | 0.7705 | 0.7806 |
| 5 | 0.9283 | 0.9248 | 0.9363 |
| 10 | 0.9633 | 0.9628 | 0.9673 |
| 15 | 0.9676 | 0.9678 | 0.9713 |
| 20 | 0.9693 | 0.9694 | 0.9727 |
| 25 | 0.9700 | 0.9702 | 0.9732 |
| 35 | 0.9709 | 0.9709 | 0.9737 |
| 50 | 0.9716 | 0.9715 | 0.9743 |

**Fig. 4.** Table 2: The R2 values for the diagnostic test used to determine the how the number of nodes in the hidden layer of a single layer neural network affected the performance of the Scenario 2 datasets.

|  |  | $R^2$ Values | | |
| --- | --- | --- | --- | --- |
|  |  | Daily | Weekly | Monthly |
| Layers and | 25 | 0.9700 | 0.9702 | 0.9732 |
| Number of | 25-10 | 0.9722 | 0.9724 | 0.9750 |
| Nodes | 25-25 | 0.9726 | 0.9727 | 0.9756 |

**Fig. 5.** Table 3: The R2 values for the diagnostic test used to determine the how the number of hidden layers and nodes within individual neural networks affected the performance of the Scenario 2 datasets.

**a**

**b**

**c**

**d**

**e**

**f**

**Fig. 6.** Figure 1: Contour plots showing the apparent relationships found across different averaged timescales for Scenario 2.

[Figure]

**Fig. 7.** Figure 2: Sensitivity analysis for Scenario 2 showing the true and predicted relationships for how each predictor affects the biomass when the other predictors are set at higher percentiles.

**Fig. 8.** Figure 3: Sensitivity analysis for Scenario 2 with Length of Day as an added predictor showing the true and predicted relationships for how each predictor affects the biomass using high percentiles.

**Supplement:**

Table 1: Details for each Scenario that include the predictor variables, the target variable, the equations used to calculate biomass, the type of source file used to acquire the values for the predictors, and a short description of each scenario.

| Scenario | Predictors | Target | Equations Used | Source File Description | Scenario Description |
|---|---|---|---|---|---|
| 1 | Macronutrient (mol kg$^{-1}$); Micronutrient (mol kg$^{-1}$); Irradiance (W m$^{-2}$) | Biomass (mol kg$^{-1}$) | 1, 2 | Monthly Output from BLING | Nutrient distributions (predictors) from BLING were fed to Eq. 1 and 2 to calculate the biomass (target) |
| 2 | Macronutrient (mol kg$^{-1}$); Micronutrient (mol kg$^{-1}$); Irradiance (W m$^{-2}$) | Biomass (mol kg$^{-1}$) | 1, 2, 5 | Daily Output from BLING | 1) Hourly values for the predictors were interpolated using the Daily Output of BLING
  1a) The macronutrient and micronutrient hourly values were calculated using a standard interpolation between the daily points.
  1b) The irradiance hourly values were calculated from Eq. 5 using the value of the BLING daily input, hour of day, time of year, and location.
2) Hourly values of the predictors were fed to Eq. 1 and 2 to calculate hourly values for the biomass (target)
3) Daily-averaged values were calculated by averaging 24 hours for each location through one year
4) Weekly-averaged values were calculated by averaging 168 hour blocks of time for each location through the year
5) Monthly-averaged values were calculated by averaging the number of hours in each month (days per month * 24) for each location through the year
6) The true relationships were calculated by using the range of the hourly values for the predictors and calculating the biomass based on Eq. 1 and 2. |
| 3 | Macronutrient (mol kg$^{-1}$); Micronutrient (mol kg$^{-1}$); Irradiance (W m$^{-2}$) | Biomass (mol kg$^{-1}$) | 6, 7 (Equations within BLING used to determine the biomass) | Monthly Output from BLING | Nutrient distributions from the BLING Output were used as the predictors; Biomass from the BLING Output itself was used as the target |

Table 2: The $R^2$ values for the diagnostic test used to determine the how the number of nodes in the hidden layer of a single layer neural network affected the performance of the time-averaged datasets of Scenario 2. The target variable was biomass (mol kg$^{-1}$). A separate NNE was trained for each of the time-averaged datasets (daily, weekly, monthly) for each set of nodes (ex. A unique NNE for the daily-averaged dataset with 1 node was trained, a unique NNE for the weekly-averaged dataset with 1 node was trained, etc.). Each NNE contained 10 individual neural networks and kept the same training and stopping specifications outlined in the original manuscript. The trained NNEs made predictions on the validation subset and the $R^2$ values were calculated based on the comparison between those predictions and the actual values of the validation subset.

|  |  | $R^2$ Values | | |
|  |  | Daily | Weekly | Monthly |
| --- | --- | --- | --- | --- |
|  | 1 | 0.5533 | 0.5472 | 0.5624 |
|  | 2 | 0.7655 | 0.7705 | 0.7806 |
|  | 5 | 0.9283 | 0.9248 | 0.9363 |
|  | 10 | 0.9633 | 0.9628 | 0.9673 |
| Number of | 15 | 0.9676 | 0.9678 | 0.9713 |
| Nodes | 20 | 0.9693 | 0.9694 | 0.9727 |
|  | 25 | 0.9700 | 0.9702 | 0.9732 |
|  | 35 | 0.9709 | 0.9709 | 0.9737 |
|  | 50 | 0.9716 | 0.9715 | 0.9743 |

Table 3: The $R^2$ values for the diagnostic test used to determine the how the number of hidden layers and nodes within individual neural networks affected the performance of the Scenario 2 time-averaged datasets. The target variable was biomass (mol kg$^{-1}$). A separate NNE was trained for each of the time-averaged datasets (daily, weekly, monthly) for each set of nodes (ex. A unique NNE for the daily-averaged dataset with 25 nodes was trained, a unique NNE for the weekly-averaged dataset with 25 nodes was trained, etc.). Each NNE contained 10 individual neural networks and kept the same training and stopping specifications outlined in the original manuscript. The trained NNEs made predictions on the validation subset and the $R^2$ values were calculated based on the comparison between those predictions and the actual values of the validation subset. The layers and number of nodes in the table are specified as follows: # nodes in first layer - # nodes in second layer. If only one number is listed, this specifies the number of nodes in the single hidden layer and that a second layer was not used.

| | | $R^2$ Values | | |
| --- | --- | --- | --- | --- |
| | | Daily | Weekly | Monthly |
| Layers and | 25 | 0.9700 | 0.9702 | 0.9732 |
| Number of | 25-10 | 0.9722 | 0.9724 | 0.9750 |
| Nodes | 25-25 | 0.9726 | 0.9727 | 0.9756 |

[Figure]

Figure 1: Contour plots showing the apparent relationships found across different averaged timescales for Scenario 2. The timescales range from 1 hour (original hourly set) up to the 720 hours (monthly). The three plots on the left show the relationships across the entire range of timescales (1 through 720 hours). The three plots on the right show the timescales at and below 24 hours. The top plots show the relationships for the macronutrient, the middle plots show the relationships for the micronutrient, and the bottom plots show the relationships for irradiance. Variables not varying across their range were set at their 50th percentile (median) value. The conditions of the sensitivity analyses were based on the conditions of the monthly-averaged

(720-hour) dataset. It was necessary to give the same conditions to the all of the time-averaged datasets so that a direct comparison could be made between the predictions from each time-averaged dataset.

[Figure]

Figure 2: Sensitivity analysis for Scenario 2 showing the true and predicted relationships for how each predictor affects the biomass when the other predictors are set at higher percentiles (90, 95, and 100). The columns correspond to the predictors and the rows correspond with the percentile value at which the other predictors were set. The black line shows the true intrinsic relationship calculated from Eq. 1 and 2. The dashed lines show the predicted apparent relationships for each time-averaged dataset (Daily – red; Weekly – blue; Monthly – green). The gray region around the dashed lines shows the standard deviation of the predictions. The conditions of the sensitivity analyses were based on the conditions of the monthly-averaged dataset. It was necessary to give the same conditions to the all of the time-averaged datasets and to the simple model so that a direct comparison could be made between the true response and the predictions from each time-averaged dataset.

[Figure]

Figure 3: Sensitivity analysis for Scenario 2 with Length of Day as an additional predictor showing the true and predicted relationships for how each predictor affects the biomass when the other predictors are set at higher percentiles (90, 95, and 100). The columns correspond to the predictors and the rows correspond with the percentile value at which the other predictors were set. The black line shows the true intrinsic relationship calculated from Eq. 1 and 2. The dashed lines show the predicted apparent relationships for each time-averaged dataset (Daily – red; Weekly – blue; Monthly – green). The gray region around the dashed lines shows the standard deviation of the predictions. The conditions of the sensitivity analyses were based on the conditions of the monthly-averaged dataset. It was necessary to give the same conditions to the all of the time-averaged datasets and to the simple model so that a direct comparison could be made between the true response and the predictions from each time-averaged dataset.

---

## Author Comment (AC2) · 25 Sep 2020

Author responses to Luke Gregor (Referee 2)

In the following responses RC stands for Referee Comment and AR stands for Author Response. For sections where draft paragraphs for the revised manuscript are included, the beginning and end of the draft paragraphs are denoted with *BD* (Begin Draft) and *ED* (End Draft).

For detailed descriptions of the tables and figures included with this Author Response,

please see the Supplemental PDF included with this Author Response.

RC0: The study by Holder and Gnanadesikan tries to assess if machine learning is able to extract the intrinsic relationship between phytoplankton growth and limiting nutrients and light from observed concentrations of nutrients and light intensity. This topic was investigated with three experiments of increasing complexity asking the following questions (with my brief understanding of the outcomes):

1. Are ML methods able to extract the relationship from observations at all at instantaneous time scales? a. Yes, but NNE is better at extracting the relationship than RF despite both achieving fair results 2. If time scales are averaged, can the relationships still be extracted? a. Not very well. In most cases the estimated half-saturation is lower than it should be. I.e. even the better of the two ML methods, NNE, is not very accurate. 3. Can the approach work in a more complex model setup where biomass losses are also accounted for?

While I appreciate the question the study is asking and think that this work is important, I found that the manuscript was not very easy to follow (my summaries of the results above might illustrate this). Part of the difficulty may be that the topic is not within my immediate field of expertise, but then I feel there are stylistic changes to be made that will improve the manuscript. I have overall comments in the document below and I linked a PDF document with comments at the very end of this document (I used Adobe Reader). I hope these comments help improve the flow of the manuscript.

AR0: We would like to thank Luke Gregor as Referee 2. We have found the comments and suggestions they provided to be very helpful in restructuring this manuscript. In particular, the supplement to these comments has provided some very specific and constructive feedback.

RC1: The title can be improved. To someone who is not familiar with the "intrinsic" and "apparent" terminology, the title is not informative. Something along the lines of : Can machine learning extract the mechanisms of phytoplankton growth from large-scale

observations?

AR1: We understand how including the terminology in the title can lead to confusion. The revised manuscript will have a different title. The current draft title we have is "Can machine learning extract the mechanisms controlling phytoplankton growth from large-scale observations? – A proof of concept study".

RC2: The use of "intrinsic" and "apparent" relationships this early in the manuscript made it difficult to understand the study as I am not familiar with the terminology.

AR2: The terms "intrinsic" and "apparent" relationships are actually terms that we are defining for the first time in this manuscript. They have not been previously introduced in oceanography literature. Since these terms are used frequently throughout the paper, we find it helpful to introduce them early, including in the abstract.

RC3: I don't have major concerns with the introduction and it builds a good case for why this study is relevant.

AR3: We thank Referee 2 for this kind complement.

RC4: The questions posed (L72-75) and ideas presented (L100-102) are useful in framing the study but are not carried clearly through the manuscript. It would be very useful for the reader to have these questions and ideas as a guide for why each experiment was performed. For example, L72-75 from the basis of experiment 1, but these questions are not explicitly answered in the discussion. And lines 100-102 form the basis of the design for experiment 2.

AR4: We have changed some aspects of the introduction based on other Referee comments, which encompass similar feedback as in RC4 above. In the revised version of the manuscript we plan to remove lines 100-102, as these lines list results in the introduction. We also plan to modify the introduction in the revised manuscript. A draft form of a portion of the introduction that more clearly highlights the main points of the paper is listed below:

*BD* To investigate when and why the link between intrinsic and apparent relationships break, we try to answer two main questions in this paper: 1. Can ML techniques find the correct underlying intrinsic relationships and, if so, what methods are most skillful in finding them? 2. How do you interpret the apparent relationships that emerge when they diverge from the intrinsic relationships we expect? In addressing the first question, we first needed to demonstrate that we had an ML method that would correctly extract intrinsic relationships from apparent relationships. We constructed a simple model in which the intrinsic and apparent relationships operated on the same time and spatial scale and were only separated by a scaling factor, but in which the environmental drivers had realistic inter-relationships. Having a better handle on the results from the first question, we were able to move onto the second question where we look at where the link between intrinsic and apparent relationships break. We modified the first scenario to allow the intrinsic and apparent relationships to operate on different timescales – allowing us to evaluate the impact of time-averaging on the retrieval of intrinsic relationships. Finally, we conduct a proof-of-concept study with real output from an ESM. *ED*

RC5: There is no overview of the methods. I think this would be useful in addition to an accompanying diagram outlining all the experiments and the use of the machine learning approaches used. It would help the reader understand the flow of the study. BLING is used throughout the study, albeit with different outputs from the model, but it may make sense to introduce the model before the experiment configurations are described.

AR5: We have included a diagram (Table 1) outlining the details of each scenario which include: the predictor variables, the target variable, the equations used to calculate biomss, a description of the source file, and a short description of each scenario.

Because the machine learning approaches are the same for each Scenario, we didn't think it would be necessary to include a table or diagram showing this. However, in the revised manuscript, we will state more clearly in the methods that the same machine

learning approaches are used for each scenario.

The main reason for including the description of BLING in the third scenario was so readers would not get confused as to which equations and model are being used for each scenario if it was introduced before the explanation of the Scenarios. However, we will consider whether to move the BLING description before the scenario explanations in the revised manuscript now that we have included a diagram (Table 1) outlining the details of each scenario.

RC6: It would make sense to formalise the following structure for each experiment:

• A brief introduction to the experiment

• HEADING for data

• HEADING for Machine learning parameterisation / application

AR6: We agree with the idea about formalizing the structure of each experiment. The revised manuscript will include a structure for each experiment similar to that described in RC6 above.

RC7: In experiment 2, the authors create hourly data by simulating variability of light conditions. The data are then averaged again to create daily, weekly and monthly data. If I understand correctly, the hourly data is analogous to the data used in experiment 1 - i.e. there is no temporal averaging in the "apparent data". It would be much more methodologically consistent to use the hourly data in experiment 1 and easier for the reader to follow. Either, the authors should implement this, or should make this explicit and state the reason that a separate experiment is needed.

AR7: Yes, the Referee is correct in their understanding that the hourly data is analogous to the data used in Scenario 1 where there was no temporal averaging. We agree that it would be easier for the reader to follow, and we spent several days testing this strategy.

The main issue we ran into was with the size of the hourly dataset. Across all longitudes, latitudes, and hours for a single year, this results in a dataset with 56,214,560 observations. We attempted to randomly sample the dataset with up to 500,000 points to train the machine learning algorithms. Quantities of observations higher than 500,000 were leading to computer crashes because of the computational power required for training the ML algorithms. While it is technically feasible to train random forests and neural networks on this number of observations, this would still require very long spans of time for training each ML method. Since we would like this paper and the methods to be accessible to everyone, we would like our Matlab code to be able to run on a standard laptop. With this in mind, we chose the first BLING scenario since it was already at monthly timescales and the number of observations was significantly less than the amount in an hourly dataset over the course of a year. The number of samples in the monthly dataset of Scenario 1 is only 77,328 compared to the 56 million of the hourly dataset.

Additionally, as we now show (please see our response AR11 to Referee 1) that adding length of day as a variable or going to very high percentiles of other variables does appear to allow the NNEs to correctly extrapolate the correct relationships even in the time-averaged datasets.

RC8: Another question is regarding the model: what is the variability of the nutrients at a daily resolution (native model resolution), and the averaged resolutions (weekly, monthly). Show some violin/box plots for the normalised data.

AR8: A figure including boxplots for the time-averaged datasets of Scenario 2 will be included in the revised manuscript. A draft version of that figure is included below in Fig. 1.

RC9: I still don't fully understand what the predictors and target variables are for each experiment and what is the role of the intrinsic? From what I understand, predictors are always the "apparent" data and biomass is the target. The intrinsic is what describes

the relationship between the biomass and the "apparent data". Please make this more clear. Addressing the points above in the structure section will help with this.

AR9: With the inclusion of Table 1 below, the predictors and target variables for each Scenario are included there. However, the revised version of the manuscript will also clarify this in the methods section as well.

Regarding the intrinsic and apparent relationships, the intrinsic relationships are those in which the effects of other variables affecting the target variable can be accounted. For example, if one is measuring the effect of macronutrient concentrations on phytoplankton in the lab, it is possible for them to hold concentrations of other variables (light, micronutrient, water temperature, salinity, etc.) at some particular value. Apparent relationships are those for which the effects of other variables affecting the target variable cannot be accounted (ex. taking measurements in the field). Another way of saying this is that intrinsic relationships are the underlying relationships governing a system where you can adjust one variable at a time (such as a lab). Apparent relationships are determined by how the intrinsic relationships combine in the environment when variables cannot be adjusted one variable at a time. We will try to clarify this distinction in the revised manuscript.

RC10: The authors should only NNE results for experiment 2 (figure 4). Is there a reason for this? My presumption is that the intrinsic relationship estimated by RF for micronutrients is poor, thus only NNE is shown. This should be cleared up (unless I missed this).

AR10: The presumption of the Referee is correct. Because the RF performs poorly and is incapable of extrapolating outside the range of the training dataset, we chose to limit further analyses of Scenario 2 to NNEs. We will clarify this in the revised manuscript.

RC11: From what I understand, the half-saturation constants are the metric for whether the method is able to capture the intrinsic from the apparent. Make this much more clear - also in the abstract

AR11: That is correct. We are using the calculated half-saturation constants as a metric to help identify if the methods are capturing the true relationships. We will clarify this in the revised manuscript.

RC12: The subheadings could be the questions posed in the introduction (see my previous comments on this section). This would help guide the reader

AR12: Yes, we agree that subheadings in the discussion could aid in guiding the reader. As the Referee suggests, we will consider using the questions posed in the introduction as subheadings. The revised manuscript may include subheadings in the discussion section.

RC13: I think the authors should make the point that given the simplicity of the definition of biomass, one would expect the ML methods to perfectly represent the Michaelis-Menten curves. The authors do correctly state that RF is less likely to estimate accurately as the method is not able to extrapolate. This then increases the importance of showing the distributions of the training and test data set distributions. A further comment: what is the envelope around the estimated curves and why is there a large variability for the NNE at larger values?

AR13: To keep the number of figures in the manuscript to a minimum, we had not included boxplots of each variable in each Scenario. However, we see the use that information can provide. The revised manuscript may include the distributions of the training and test subsets for each Scenario in the Supplementary Materials section.

The gray regions around the dashed lines for the random forest (RF) and neural network ensemble (NNE) predictions show the standard deviation in the predictions. For example, the NNEs are composed of 10 individual neural networks and each one produces its own predictions. For the sensitivity analysis figures, the dashed lines for NNE show the average prediction of those 10 individual neural networks. Similarly, the gray regions show the range of one standard deviation for those predictions. We will clarify this in the revised manuscript.

The large variability for the NNEs at the larger values is likely because those particular conditions are outside the range of the dataset on which the NNEs were trained. For example, it is rare that any of the observations would have high macronutrient, high micronutrient, and high irradiance occurring at the exact same time and location. Without any observations in the training subset meeting those types of criteria and the NNE never having seen what those conditions actually produce, the NNE predictions become less certain.

RC14: The discussion around scenario/experiment 3 is not clear and I don't feel that there is a take-home message after reading this section.

AR14: The purpose of Scenario 3 is largely to provide a proof-of-concept to how the techniques we demonstrate in Scenarios 1 and 2 can be applied to Earth System Model output. The revised manuscript will expand on this and better highlight the main goal of Scenario 3.

RC15: The captions are not standalone for both figures and tables.

AR15: The revised manuscript will include more detailed descriptions of the tables and figures. A draft version for Figure 2 of the original manuscript currently reads:

*BD* Figure 2: Sensitivity analysis for Scenario 1 showing the true and predicted relationships for how each predictor affects the biomass when the other predictors are set at specific percentiles. The columns correspond to the predictors and the rows correspond with the percentile value at which the other predictors were set. The black line shows the true intrinsic relationship calculated from Eq. 1 and 2. The dashed lines show the predicted apparent relationships for each method (MLR – red; RF – blue; NNE – green). The gray region around the RF and NNE dashed lines shows the standard deviation of the predictions. *ED*

RC16: The reader needs to know what the target variable in each table is and there are no units.

AR16: The revised manuscript will include the target variable and its units in the description of each table.

RC17: What is the envelope around the dashed lines.

AR17: The gray regions around the dashed lines for the random forest (RF) and neural network ensemble (NNE) predictions show the standard deviation in the predictions. For example, the NNEs are composed of 10 individual neural networks and each one produces its own predictions. For the sensitivity analysis figures, the dashed lines for NNE show the average prediction of those 10 individual neural networks. Similarly, the gray regions show the range of one standard deviation for those predictions.

RC18: Please also note the supplement to this comment: https://bg.copernicus.org/preprints/bg-2020-262/bg-2020-262-RC2-supplement.pdf

AR18: The additional Referee comments in the supplement are very helpful. We will address these in a separate Author Response and/or implement the suggestions in the revised manuscript.

Please also note the supplement to this comment:
https://bg.copernicus.org/preprints/bg-2020-262/bg-2020-262-AC2-supplement.pdf

| Scenario | Predictors | Target | Equations Used | Source File Description | Scenario Description |
|---|---|---|---|---|---|
| 1 | Macronutrient (mol kg$^{-1}$); Micronutrient (mol kg$^{-1}$); Irradiance (W m$^{-2}$) | Biomass (mol kg$^{-1}$) | 1, 2 | Monthly Output from BLING | Nutrient distributions (predictors) from BLING were fed to Eq. 1 and 2 to calculate the biomass (target) |
| 2 | Macronutrient (mol kg$^{-1}$); Micronutrient (mol kg$^{-1}$); Irradiance (W m$^{-2}$) | Biomass (mol kg$^{-1}$) | 1, 2, 5 | Daily Output from BLING | 1) Hourly values for the predictors were interpolated using the Daily Output of BLING  1a) The macronutrient and micronutrient hourly values were calculated using a standard interpolation between the daily points.  1b) The irradiance hourly values were calculated from Eq. 5 using the value of the BLING daily input, hour of day, time of year, and location.  2) Hourly values of the predictors were fed to Eq. 1 and 2 to calculate hourly values for the biomass (target)  3) Daily-averaged values were calculated by averaging 24 hours for each location through one year  4) Weekly-averaged values were calculated by averaging 168 hour blocks of time for each location through the year  5) Monthly-averaged values were calculated by averaging the number of hours in each month (days per month * 24) for each location through the year  6) The true relationships were calculated by using the range of the hourly values for the predictors and calculating the biomass based on Eq. 1 and 2. |
| 3 | Macronutrient (mol kg$^{-1}$); Micronutrient (mol kg$^{-1}$); Irradiance (W m$^{-2}$) | Biomass (mol kg$^{-1}$) | 6, 7 (Equations within BLING used to determine the biomass) | Monthly Output from BLING | Nutrient distributions from the BLING Output were used as the predictors; Biomass from the BLING Output itself was used as the target |

**Fig. 1.** Table 1: Details for each Scenario

[Figure]

**Fig. 2.** Figure 1: Boxplots showing the variability in each of the predictor and target variables for each time-averaged dataset of Scenario 2.

**Supplement:**

Table 1: Details for each Scenario that include the predictor variables, the target variable, the equations used to calculate biomass, the type of source file used to acquire the values for the predictors, and a short description of each scenario.

| Scenario | Predictors | Target | Equations Used | Source File Description | Scenario Description |
|---|---|---|---|---|---|
| 1 | Macronutrient (mol kg$^{-1}$); Micronutrient (mol kg$^{-1}$); Irradiance (W m$^{-2}$) | Biomass (mol kg$^{-1}$) | 1, 2 | Monthly Output from BLING | Nutrient distributions (predictors) from BLING were fed to Eq. 1 and 2 to calculate the biomass (target) |
| 2 | Macronutrient (mol kg$^{-1}$); Micronutrient (mol kg$^{-1}$); Irradiance (W m$^{-2}$) | Biomass (mol kg$^{-1}$) | 1, 2, 5 | Daily Output from BLING | 1) Hourly values for the predictors were interpolated using the Daily Output of BLING
  1a) The macronutrient and micronutrient hourly values were calculated using a standard interpolation between the daily points.
  1b) The irradiance hourly values were calculated from Eq. 5 using the value of the BLING daily input, hour of day, time of year, and location.
2) Hourly values of the predictors were fed to Eq. 1 and 2 to calculate hourly values for the biomass (target)
3) Daily-averaged values were calculated by averaging 24 hours for each location through one year
4) Weekly-averaged values were calculated by averaging 168 hour blocks of time for each location through the year
5) Monthly-averaged values were calculated by averaging the number of hours in each month (days per month * 24) for each location through the year
6) The true relationships were calculated by using the range of the hourly values for the predictors and calculating the biomass based on Eq. 1 and 2. |
| 3 | Macronutrient (mol kg$^{-1}$); Micronutrient (mol kg$^{-1}$); Irradiance (W m$^{-2}$) | Biomass (mol kg$^{-1}$) | 6, 7 (Equations within BLING used to determine the biomass) | Monthly Output from BLING | Nutrient distributions from the BLING Output were used as the predictors; Biomass from the BLING Output itself was used as the target |

[Figure]

Figure 1: Boxplots showing the variability in each of the predictor and target variables for each time-averaged dataset of Scenario 2. The red line in the boxplots shows the median value, the box edges correspond to the 25th and 75th percentile values, and the whiskers correspond to the maximum and minimum.

---

## Author Comment (AC3) · 27 Sep 2020

Author responses to Luke Gregor's (Referee 2) comments in the Supplemental PDF

The Supplemental PDF attached to Referee 2's comments was a PDF of the original submitted manuscript with specific comments by Referee 2 as highlighted PDF comments.

Any minor grammatical errors (commas, periods, and other punctuation) that were noted in Referee 2's comments will be corrected in the revised manuscript. So the discussion can be focused on the comments, we have not included those grammatical errors in this Author Response. However, we do want to thank Referee 2 for finding grammatical errors that we missed.

For ease of reading, in responses addressing the specific comments we have included the referenced paragraph from the original manuscript, along with their associated line numbers in black-colored font. The text sections in the original manuscript that were highlighted by Referee 2 are in orange-colored font. Any line numbers that are referenced refer to the line numbers of the original submitted manuscript. Referee 2's comments then follow the paragraph in green-colored font and our Author Responses follow this in red-colored font.

Acronyms used in this Author Response include OMT (Original Manuscript Text), RHS (Referee Highlighted Section), Referee Comment (RC), and Author Response (AR).

OMT:

Lines 10-30

**Abstract.** Controls on phytoplankton growth are typically determined in two ways: by varying one driver of growth at a time such as nutrient or light in a controlled laboratory setting (intrinsic relationships) or by observing the emergence of relationships in the environment (apparent relationships). However, challenges remain when trying to take the intrinsic relationships found in a lab and scaling them up to the size of ecosystems (i.e., linking intrinsic relationships in the lab to apparent relationships in large ecosystems). We investigated whether machine learning (ML) techniques could help bridge this gap. ML methods have many benefits, including the ability to accurately predict outcomes in complex systems without prior knowledge. Although previous studies have found that ML can find apparent relationships, there has yet to be a systematic study that has examined when and why these apparent relationships will diverge from the underlying intrinsic relationships. To investigate this question, we created three scenarios: one where the intrinsic and apparent relationships operate on the same time and spatial scale, another model where the intrinsic and apparent relationships have different timescales but the same spatial scale, and finally one in which we apply ML to actual ESM output. Our results demonstrated that when intrinsic and apparent relationships are closely related and operate on the same spatial and temporal timescale, ML is able to extract the intrinsic relationships when only provided information about the apparent relationships. However, when the intrinsic and apparent relationships operated on different timescales (as little separation as hourly to daily), the ML methods underestimated the biomass in the intrinsic relationships. This was largely attributable to the decline in the variation of the measurements; the hourly time series had higher variability

than the daily, weekly, and monthly-averaged time series. Although the limitations found by ML were overestimated, they were able to produce more realistic shapes of the actual relationships compared to MLR. Future research may use this type of information to investigate which nutrients affect the biomass most when values of the other nutrients change. From our study, it appears that ML can extract useful information from ESM output and could likely do so for observational datasets as well.

RHS1: Abstract (Line 10)

RC1: General comment: I find that the language used in the abstract might complicate the message.

AR1: Some of the confusion appears to be in the language and terminology used in the beginning of the abstract, which includes the lines mentioned in RC2, RC3, and RC4. The revised manuscript will clarify the language in the abstract to reflect the main points of the paper more accurately. A draft version of the revised manuscript includes rewriting the first five sentences of the abstract from the original manuscript and replacing them with:

> "A key challenge for biological oceanography is relating the physiological limitations controlling phytoplankton growth to the spatial distribution of those plankton. Physiological mechanisms are often isolated by varying one driver of growth such as nutrient or light in a controlled laboratory setting producing what we call "intrinsic relationships". We contrast these with the "apparent relationships" which emerge in the environment in climatological data. Although previous studies have found that machine learning (ML) can find apparent relationships, there has yet to be a systematic study examining when and why these apparent relationships diverge from the underlying intrinsic relationships found in the lab, and how and why this may depend on the method applied."

RHS2: emergence of relationships in the environment (apparent relationships). (Line 12)

RC2: This could be much more clear and explicit - i.e. observed nutrient concentrations and light intensity.

AR2: Please see our response in AR1.

RHS3: We investigated whether machine learning (ML) techniques could help bridge this gap. (Lines 14-15)

RC3: Be much more specific here. See my general comments what the title should be.

AR3: Please see our response in AR1.

RHS4: prior knowledge. (Line 16)

RC4: prior knowledge needs to be qualified - ML uses data to predict. Perhaps the authors mean "knowledge of the system" as stated later

AR4: The revised manuscript will clarify this with the term "knowledge of the system". Please also see our response in AR1.

RHS5: apply ML to actual ESM output. (Line 21)

RC5: To do what?

AR5: The objective was to apply ML to actual ESM output as a proof-of-concept to the kinds of relationships and information one can find. Please also see our response in AR1.

RHS6: intrinsic relationships. (Line 25)

RC6: Why are you estimating intrinsic relationships? Is this predicted with the intrinsic relationship?

AR6: This sentence has been reworded. The new sentence reads:

"When the intrinsic and apparent relationships operated on different timescales (as little separation as hourly to daily), NNEs fed with apparent relationships in time-averaged data produced responses with the right shape but underestimated the biomass."

A key point here is that the intrinsic relationships are what gets coded into the models. If these are incorrect (i.e. the model gets the right answer in a given location due to compensating errors in the intrinsic relationships) it will have the wrong sensitivity to climate change.

RHS7: limitations found by ML were overestimated, (Line 27)

RC7: First mention of limitations - does this refer to limitations of growth?

AR7: The revised manuscript will make it clearer what is meant by the term "limitations". Please see our draft version in AR1.

RHS8: MLR (Line 28)

RC8: write this out in full

AR8: The revised manuscript will have this written out as "Multiple Linear Regression".

OMT:

Lines 44-56

Limitations on phytoplankton growth are usually characterized in two ways – which we term intrinsic and apparent. Intrinsic relationships are those where the effect of one driver (nutrient/light) at a time is observed, while all others are held constant (often at levels where they are not limiting). An example of such intrinsic relationships is the Michaels-Menten growth rate curves that emerge from laboratory experiments (Eppley and Thomas, 1969). Apparent relationships are those which emerge in the observed environment. An example of apparent relationships is those that emerge from satellite observations, which provide spatial distributions

of phytoplankton on timescales (say a month) much longer than the phytoplankton doubling time, which can be compared against monthly distributions of nutrients. A significant challenge that remains is determining how intrinsic relationships found in the laboratory scale up to the apparent relationships observed at the ecosystem scale (i.e., scaling the small to the large). Differences may arise between the two because apparent relationships reflect both intrinsic growth and loss rates, which are near balance over the long monthly timescales usually considered in climatological analyses. Biomass concentrations may thus not reflect growth rates. Differences may also arise because different limitation factors may not vary independently.

RHS9: Limitations on phytoplankton growth are usually characterized in two ways (Line 44)

RC9: This paragraph makes the terminology used in the abstract much clearer

AR9: The terms "intrinsic" and "apparent" relationships are terms that we define for the first time. To the best of our knowledge, these terms have not previously appeared in oceanography literature. Because we use these terms frequently throughout the manuscript, we included them in the abstract.

OMT:

Lines 58-74

Earth System Models (ESMs) have proved valuable in linking intrinsic and apparent relationships. The intrinsic relationships are programmed into ESMs as equations that are run forward in time and the output is typically provided as monthly-averaged fields. The output of these ESMs is then compared against observed fields such as chlorophyll and nutrients and can be analyzed to find apparent relationships between the two. If the ESM output is close to the observations we find in nature, we say that the ESM is performing well. However, as recently pointed out by Löptien and Dietze (2019), ESMs can trade-off biases in physical parameters with biases in biogeochemical parameters (i.e., they can arrive at the same answer for different reasons). Using two versions of the UVic 2.9 ESM, they showed that they could increase mixing (thus bringing more nutrients to the surface) while simultaneously allowing for this nutrient to be more efficiently cycled – producing similar distributions of surface properties. However, the carbon uptake and oxygen concentrations predicted by the two models diverged under climate change. Similarly, Sarmiento et al. (2004) showed that physical climate models would be expected to produce different spatial distributions of physical biomes due to differences in patterns of upwelling and downwelling, as well as the annual cycle of sea ice. These differences would then be expected to be reflected in differences in biogeochemical cycling, independent of differences in the biological models. These studies highlight the importance of constraining not just individual biogeochemical fields, but also their relationships with each other. What is less clear is: 1. Can robust relationships be found? 2. If so, what methods are most skillful in finding them? 3. How do you interpret the apparent relationships that emerge when they diverge from the intrinsic relationships we expect?

RHS10: relationships (Line 73)

RC10: between what and what?

AR10: Relationships in this sentence refers to relationships in very broad terms between biogeochemical fields and the target variable. The target variable could be phytoplankton biomass, chlorophyll concentrations, or other biogeochemical variables.

We have plans to revise this section in the revised manuscript and our current draft removes these questions from this paragraph and combines them with other sentences in the last paragraph of the introduction to more clearly highlight the main points of the manuscript.

OMT:

Lines 76-81

Recently, researchers have turned to machine learning (ML) to help in uncovering the dynamics of ESMs. ML is capable of fitting a model to a dataset without any prior knowledge of the system and without any of the biases that may come from researchers about what processes are most important. As applied to ESMs, ML has mostly been used to constrain physics parameterizations, such as longwave radiation (Belochitski et al., 2011; Chevallier et al., 1998) and atmospheric convection (Brenowitz and Bretherton, 2018; Gentine et al., 2018; Krasnopolsky et al., 2010, 2013; O'Gorman and Dwyer, 2018; Rasp et al., 2018).

RHS11: As applied to ESMs, ML has mostly been used to constrain physics parameterizations, such as longwave radiation (Belochitski et al., 2011; Chevallier et al., 1998) and atmospheric convection (Brenowitz and Bretherton, 2018; Gentine et al., 2018; Krasnopolsky et al., 2010, 2013; O'Gorman and Dwyer, 2018; Rasp et al., 2018). (Lines 78-81)

RC11: There is also now the study by Kasim et al (preprint). https://www.researchgate.net/publication/338762727_Up_to_two_billion_times_acceleration_of _scientific_simulations_with_deep_neural_architecture_search

AR11: Thank you for bringing this publication to our attention. In the revised manuscript we will conduct an additional search for the information in this paragraph of the manuscript to ensure we include recent literature.

OMT:

Lines 83-94

With regards to phytoplankton, ML has not been explicitly applied within ESMs but has been used on phytoplankton observations (Bourel et al., 2017; Flombaum et al., 2020; Kruk and Segura, 2012; Mattei et al., 2018; Olden, 2000; Rivero-Calle et al., 2015; Scardi, 1996, 2001; Scardi and Harding, 1999) and has used ESM output as input for an ML model trained on phytoplankton observations (Flombaum et al., 2020). Rivero-Calle et al. (2015) used random forest (RF) to identify the drivers of coccolithophore abundance in the North Atlantic through feature importance measures and partial dependence plots. The authors were able to find an

apparent relationship between coccolithophore abundance and environmental levels of $CO_2$, which was consistent with intrinsic relationships between coccolithophore growth rates and ambient $CO_2$ reported from 41 laboratory studies. They also found consistency between the apparent and intrinsic relationships between coccolithophores and temperature. While they were able to find links between particular apparent relationships found with the RFs and intrinsic relationships between laboratory studies, it remains unclear when and why this link breaks.

RHS12: it remains unclear when and why this link breaks. (Line 93)

RC12: The agreement between these two variables might be due to the scales of variability? Or a consistent reponse between the response of coccolithophores to temperature at a low and resolutions

AR12: Thank you for these suggestions. As we show in this paper, scales of variability can affect the link between intrinsic and apparent relationships.

OMT:

Lines 95-102

ML has been used to examine apparent relationships of phytoplankton in the environment (Flombaum et al., 2020; Rivero-Calle et al., 2015; Scardi, 1996, 2001) and it is reasonable to assume that ML could find intrinsic relationships when provided a new independent dataset from laboratory growth experiments. However, it has yet to be determined under what circumstances the apparent relationships captured by ML are no longer equal to the intrinsic relationships that actually control phytoplankton growth. In this paper, we identify two drivers of such divergence. The first is colimitation that limits the biological responses actually found in the ocean, which causes non-parametric ML methods to produce apparently non-physical results. The second is climatological averaging of the input and output variables, which can distort these relationships in the presence of non-linearity.

RHS13: identify (Line 99)

RC13: The use of "identify" here makes me think that this is a result from the study. Perhaps "propose"?

AR13: Other reviewers have also pointed to this portion of the introduction as possibly containing results from the paper. To avoid confusion, we will remove the following sentences from the introduction of the revised manuscript:

> "In this paper, we identify two drivers of such divergence. The first is colimitation that limits the biological responses actually found in the ocean, which causes non-parametric ML methods to produce apparently non-physical results. The second is climatological averaging of the input and output variables, which can distort these relationships in the presence of non-linearity."

OMT:

Lines 115-118

In the first scenario, we wanted to determine how well different ML methods could extract intrinsic relationships when only provided information on the apparent relationships and when the intrinsic and apparent relationships were operating on the same timescale. In this scenario, the apparent relationships were simply the result of multiplying the intrinsic relationships between predictors and biomass by a scaling constant.

RHS14: In this scenario, the apparent relationships were simply the result of multiplying the intrinsic relationships between predictors and biomass by a scaling constant. (Lines 117-118)

RC14: I would add that three machine learning methods are used after this sentence.

AR14: In the revised manuscript, we will introduce earlier in the methods that we are using three machine learning methods and what they are (MLR, RF, and NNE).

OMT:

Lines 148-152

The final dataset consisted of three input/predictor variables and one response term with a total of 77,328 "observations." The input variables given to each of three ML methods (Multiple Linear Regression, Random Forests, and Neural Network Ensembles, described in more detail below) were the concentrations (not the limitation terms) for the micronutrient, macronutrient, and light. The response variable was the biomass we calculated from Eq. 1 and 2.

RHS15: (Multiple Linear Regression, Random Forests, and Neural Network Ensembles, described in more detail below) (Lines 149-150)

RC15: I would introduce the use of these methods earlier. i.e. the general concept explained briefly in the opening paragraph.

AR15: Please see our response in AR14.

OMT:

Lines 154-162

The dataset was then randomly split into training and testing subsets, with 60% of the observations going to the training subset and the remainder going to the testing subset. This provided a convenient way to test the generalizability of each ML method by presenting them with "new" observations from the test subset and ensuring the models did not overfit the data. The input and output values for the training subset were then used to train a model for each ML method. Once each method was trained, we provided the trained models with the input values of

the testing subset to acquire their respective predictions. These predictions were then compared to the actual output values of the test subset. To assess model performance, we calculated the coefficient of determination ($R^2$), the mean squared error (MSE), and the root mean squared error (RMSE) between the ML predictions and the actual output values for the training and testing subsets.

RHS16: randomly split into training and testing subsets, (Line 154)

RC16: COMMENT: I'm usually not a fan of random splits (particularly in a simulated environment), as training and testing data would have very similar distributions. But since the goal here is to see if it is possible to do exactly that (can ML capture the relationships), it makes sense.

AR16: Our main purpose for splitting the data into training and testing subsets is to ensure the machine learning methods are not overfitting the data. As you (Referee 2) state in RC16, we want to ensure that the machine learning models capture the relationships.

RHS17: convenient (Line 155)

RC17: Not sure I'd use convenient here. This is standard machine learning practice.

AR17: Because it is a standard machine learning practice, the revised manuscript will replace the term "convenient" with "standard" in the referenced sentence.

RHS18: "new" observations from the test subset and ensuring the models did not overfit the data. (Line 156-157)

RC18: Would be great to see a plot of the distribution of the training and test data (box plot). as well as the spatial distribution.

AR18: Since the observations in the training and testing subsets were randomly sampled from a large dataset, we felt it was apparent that the subsets contained observations of equal magnitude and spatiotemporal distribution. This was further reinforced in the similar performances of the training/testing subsets for the machine learning methods. However, we would like to remind the Referee and other readers that source files and code are freely available on the Zenodo data repository (https://doi.org/10.5281/zenodo.3932388, Holder and Gnanadesikan, 2020).

Holder, C. D. and Gnanadesikan, A.: Linking intrinsic and apparent relationships between phytoplankton and environmental forcings using machine learning - What are the challenges?, doi:10.5281/zenodo.3932388, 2020.

OMT:

Line 195

$$\overline{L_{Irr}} = \frac{\overline{Irr}}{K_{Irr}+Irr} \neq \frac{\overline{Irr}}{K_{Irr}+\overline{Irr}} \tag{4}$$

$$\overline{L_{Irr}} = \overline{\frac{Irr}{K_{Irr}+Irr}} \neq \frac{\overline{Irr}}{K_{Irr}+\overline{Irr}} \tag{4}$$

(Line 195)

RC19: Maybe add brackets around the fraction with the overbar applied to the entire equation

AR19: To clarify that the overbar applies to the entire fraction term, the revised manuscript will change the highlighted term to include brackets.

OMT:

Lines 196-198

(Eq. 4 appears before this in the original manuscript)

where the overbar denotes a time-average, and Irr stands for irradiance (light). We wanted to investigate how such time averaging biased our estimation of the intrinsic relationships from the apparent ones; i.e., how does the link between the intrinsic and apparent relationships change with different amounts of averaging over time?

RHS20: how does the link between the intrinsic and apparent relationships change with different amounts of averaging over time? (Lines 197-198)

RC20: Having this idea in the title would make the manuscript so much clearer!

AR20: We plan on changing the title of the manuscript in the revised version. Similar to what you (Referee 2) proposed in the general comments, our new draft title currently reads as: "Can machine learning extract the mechanisms controlling phytoplankton growth from large-scale observations? – A proof of concept study".

OMT:

Lines 204-210

(Eq. 5 appear before this in the original manuscript)

where $Irr_{Int}$ is the hourly interpolated value of irradiance, $Irr_{daily}$ is the **daily-mean** value of irradiance, t is the hour of the day being interpolated, $t_{sunrise}$ is the hour of sunrise, and $T_{Day}$ is the total length of the day. The resulting curve preserves the day to day variation in the daily mean irradiance due to clouds but allows a realistic variation over the course of the day. The hourly values for the micronutrient and macronutrient were assigned using a standard interpolation between each of the daily values. These hourly interpolated values were then used to calculate the hourly biomass from Eq. 1 and 2. Note that we are not claiming the biomass itself would be

zero at night but assume that on a long enough timescale, it should approach the average of the hourly biomass.

RHS21: The hourly values for the micronutrient and macronutrient were assigned using a standard interpolation between each of the daily values. (Lines 207-208)

RC21: i.e. light is the only input/predictor that varies hourly.

AR21: The revised manuscript will include the clarification that light is the only predictor variable that varies with a daily cycle.

OMT:

Lines 221-224

As a demonstration of their capabilities, the ML methods were also applied directly to monthly averaged output from the BLING model itself using the same predictors in Scenarios 1 and 2, but using the biomass calculated from the actual BLING model. As described in Galbraith et al. (2010), BLING is a biogeochemical model where biomass is diagnosed as a non-linear function of the growth rate smoothed in time. The growth rates, in turn, have the form (continues on to Eq. 6 in the original manuscript)

RHS22: biomass is diagnosed as a non-linear function of the growth rate smoothed in time. (Lines 223-224)

RC22: Aha! This gives ligitimacy to the assumption made in EQ1 and Scenario 2, where hourly data is averaged to give biomass. I use bring this as a justification for the assumption.

AR22: The reviewer rightly points out that this assumption could have been pointed out earlier, and that this might have motivated the manuscript better. We now say at line 40 that:

> "As we will show, under certain formulations of ecosystem dynamics the phytoplankton biomass has a direct relationship to this growth rate."

OMT:

Line 224

$$B = \left( \frac{\tilde{\mu}}{\lambda} + \frac{\tilde{\mu}^s}{\lambda^s} \right) S_* \tag{7}$$

RHS23:

$$B = \left( \frac{\tilde{\mu}}{\lambda} + \frac{\tilde{\mu}^s}{\lambda^s} \right) S_* \tag{7}$$

(Line 234)

RC23: What is superscript a?

AR23: That is actually a "3," not a lowercase a. The µ and λ in the fraction term after the "+" are both cubed. We will make the fonts in the equation larger in the revised manuscript.

OMT:

Lines 270-273

It should be noted that we are not trying to suggest that MLR is always ineffective for studying ecological systems. MLR is a very useful and informative approach for studying linear relationships within marine ecological systems (Chase et al., 2007; Harding et al., 2015; Kruk et al., 2011). However, we highly encourage our readers to try ML as it can provide insight into the non-linear portions of a dataset.

RHS24: It should be noted that we are not trying to suggest that MLR is always ineffective for studying ecological systems. MLR is a very useful and informative approach for studying linear relationships within marine ecological systems (Chase et al., 2007; Harding et al., 2015; Kruk et al., 2011). However, we highly encourage our readers to try ML as it can provide insight into the non-linear portions of a dataset.

RC24: Good! I was going to comment on this if it was not mentioned in the text.

AR24: Yes, we wanted to make it clear that we were not invalidating multiple linear regression. It is a very useful method!

OMT:

Lines 276-285

RFs are an ensemble ML method utilizing a large number of decision trees to turn "weak learners" into a single "strong learner" by averaging multiple outputs (Breiman, 2001). In general, RFs work by sampling (with replacement) about two-thirds of a dataset and constructing a decision tree. At each split, the random forest takes a random subset of the predictors and examines which variable can be used to split a given set of points into two maximally distinct groups. This use of random predictor subsets helps to ensure the model is not overfitting the data. The process of splitting the data is repeated until an optimal tree is constructed or until the stopping criteria are met, such as a set number of observations in every branch (then called a leaf / final node). The process of constructing a tree is then repeated a specified number of times, which results in a group (i.e., "forest") of decision trees. Random forests can also be used to construct regression trees in which a new set of observations traverse each decision tree with its associated predictor values and the result from each tree is aggregated into an averaged value.

RHS25: two-thirds of a dataset and constructing a decision tree. (Line 278)

RC25: might be good to add that this is commonly referred to as bootstrap aggregation, or bagging in the machine learning world

AR25: We will include this term in the revised manuscript.

RHS26: Random forests can also be used to construct regression trees in which a new set of observations traverse each decision tree with its associated predictor values and the result from each tree is aggregated into an averaged value. (Lines 283-285)

RC26: This sentence is not completely clear. Are you trying to say that the predicted value is the average of all tree's prediction values.

AR26: Yes, that is what we are trying to say. When using random forest for regression (instead of classification), the predicted value is the average of all the individual trees' predictions.

OMT:

Lines 287-293

Here, we used the same parameters for RF in the three scenarios to allow for a direct comparison between the scenarios and to minimize the possible avenues for errors. Each RF scenario was implemented using the TreeBagger function in MATLAB 2019b, where 500 decision trees were constructed with each terminal node resulting in a minimum of five observations per node. An optimization was performed to decide the number of decision trees that minimized the error while still having a relatively short runtime of only several minutes. For reproducible results, the random number generator was set to "twister" with an integer of "123". Any remaining options were left to their default values in the TreeBagger function.

RHS27: minimum of five observations per node. (Line 290)

RC27: How did you decide on 5? This might result in overfitting given the nature and size of the training data set. Is there some sort of hyper-parameter selection process?

AR27: Five observations per node was the default number in the Matlab function used to construct the random forests. This means that at the end of each leaf of a decision tree within the random forest, five observations were being averaged for that single leaf. However, we still would not expect random forests to overfit the data. By construction, random forests generally do not overfit datasets for a couple reasons:

1. The random way in which data is selected to build individual decision trees.
   a. When any one decision tree is being constructed, the data is randomly selected with replacement from the available data until it reaches the same number of observations in the dataset. In general, this type of random sampling means that about 2/3 of the observations in a dataset will be captured by this type of sampling. This is then used to construct a decision tree. The process is repeated until the specified number of trees is reached. In our case, this would mean that our training subset is being randomly sampled for the construction of the decision

trees. This type of decision tree construction for random forests also means that no decision tree will be trained with every sample, which further decreases the likelihood of overfitting.

2. The random way in which the predictors are used in the construction of individual decision trees.

   a. When the decision trees are being constructed, only some of the predictors are available each time a split is determined. These predictors are randomly selected. Given those predictors, an error metric is used to determine the best split that will minimize the error. This random way in which predictors are selected decreases the chances of overfitting the data.

Additionally, we only allowed the random forest to be trained on the training subset. The other 40% of the data was in the testing subset and had never been "seen" by the random forests. Since the performance metrics of the training and testing subsets for random forest were very similar, this suggested to us that the relationships had been captured by the random forest in the training subset and those same relationships were present in the testing subset.

OMT:

Lines 304-310

Feed-forward NNs consist of nodes connected by synapses (or weights) and biases with one input layer, (usually) at least one hidden layer, and one output layer. The nodes of the input layer correspond to the input values of the predictor variables, and the hidden and output layer nodes each contain an "activation function." Each node from one layer is connected to all other nodes before and after it. The values from the input layer are transformed by the weights and biases connecting the input layer to the hidden layer, put through the activation function of the hidden layer, modified by the weights and biases connecting the hidden layer to the output layer, and finally entered into the final activation function of the output node.

RHS28: synapses (Line 304)

RC28: I would stick with weights rather than synapses. There has been a move away from comparing NNs with the brain as this gives far too much credit to the capabilities of NNs

AR28: The revised manuscript will use the term "weights" in place of "synapses".

RHS29: Each node from one layer is connected to all other nodes before and after it. (Line 306-307)

RC29: By design FFNNs are fully connected, meaning that each node from one layer is connected to all other nodes in preceding and succeding layers

AR29: Yes, that is correct. We included this line so that readers who may not be experienced in machine learning or neural networks understood the details of FFNNs.

OMT:

Lines 322-330

To minimize the differences between scenarios, we used the same framework for the NNs in each scenario. Each NN consisted of three input nodes (one for each of the predictor variables), 25 nodes in the hidden layer, and one output node. The activation function within the hidden nodes was a hyperbolic tangent sigmoid function and the activation function within the output node used a linear function. The stopping criteria for each NN was set as a validation check such that the training stopped when the error between the predictions and observations increased for six consecutive epochs. An optimization was performed to decide the number of nodes in the hidden layer that 11 minimized the error while maintaining a short training time. Additionally, sensitivity analyses were performed using different activation functions to ensure the choice of activation function had minimal effect on the outcome and apparent relationships found by the NNEs.

RHS30: validation check (Line 326)

RC30: What portion of the data was used for validation?

AR30: We used the default values in the Matlab function for training feedforward neural networks. These default values are 70% in the training set, 15% in the validation set, and 15% in the testing set. The function used in Matlab randomly partitions the data into these categories.

It should be noted that those partition values were only applied to our training subset, not the testing subset. The Matlab function used to train the NNs partitions our training subset into its own training, validation, and test sets. For example, this means that out of the 46,397 observations in our training subset of Scenario 1, 32,477 observations (70%) went to the training set of the NN function, 6,960 observations (15%) went to the validation set of the NN function, and 6,960 observations (15%) went to the test set of the NN function.

RHS31: An optimization was performed to decide the number of nodes in the hidden layer that (Line 327)

RC31: What kind of optimisation? Grid search? What ranges of nodes were explored?

AR31: Please see our responses to Referee 1 (AR8) for these details.

RHS32: ensure the choice of activation function had minimal effect on the outcome and apparent relationships found by the NNEs. (Lines 329-330)

RC32: I would phrase this differently. Perhaps the activation function does make a difference. There are tens of activation functions (https://stats.stackexchange.com/questions/115258/comprehensive-list-of-activation-functions-in-neural-networks-with-pros-cons). It would be more accurate to list the activation functions tested and state that it did not make a difference within these options.

AR32: The revised manuscript will list the activation functions that we tested.

OMT:

Lines 332-335

Each NNE scenario used the feedforwardnet function in MATLAB 2019b. Any options not previously specified remained at their default values in the feedforwardnet function. The NNEs contained ten individual NNs for each scenario. For reproducibility, the random number generator was set to "twister," and the random number seed was set to the respective number of its NN (i.e., 1, 2, 3, up to 10).

RHS33: For reproducibility, the random number generator was set to "twister," and the random number seed was set to the respective number of its NN (i.e., 1, 2, 3, up to 10). (Lines 334-335)

RC33: Details like this can be either in supplementary material and/or in the code.

AR33: Since this information is already in the code, we will remove this sentence from the revised manuscript.

OMT:

Lines 337-341

Each variable was scaled between -1 and 1 based on its respective maximum and minimum. This step ensures that no values are too close to the limits of the hyperbolic tangent sigmoid activation function, which would significantly increase the training time of each NN. These scalings were also applied to the RF and MLR methods for consistency between methods and the scaling did not affect the results of either method (results not shown). The results presented in this paper were then transformed back to their original scales to avoid confusion from scaling.

RHS34: scaled between -1 and 1 based on its respective maximum and minimum. (Line 337)

RC34: I'm interested to know why data wasn't scaled with MEAN and STDEV rather? This approach is usually a bit more robust to outliers. Perhaps this is not a problem with model data? i.e. model averages will not have outliers?

AR34: Please see our response in AR35.

RHS35: This step ensures that no values are too close to the limits of the hyperbolic tangent sigmoid activation, (Line 337-338)

RC35: Scaling the data ensures that the gradient of each variable has the same "steepness". https://stats.stackexchange.com/questions/322822/how-normalizing-helps-to-increase-the-speed-of-the-learning

AR35: Yes, it makes sure they have the same "steepness," but it also ensures that the output of the activation function we are using (tangent sigmoid; tanh) are concentrated in a narrow range.

For example, if the input to the tanh activation function is between -1 and 1 (inside the red bars of Fig. 1), the range of the output is between about -0.76 and 0.76. In contrast, if the input is outside the range of -1 to 1 (outside the red bars of Fig. 1), the output quickly approaches the extremes of -1 and 1 on the y-axis. If the outputs are toward the extreme ends, this can cause the NNs to get "stuck" in those extremes during training which affects how much the weights can be adjusted during each epoch (ie. more epochs are needed for training which leads to longer training times).

[Figure]

**Figure 1:** Tangent sigmoid activation function.

RHS36: These scalings were also applied to the RF and MLR methods for consistency between methods and the scaling did not affect the results of either method (results not shown). (Lines 339-340)

RC36: Rephrase: Scaling of the input variables is not necessary for RF and MLR, but was still applied for each of the methods for the sake of consistency with NNE.

AR36: The revised manuscript will rephrase the highlighted sentence.

RHS37: paper were then transformed back to their original scales to avoid confusion from scaling. (Line 341)

RC37: Scaling the output/target variable is not usually done. Though this would make no difference. I think you can thus leave this sentence out.

AR37: We received questions in past submissions when we did not specify that the values were transformed back to their original values. For clarity, we would like to keep this sentence (or something similar) in the revised manuscript.

OMT:

Lines 345-349

In Scenario 1, the RF and NNE both outperformed the MLR as demonstrated by higher $R_2$ values, lower MSE, and lower RMSE (Table 1). The decreased performance of the MLR is not inherently surprising, given the non-linearity of the underlying model, but it does demonstrate that the range of nutrients and light produced as inputs by ESM2Mc is capable of producing a non-linear response. Additionally, each method showed similar performances between the training and testing subsets suggesting adequate capture of the model dynamics in both subsets.

RHS38: Additionally, each method showed similar performances between the training and testing subsets suggesting adequate capture of the model dynamics in both subsets. (Lines 348-349)

RC38: I have a feeling that this might be due to the random shuffling of testing and training data; i.e. the training dataset is almost perfectly representative of the test dataset

AR38: It was our intention that the training and testing subsets be representative of one another and of the complete dataset. The highlighted sentence provides support to that intention.

OMT:

Lines 367-373

When we computed an "effective" half-saturation for the nutrient curves in the top row of Fig. 2, we got values for $K_N$ that were far lower than the actual ones specified in the model (Table 4). The "effective" half-saturation of when other predictors are held at their $25_{th}$ percentile for the micro- and macronutrient were underestimated by one and two orders of magnitude, respectively. It was only at the higher percentiles that the micronutrient "effective" half-saturation was adequately captured when the macronutrient was not limiting. Furthermore, the "effective" half-saturation of the macronutrient was not captured even when the other variables were held at their $75_{th}$ percentiles because the $75_{th}$ percentile of the micronutrient still limited growth.

RHS39: "effective" (Line 367)

RC39: why is this in quotations - maybe pseudo/quasi is a better word here?

AR39: These quotations will be removed in the revised manuscript. Additionally, we will remove other unnecessary quotation marks in the revised manuscript.

OMT:

Lines 424-436

Despite the fact that it agreed well with the observations, the RF prediction deviated from the true response to a given variable when other variables are held at higher percentiles (Fig. 2). This can likely be explained by the range of the training subset and how RFs acquire their predictions. When presented with predictor information, RFs rely on the information contained within their training data. If they are presented with predictor information that goes outside the range of the dataspace of the training set, RFs will provide a prediction based on the range of the training set. When performing the sensitivity analysis, the values of the predictors in the higher percentiles were probably outside the range of the training subset. For example, the bottom left plot of Fig. 2 shows how RF deviates from the true response as the concentration of the macronutrient increases – actually decreasing as nutrient increases despite the fact that such a result is not programmed into the underlying model. Although there may be observations in the training subset where the light and micronutrient are at their $75_{th}$ percentile values when the macronutrient is low, there likely are not any observations where high levels of the macronutrient, micronutrient, and light are co-occurring. Without any observations meeting that criteria, the RF provided the highest prediction it could based on the training information. We discuss this point in more detail below.

RHS40: it (Line 424)

RC40: It would be better to put "the RF prediction" first and then "it"

AR40: The revised manuscript will restructure this sentence to be: "Despite the fact that the RF prediction agreed well with the observations, it deviated from the true response to a given variable when other variables were held at higher percentiles."

OMT:

Lines 462-474

When comparing the apparent relationships of the time-averaged datasets with those of the hourly intrinsic relationships, the methods almost always underestimated the true response to light and nutrient (Fig. 3 and 4). This result is not entirely unexpected. The averaging of the hourly values into daily, weekly, and monthly timescales quickly leads to a loss of variability, especially for light (Fig. 5). In fact, the variability was lost in the daily time averaging with the longer timescales showing only small differences in the possible range of values (Fig. 5). The loss of variability means that the light limitation computed from the averaged light is systematically higher than the averaged light limitation. To match the observed biomass, the asymptotic biomass at high light has to be systematically lower (see Appendix A for the mathematical proof). Differences were much smaller for nutrients as they varied much less over the course of a month in our dataset. Our results emphasize that when comparing apparent

relationships in the environment to intrinsic relationships from the laboratory, it is essential to take into account which timescales of variability averaging has removed. Insofar as most variability is at hourly time scales, daily-, weekly-, and monthly-averaged data will produce very similar apparent relationships (Fig. 4). But if there was a strong week-to-week variability in some predictor, this may not be the case.

RHS41: has (Line 472)

RC41: was / has been

AR41: This correction will be made in the revised manuscript.

OMT:

Lines 495-501

The large increases in biomass in the micronutrient plots and hindrance of biomass in the light and macronutrient plots suggest that the system is limited by the concentration of micronutrient (Fig. 7). The biomass remained low even when macronutrient and light were at favorable levels because even when at the $75_{th}$ percentile value, the micronutrient was still limiting (Fig. 8). Conceptually this makes sense since the micronutrient limitation in the BLING model hinders growth, but also limits the efficiency of light-harvesting (Galbraith et al., 2010). Additionally, the computation of the "effective" half-saturation constants demonstrates that the half-saturation constant for light drops sharply as nutrients drop (Table 4).

RHS42: computation of the "effective" (Line 500)

RC42: Could be replaced by "estimated"

AR42: Yes, "estimated" seems to be the word more in line with our intention. The term "effective" will be replaced in the revised manuscript.

OMT:

Lines 504-507

Our main objective in this manuscript was to use ML to determine under what conditions intrinsic and apparent relationships between phytoplankton are no longer equal, to identify whether such divergence depends on the ML method or how the input data is handled, and to understand how such divergence is related to underlying biological dynamics.

RHS43: how the input data is handled, (Line 506)

RC43: Be more specific here, bring in the time aspect.

AR43: The revised manuscript will include more specific information, such as temporal averaging.

OMT:

Lines 538-548

Both RFs and NNEs performed well when the predictions they were asked to make were within the range of the training data. However, the sensitivity analyses illustrated the impact of RFs inability to extrapolate outside that range and that RF's suggested systematic decreases in biomass at high values of a limiting variable. Nonetheless, RFs were able to capture the same relationships as the NNEs when the sensitivity analysis was querying environments within the range of the training data. It seems that as long as RFs are presented with information across the range of the dataset, RFs will perform just as well as NNEs in a sensitivity analysis. This strengthens the conclusions of Rivero-Calle et al. (2015) in that physiologically reasonable relationships between forcing variables and biomass found using RF are reliable so long as the forcing variables (in this case $pCO_2$ and temperature) vary over their entire range independently of other variables (nutrients and light). However, when variation in $pCO_2$ is related to variation in nutrients and light (i.e., in the seasonal climatology where $pCO_2$ is high in the winter, light is low, and nutrients are high) RFs are unable to extract a clear signal of $pCO_2$ limitation.

RHS44: This strengthens the conclusions of Rivero-Calle et al. (2015) in that physiologically reasonable relationships between forcing variables and biomass found using RF are reliable so long as the forcing variables (in this case pCO2 and temperature) vary over their entire range independently of other variables (nutrients and light). However, when variation in pCO2 is related to variation in nutrients and light (i.e., in the seasonal climatology where pCO2 is high in the winter, light is low, and nutrients are high) RFs are unable to extract a clear signal of pCO2 limitation. (Lines 543-548)

RC44: This is new literature in the conclusion. I would try to bring this into the discussion rather.

AR44: We will move the highlighted sentences to the discussion in the revised manuscript.

OMT:

Lines 562-573

ML techniques have several benefits that could make them useful for biological oceanographers and ecosystem modelers. Many ML methods (including the two presented here) do not require any prior knowledge of a system to construct a model. Additionally, new methods are continually being developed for viewing the dynamics of the ML models. Given these advantages, ML could provide a compact form for representing relationships between ecosystem parameters such as biomass and primary productivity and their environmental drivers (nutrients and light) in observational data and complex models. Preliminary work indicates that we can use NNEs in particular to: 1. Compare model relationships with those derived from observational datasets, rather than simply using spatial patterns of errors. 2. Evaluate whether differences between models reflect important differences in biological parameters or whether they are due to

differences in the physical circulation. We would expect that two different physical models run with the same biological scheme would produce the same relationships. 3. Evaluating whether global warming really would be expected to drive ecosystems outside their historical parameter range. We will report on these results in a future manuscript.

RHS45: dynamics (Line 564)

RC45: What do you mean by dynamics? Do you mean feature importances as an example?

AR45: In this sentence, we were referring broadly to new ways of viewing the relationships that are found by machine learning methods.

RHS46: compact form (Line 565)

RC46: unclear what is meant by this?

AR46: In this sentence, we were trying to state that machine learning could provide a way of comparing relationships found in observational datasets to relationships found in output of Earth System Models.

RHS47: relationships (Line 568)

RC47: relationships of what?

AR47: In this sentence, we were referring broadly to biogeochemical relationships in Earth System Models and observational datasets.

RHS48: Evaluate whether differences between models reflect important differences in biological parameters or whether they are due to differences in the physical circulation. (Lines 569-570)

RC48: Also, as pointed out by the author in the introduction, it may also differentiate whether models are similar for the right reason.

Would one not have to be careful of the implementation of the "intrinsic" equations and half-saturation constants of the model?

AR48: Yes, one would need to be careful of the implementation of the intrinsic equations and half-saturation constants. We are currently working on a manuscript that discusses such questions.

OMT:

Line 758

Table 3: Scenario 3 comparison of MLR, RF, and NNE method performance for the training and testing sets.

RHS49: Scenario 3 comparison of MLR, RF, and NNE method performance for the training and testing sets. (Line 758)

RC49: Be specific about what the target variable is (biomass). Further, there are no units

AR49: The revised manuscript will include the target biomass variable, along with its units, in the captions for the Tables.

OMT:

Line 763

(References Table 4 in the original manuscript)

RHS50: -2.11 x $10^4$ (This is the value in Table 4 associated with the 25$^{th}$ percentile of the Micronutrient for Scenario 3; Line 763)

RC50: Typo?

AR50: That is not a typo. One result we found was limitations in the variables can affect the estimate of the half-saturation. The issue is that there are not any observations where the macronutrient is at the 25$^{th}$ percentile or below when the micronutrient is limiting. The micronutrient is more or less uncorrelated with biology in this range and any half-saturations derived from it will be poor estimates. This clarification will be made in the revised manuscript.

RHS51: 1.85 (This is the value in Table 4 associated with the 25$^{th}$ percentile of the Light variable for Scenario 3; Line 763)

RC51: Should these light half-saturation estimates be the same as those above? If so, why is this so far off? If not, then make this clear!

AR51: In BLING, we use the Geider et al. model for light limitation. In this model, the chlorophyll to carbon ratio $\theta$ adjusts with the light; it becomes lower as light gets higher and higher as light gets lower. Since $Irr_k \propto \frac{Lim_{nut}}{\theta}$, this means that the ratio $Irr/Irr_k$ ends up being a lot more constant than one might expect – essentially it only drops to zero when the plankton cannot make any more chlorophyll. Please note that in Scenarios 1 and 2 we assumed that $Irr_k$ was independent of $Irr$. At very low values when nutrients are highly limiting, $Irr_k$ is very small; while at higher values of nutrients, it is larger.

OMT:

Lines 791-792

Figure 6: Scatter plots from the BLING model (a: surface biomass vs. temperature-normalized growth rate; b: mean nutrient limitation vs. monthly-averaged nutrients; c: mean light limitation vs. monthly-averaged Irr, $Irr_k$).

RHS52: Figure 6: Scatter plots from the BLING model (a: surface biomass vs. temperature-normalized growth rate; b: mean nutrient limitation vs. monthly-averaged nutrients; c: mean light limitation vs. monthly-averaged Irr, $Irr_k$). (Lines 791-792)

RC52: should the dependent (predicted) variables not be on the y-axes?

AR52: As it is currently constructed, the horizontal axis represents an "input" constructed from monthly mean variables (light, nutrient), while the vertical axis represents a target computed by the model. We will clarify this in the revised manuscript.

OMT:

Lines 795-798

Figure 7: Sensitivity analysis for Scenario 3 with the columns corresponding to the predictors and the rows corresponding with the percentile value at which the other predictors were set. The gray circles show the observations from the BLING model and the dashed lines show the predicted apparent relationships for each method.

RHS53: Figure 7: Sensitivity analysis for Scenario 3 with the columns corresponding to the predictors and the rows corresponding with the percentile value at which the other predictors were set. The gray circles show the observations from the BLING model and the dashed lines show the predicted apparent relationships for each method.

RC53: I have two comments here:

> 1) It would be more useful to have a 2D histogram in this data representation. Represent this data as a 2D contour where the colormap is scaled logarithmically. Using a grey colormap will then allow you to still plot the dashed lines in colour.

> 2) What are the actual curves for the intrinsic relationship?

AR53: Addressing comment 1: Yes, that is a good suggestion. We will try that implementation to see if it improves the layout of the plots.

Addressing comment 2: Since Scenario 3 was being used as a proof-of-concept, we did not include the intrinsic relationships. The reason for this was we were attempting to demonstrate the type of information one could gain from having access to Earth System Model output only, but not necessarily access to the computational resources to run the Earth System Model code itself.

RHS54: (The Referee comment is posted next to the Legend in Figure 7; Line 794)

RC54: Are the different models necessary here? If RF is susceptible to "missing data" why should it be used in an even more complex scenario? The same applies for MLR - it is clearly not complex enough to capture this signal.

AR54: The revised manuscript will only include sensitivity curves for the NNE in this figure.

RHS55: (The Referee comment highlights the label "75th percentile" in Figure 7; Line 794)

RC55: Is it still useful to show the different percentiles at this point? Is this point not proven in experiment 2.

AR55: The different percentiles still affect the result. For example, in Figure 4 the predicted biomass values increase with higher percentiles. The values on the y-axis change between the subplots.

RHS56: (The Referee comment highlights the "P" in the units for Biomass in Figure 7; Line 794)

RC56: What is P?

AR56: Here, P stands for phosphorus. This is analogous to the macronutrient term.

OMT:

Lines 801-802

Figure 8: A 3-D scatter plot showing the concentrations from Scenario 3 for the macronutrient, micronutrient, and light with the color of the data points corresponding to the biomass concentrations.

RHS57: Figure 8: A 3-D scatter plot showing the concentrations from Scenario 3 for the macronutrient, micronutrient, and light with the color of the data points corresponding to the biomass concentrations. (Lines 801-802)

RC57: I don't think this is a very informative plot. It is difficult to see what is really going on with the majority of the data being obscured by other data. In addition to this, it is quite strange to have the x-axis (macro) going from large to small, rather than small to large

AR57: The main purpose of this figure was to show that some of the relationships found by the sensitivity analysis were reflected in the data. Specifically, that increases in the micronutrient led to large increases in biomass.

It was necessary to plot it with the macronutrient going from large to small, otherwise the area we were wanting to focus on would be obscured by the other data.

We will consider removing this plot in the revised manuscript.

---

## Author Comment (AC4) · 27 Sep 2020

In our previous Author Responses to Referee 1's comments, we incorrectly stated that we would replace the term "testing subset(s)" with the term "validation subset(s)" in the revised manuscript. For reference, please see AR6 of our Author Responses to Referee 1's comments.

As we understand it, validation sets are used to tune the parameters of machine learning models and testing sets are used to assess the performance of the trained machine

learning models. Since we use our test subset to assess the performance of our trained machine learning models, we should keep the term "testing subset(s)". In the revised manuscript, we will keep the term "testing subset(s)," instead of replacing it with the term "validation subset(s)".

We apologize for this error in our initial Author Response.

---

## Author Response (AR1)

Dear Editor,

We would like to thank the reviewers for their constructive feedback. We made sure to take their comments into account as we were revising the manuscript.

The changes we have made to the revised manuscript were rather extensive, but some of the major changes that were made included:

- Clearer descriptions of each Scenario, including information on the predictor and target variables, the equations that were used, and details on the source files. This information can be found in Table 1 of the revised manuscript.
- Details on how we decided on particular settings for the machine learning techniques.
- A method for visualizing how relationships changed across different timescales, allowing one to see how intrinsic and apparent relationships are linked
- Demonstration of the unreliability of estimating half-saturation coefficients from the sensitivity analysis curves as a quantitative measure of machine learning accuracy
- A method for visualizing how predictor variables interact and how this interaction affects the target variable
- Combined the results and discussion sections into a single section
- Rewrote the conclusion section to reflect the updated information in the revised manuscript

It was necessary to combine the results and discussion into a single section because information that was learned in one scenario was then applied to the next scenario. For ease of reading and the flow of the paper, it made sense to interpret the results and lessons learned from one scenario so they could be put into context in the next.

We realize that the updated author responses are rather long, but we wanted to ensure that we provided detailed answers for all the questions and comments. Additionally, part of the reason for the length of these updated author responses was to address the supplemental comments for Reviewer 2, which provided helpful individual highlighted comments in a PDF of the original manuscript. For reference, we included the entire paragraph from the original manuscript for any section of the manuscript that had a comment or suggestion in the Supplemental PDF.

We sincerely appreciate your time and consideration of our manuscript.

Thank you,

Christopher Holder

**Updated Author Responses to Comments from Anonymous Referee 1**

In our previous Author Comments, we occasionally stated phrases such as "(Some change) will be included in the revised manuscript." Where applicable, we have updated our Author Responses below and now reference the actual revised manuscript.

In the following responses RC stands for Referee Comment and AR stands for Author Response. Referee Comments are in black-colored font and Author Responses are in red-colored font. Unless specified otherwise, references to figure/table numbers and line numbers refer to those of the original manuscript.

RC0: The authors try to find a ML method that can establish and help to explain the link between intrinsic and apparent relationships of phytoplankton and environmental forcing. Three different methods were tested: Multiple Linear Regression (MLR), Random Forests (RFs) and Neural Network Ensembles (NNEs). The tests were provided on three different Scenarios. The authors found that the NNEs reproduce well the observed biomass (biomass estimated from intrinsic relationships) based on the knowledge of only apparent relationships when both relationships operate on the same spatial and temporal timescales. All methods fail when the intrinsic and apparent relationships operated on different timescales. However, the main authors' conclusion is that ML methods still can give an information on shapes of intrinsic relationships. Using the Earth System Models (ESM) in their third Scenario they show that the ML methods can extract useful information from this model that can used for an examination of interaction between input variables.

The article arises an interesting subject. However, it misses a clear explanation how the trained data and data for validation were constructed; the explanation of the role of input data, especially the physical meaning of the choice to fix them at 25th, 50th and 75th percentile for a sensitive analysis. I also think that the authors did not use all possible ML capacities for example, test more hidden layers in NNs, or add few more input variables, environmental ones, maybe a temperature.

Also, the main results that the NNEs can reproduce the general shape of intrinsic relationships does not have a systematic character: it is clearly seen that in some cases the NNEs shapes reproduce a different behavior compare to observation data. If the authors want to keep this conclusion, they will have to provide an additional analysis of conditions under which their conclusion is persistent.

The article in its current state needs a serious major correction.

AR0: We want to thank Anonymous Referee #1. We have found the comments and suggestions they provided to be very helpful in restructuring this manuscript.

RC1: It is hard to understand reading the introduction what is the main aim of the article. I have found three:

AR1: We understand how some of these statements may have caused confusion as to the main purpose of the paper. We address these in the following author responses AR1.1, AR1.2, AR1.3, and AR1.4.

RC1.1: "A significant challenge that remains is determining how intrinsic relationships found in the laboratory scale up to the apparent relationships observed at the ecosystem scale (i.e., scaling the small to the large)."

AR1.1: It was our intent that this served as more of a big-picture gap that has yet to be filled/solved. With this serving as the big-picture statement, it serves as a leadup to rest of the introduction. In this paper, we are not necessarily trying to answer the entirety of this big-picture statement; rather, we are attempting to answer part of it by examining some smaller aspects of it.

RC1.2: "What is less clear is: 1. Can robust relationships be found? 2. If so, what methods are most skillful in finding them? 3. How do you interpret the apparent relationships that emerge when they diverge from the intrinsic relationships we expect?"

AR1.2: This statement was intended to serve as a link between the two paragraphs. However, we can see the confusion, especially given that the questions are numbered which would typically signify importance. We have removed this statement in its original paragraph and have revised it to be two main questions near the end of the introduction. The modified structure of the revised manuscript is included below in AR1.4.

RC1.3: "To investigate when and why the link between intrinsic and apparent relationships break. . ."

AR1.3: This is now the main purpose of the manuscript. We attempt to investigate this by answering two main questions. These questions and the modified structure of the manuscript are included below in AR1.4.

RC1.4: If the two first citations can be linked to the title "What are the challenges?", the last one, what the authors actually did, is not according to the title "What are the challenges?". I will advise to authors modify the structure of the introduction and emphasize what is    the main idea of this article.

AR1.4: We have changed the title of the manuscript to be more reflective of the main points. The revised title is "Can machine learning extract the mechanisms controlling phytoplankton growth from large-scale observations? – A proof of concept study"

The section of the introduction that more clearly defines the main points of the paper is included below for reference:

> To investigate when and why the link between intrinsic and apparent relationships break, we try to answer two main questions in this paper:
>
> 1. Can ML techniques find the correct underlying intrinsic relationships and, if so, what methods are most skillful in finding them?
> 2. How do you interpret the apparent relationships that emerge when they diverge from the intrinsic relationships we expect?

In addressing the first question, we first needed to demonstrate that we had a ML method that would correctly extract intrinsic relationships from apparent relationships. We constructed a simple model in which the biomass is directly proportional to the time-smoothed growth rate. In this scenario, intrinsic and apparent relationships operated on the same time and spatial scale and were only separated by a scaling factor, but the environmental drivers of phytoplankton growth had realistic inter-relationships. Having a better handle on the results from the first question, we were able to move onto the second question where we looked at where the link between intrinsic and apparent relationships diverged. We modified the first scenario so that the apparent relationships use a time-averaged input (similar to what would be used in observations), but the intrinsic relationships operate by smoothing growth rates derived from hourly input. Finally, we conduct a proof-of-concept study with real output from the ESM used to generate the inputs for scenarios 1 and 2, in which the biomass is a nonlinear function of the time-smoothed growth rate.

RC2: The conclusion also has to be rewritten in clear way:  what are the challenges authors have found to link intrinsic and apparent relationships? The authors said: "Our main objective in this manuscript was to use ML to determine under what conditions intrinsic and apparent relationships between phytoplankton are no longer equal. . ." This objective was not clear in the introduction, and again, does not correspond well to the title of the article.

AR2: We have mostly rewritten the entire Conclusions section and feel that the new Conclusions section better reflects the main objectives and points of the paper.

RC3: I advise you also to avoid the non-explained abbreviations in the abstract, like line 21: "ESM", line 28: "MLR".

AR3: We have removed the non-explained abbreviations from the abstract. They have been replaced with their unabbreviated definitions.

RC4: It would be better not mentioning the results in the introduction (lines 100-103), and instead prepare your readers for the structure of the article.

AR4: We agree with this comment. The lines mentioning the results in the introduction have been removed.

RC5: In the section "2. Methods" I suggest you provide a scheme or formula what exactly you were using as input/predictors and output/target and how it links to your equations 1 and 2 etc. This will simplify the understanding. It is especially important for the description of the Scenario 2. It is unclear, are the values of biomass with which the authors compare their results calculated based on hourly values? Is the biomass for target in learning algorithm smoothed or calculated based on smoothed predictors? It will strongly affect the results.

AR5: A new table has been included in the revised manuscript (Table 1 in the revised manuscript). It includes the following details for each scenario: the predictor variables,

the target variable, the equations used in each scenario, a description of the source file, and a description of the scenario, such as how the target biomass was calculated.

RC6: The word "target" was not used in your article. It is common word in the ML domain and can also help to better understand the method especially for readers who only start to use ML techniques. Also, I would suggest to use the word "validation" instead of "testing".

AR6: We have replaced the term "response variable(s)" with the term "target variable(s)".

Additionally, as we understand it, validation sets are used to tune the parameters of machine learning models and testing sets are used to assess the performance of the trained machine learning models. Since we use our test subset to assess the performance of our trained machine learning models, we chose to keep the term "testing subset(s)".

RC7: To perform a sensitive analysis the authors fixed two of three predictors at different percentiles. It misses the explanation why 25th, 50th and 75th percentiles were chosen, what is the physical meaning of this choice and how it influences the results? For example, does 75th percentiles represent the extremes and what does it mean? I think that it is important to explain it to better understand the results. Please, clarify that the sensitivity analysis was done already on trained ML model.

AR7: The main reason for choosing these particular percentile values was so that we could examine conditions in a domain space that may be found as an actual observation. Additionally, we wanted to avoid extreme percentiles (1$^{st}$ and 99$^{th}$) because the standard deviation in the predictions of the trained ML models is very large at these extremes.

We have also clarified that the sensitivity analysis was performed on the trained ML models.

The revised version of the manuscript includes these changes, along with more details about how the sensitivity analysis was performed and explains the reasons for why we chose particular percentile values. The relevant paragraphs of the revised manuscript are included below:

A draft rewrite of the portion of the paper describing the sensitivity analyses is listed below:

> Following this, a sensitivity analysis was performed on the trained ML models. We allowed one predictor to vary across its min-max range while holding the other two input variables at specific percentile values. This was repeated for each predictor. This allowed us to isolate the impact of each predictor on the biomass – creating "cross-sections" of the dataset where only one variable changed at a time. For comparison, these values were also run through Eq. 1-3 to calculate the true response of how the simple phytoplankton model would behave. This allowed us to view which of the models most closely reproduced the underlying intrinsic relationships of the simple phytoplankton model.

For our main sensitivity analyses, we chose to hold the predictors that were not being varied at their respective 25[th], 50[th], and 75[th] percentile values. We chose to use these particular percentile values for several reasons:

1. It allowed us to avoid the extreme percentiles (1[st] and 99[th]). As we approach these extremes, the uncertainty in the predictions grows quite rapidly because of the lack of training samples within that domain space of the dataset. For example, there are no observations which satisfy the conditions of being in the 99[th] percentile of two variables simultaneously. This extreme distance outside of the training domain generally leads to standard deviations in predictions that are too large to provide a substantial level of certainty about the ML model's predictions.
2. Similar to the idea that we can avoid the extremes, we also chose these values as they are quite typical values for the edges of box plots. Generally, values within the range of the 25[th] to 75[th] percentiles are not considered outliers. Along those lines, we wanted to examine the conditions in a domain space that are likely to be found in actual observational datasets, with the reasoning that if there was high uncertainty in the ML predictions at these more moderate levels, there would be even higher uncertainty towards the extremes.

RC8: Did the authors try to increase the number of hidden layers in their NNs? It is known that the introduction of more hidden layers can improve the results. It would be interesting to see if there is any effect from the number of hidden layers in this particular problem.

AR8: In responding to this particular comment, we are assuming the Referee is referring to the outcomes of Scenario 2 since the performance in Scenario 1 was already robust.

For our manuscript we chose to use single hidden layer neural networks with 25 nodes in the hidden layer for several reasons:

- We ran a diagnostic test to determine the ideal number of nodes for each neural network in Scenario 2 (Table B1 in the revised manuscript). From the $R^2$ values in Table B1, it can be observed that the performance of each NNE begins to plateau around 10-15 nodes. However, the inclusion of additional nodes up to about 25 nodes did not cause the training times to increase drastically. We chose to include these extra nodes due to their moderate training times and robust performance.
- All three Scenarios showed ideal performances in the 10-20 node range, but similar to the first point, we chose to include extra nodes due to their moderate training times and good performance. Additionally, this allowed us to be consistent in the architecture of the neural networks across all the Scenarios and helped to minimize the differences between them.
- We considered adding more hidden layers to the neural networks (Table B2 in the revised manuscript), but the slight increase in performance did not seem worthwhile due to significantly increased training times. As can be observed in Table B2, the performance did not significantly increase from a single hidden layer to when a second layer with additional nodes

was included (0.9700 vs 0.9722-0.9726).

- Although some applications may require additional hidden layers, it is generally accepted that one hidden layer can approximate most functions provided there are sufficient nodes in the hidden layer and the nodes utilize squashing functions, such as sigmoid functions (Hornik et al., 1989).

Hornik, K., Stinchcombe, M. and White, H.: Multilayer feedforward networks are universal approximators, Neural Networks, 2(5), 359–366, doi:10.1016/0893-6080(89)90020-8, 1989.

RC9: The authors did not provide how they scaled their variables (lines 337-341). This procedure is known as normalization of variables. Normalization ensures that all predictors fall within a comparable range and avoids giving more weight to predictors with large variability ranges.

AR9: We have included the equations that we used to scale and unscale the values of each variable (Eq. 9 and 10 in the revised manuscript).

The text of the revised manuscript is below:

Each variable was scaled between -1 and 1 based on its respective maximum and minimum (Eq. 9).

$$V_S = \frac{max_S - min_S}{max_U - min_U} (V_U - min_U) + min_S \tag{9}$$

where V is the value of the variable being scaled, S stands for the scaled value, and U represents the unscaled value. This step ensures that no values are too close to the limits of the hyperbolic tangent sigmoid activation function, which would significantly increase the training time of each NN. Additionally, this normalization ensures that each predictor falls within a similar range, so more weight is not provided to variables with larger ranges. Although scaling is not necessary for RF and MLR, the scalings used for the NNE were still applied to each method for consistency. The results presented in this paper were then transformed back to their original scales to avoid confusion from scaling (Eq. 10).

$$V_U = \frac{max_U - min_U}{max_S - min_S} (V_S - min_S) + min_U \tag{10}$$

Where the letters represent the same values as in Eq. 9.

RC10: It is hard to agree that the NNEs and RFs represent well the behavior as for example on Figure 3 in left column the NNEs show a strong increase at the end of the macronutrient range that does not in agreement with the observation values; and there is a false decrease in the middle column at 25th percentile.

AR10: These behaviors likely have several causes:

1. In the areas of the dataspace where there are few or no observations, the NNEs are less certain about the predictions. This can be noted as the large gray areas in the middle column for the 25th percentile in Fig. 3 (Fig. 4 in the revised manuscript). Additionally, the uncertainties in the previously mentioned subplot may seem large due to the magnitude of the predictions for those conditions. For example, the y-axis of the middle column for the 25th percentile of Fig. 3 (Fig. 4 in the revised manuscript), shows the biomass in the range of $10^{-7}$. The other percentile plots in the middle column of Fig. 3 (Fig. 4 in the revised manuscript) show biomass in the range of $10^{-6}$, a full order of magnitude larger than the biomass in the 25th percentile of the same column. This point is reinforced by the contour plots in Fig. 9 (revised manuscript), which show the density of observations for pairs of predictors. The areas where the NNEs show false increases and decreases correspond to areas where there are no observations to constrain the NNEs.

2. Since the lines in the sensitivity analyses for the NNEs are the average response of ten individual neural networks, it's possible that the NNE line in the sensitivity analyses could be pulled higher or lower in areas of higher uncertainty due to extreme outlying predictions. This might explain the behavior of the increase in biomass at the end of the macronutrient range in the 75th percentile subplot of the left column in Fig. 3 (Fig. 4 in the revised manuscript). Additionally, since the random forest is an average of 500 trees it may be less susceptible to extreme outlying predictions, which would help to explain why that behavior is not seen in the random forest predictions for the previously mentioned macronutrient subplot.

We have only listed some of the potential causes, so others may exist to explain the deviations in behavior.

RC11: It is interesting to know if the authors have an idea about new parameter or variable that can bring back the information on hourly variability lost due to the time-averaging to improve the results?

AR11: We have considered a couple of ideas to attempt to bring back the variability lost due to the time-averaging:

1. We implemented a method that allowed us to visualize how the apparent relationships change from the hourly timescale through to the monthly averaged timescale. This method has now been incorporated into the revised manuscript (Fig. 7 of revised manuscript). The details of the method are included in Appendix D of the revised manuscript and is included here for reference:

To capture the apparent relationships ranging from the hourly to monthly averaged timescales, we averaged the hourly dataset over a range of timespans. Specifically, we averaged over the timespans of 1-hour (original hourly dataset), 2, 3, 4, 6, 8, 12, 24, 48, 72, 168 (weekly), and 720 (monthly) hours. The timescales had to be multiples of, or divisible by, 24 hours. Hours that did not meet these criteria would mean that hours from one day would be averaged with hours from another day. For example, using a 7-hour timespan for averaging would have meant that the last three hours of Day 1 were being averaged with the first four hours of Day 2.

We trained one NNE for each of the averaged timescales. Each NNE contained ten individual NNs. The NNs kept the same training criteria specified in the manuscript.

After training the NNEs, we performed a sensitivity analysis on each of them to visualize the predicted apparent relationships. The percentile values for variables that were not varying were set at their $50^{th}$ percentile (median) values. We then plotted all the predicted curves on a single surface plot so we could view the relationships of all the timescales at once. Additionally, because the greatest variability was lost in the first 24 hours, we also focused on the apparent relationships for the timespans that were less than or equal to 24 hours.

2. The following method was included in our original author responses to Referee 1 and it seemed worthwhile to include here. Ultimately, we chose not to use this method in the revised manuscript because the estimation of the half-saturation coefficients proved to be an unreliable way to quantitatively assess the ML's ability to reproduce the relationships.

We included additional variables and tested different percentile values.

We included the length of day (the number of hours in a day that a particular location received sunlight above 0 W m-$^2$) as a predictor for each of the time-averaged datasets of Scenario 2 and trained a new set of NNEs. With the exception of the number of predictors, the NNEs followed the same specifications listed in the original manuscript (10 individual NNs in each NNE, 25 nodes in hidden layer, one hidden layer, stopping criteria, etc.).

We knew from previously testing the higher percentile values ($>90^{th}$ percentile) for the sensitivity analyses that the higher percentiles led to greater uncertainty in the predictions (Fig. AR11.1). Although the true relationships were captured in the $100^{th}$ percentile plots, insofar as they were within the gray standard deviations

of predictions, the uncertainty in the predictions had a range equal to or greater than the response (Fig. AR11.1 g, h, i).

When we included length of day as a predictor and used the high percentile values of the other variables, we found that the 100th percentile plots captured the true relationship and the gray standard deviation ranges decreased (Fig. AR11.2 i, j, k, l). Furthermore, all three of the time-averaged datasets were able to capture the correct relationships.

[Figure]

Figure AR11.1: Sensitivity analysis for Scenario 2 showing the true and predicted relationships for how each predictor affects the biomass when the other predictors are set at higher percentiles (90, 95, and 100). The columns correspond to the predictors and the rows correspond with the percentile value at which the other predictors were set. The black line shows the true intrinsic relationship calculated from Eq. 1 and 2. The dashed lines show the predicted apparent relationships for each time-averaged dataset (Daily – red; Weekly – blue; Monthly – green). The gray region around the dashed lines shows the standard deviation of the predictions. The conditions of the sensitivity analyses were based on the conditions of the monthly-averaged dataset. It was necessary to give the same conditions to the all of the time-averaged datasets and to the simple model so that a direct comparison could be made between the true response and the predictions from each time-averaged dataset.

[Figure]

Figure AR11.2: Sensitivity analysis for Scenario 2 with Length of Day as an additional predictor showing the true and predicted relationships for how each predictor affects the biomass when the other predictors are set at higher percentiles (90, 95, and 100). The columns correspond to the predictors and the rows correspond with the percentile value at which the other predictors were set. The black line shows the true intrinsic relationship calculated from Eq. 1 and 2. The dashed lines show the predicted apparent relationships for each time-averaged dataset (Daily – red; Weekly – blue; Monthly – green). The gray region around the dashed lines shows the standard deviation of the predictions. The conditions of the sensitivity analyses were based on the conditions of the monthly-averaged dataset. It was necessary to give the same conditions to the all of the time-averaged datasets and to the simple model so that a direct comparison could be made between the true response and the predictions from each time-averaged dataset.

RC12: The article misses the total results for Scenarios 2 and 3 like on Figure 1.

AR12: The contour plots of Figure 1 (original and revised manuscript) were mostly intended to provide a visual representation of the difference in performance of the ML methods compared to multiple linear regression. Table 1 (Table 2 in revised manuscript) serves the same purpose as Figure 1 (original and revised manuscript), but with performance metrics instead of visual representation. Additionally, Table 1 (Table 2 in revised manuscript) provides quantifiable measures of each method's performance, whereas Figure 1 (original and revised manuscript) only allows for a visual qualitative measure. Since Table 1 (Table 2 in revised manuscript) serves as a quantifiable metric of performance we chose to use only tables for Scenarios 2 and 3, instead of contour plots like those in Figure 1 (original and revised manuscript).

RC13: In the Discussion of results for Scenario 3 the authors reasoned about BLING model behavior and did not mention their results. It would be interesting to know the authors' thoughts about NNEs behavior on Figure 7 middle column at 75th percentile.

AR13: From our best estimation, we assume that Referee 1 is indicating that we do not mention the specific results of any one particular ML method. For example, we did not differentiate between the responses found by RFs and NNEs.

In the Discussion section of Scenario 1, we mention that RF does not possess the same extrapolation capabilities as NNEs. Because of RF's inability to extrapolate, we focused our discussion of Scenario 3 on the relationships found by NNEs. However, it is now clear to us that we should either specify that point in the Discussion of Scenario 3 or discuss the relationships found by each method. The revised manuscript now only includes the relationships of the NNEs.

RC14: I have also found that the authors wrote too much on things that they did not do: lines 171-177.

AR14: We originally included this section because partial dependence plots (PDP) are a common visualization technique used in ML, especially for random forest (RF) analysis. In previous versions of this manuscript, initial comments (before submission to any journal) typically cited the use of PDPs vs sensitivity analyses as being an important distinction. So we felt it necessary to provide a brief explanation of how the use of sensitivity analyses differs from PDPs.

RC15: Lines 260 – 273: I suggest to rewrite these two paragraphs, there is a mention of results that have not yet been presented; and I do not see the necessity to mention that the authors "previously had little experience with ML."

AR15: We understand the confusion with these paragraphs. In past submissions, we were asked several questions:

1. Why had we chosen to use Random Forests and Neural Network Ensembles, instead of other machine learning algorithms?
2. Why did we choose to compare the performance of the machine learning algorithms to Multiple Linear Regression?

We tried to address these questions with these two paragraphs (lines 260-273), but we acknowledge that we may have included additional details that may not be needed. To address these two questions and still include details we find necessary, we have shortened the first of the two paragraphs mentioned in the Referee Comment. The text in the revised manuscript now reads:

We chose to use Random Forests (RFs) and Neural Network Ensembles (NNEs) in this manuscript. Although other ML methods exist, the list of possible choices is rather long. It was decided that the number of ML algorithms being compared would be limited to RFs and NNEs, given their popularity in studying ecological systems. Additionally, we

chose to compare the performance of the ML techniques to the performance of Multiple Linear Regression (MLR), which allows us to quantify the importance of nonlinearity.

Regarding the second paragraph, comments by other Referees have suggested the second paragraph (lines 270 to 273) remain unaltered. It provides an explicit statement by us that we are not trying to invalidate results or discourage the use of MLR in marine ecological systems.

RC16: Please use the figures captions like "a", "b" etc.

AR16: All figures that have subplots now have letter captions for each of the subplots. Additionally, the figure descriptions refer to the letter captions for subplots instead of their location within each figure. For example, when referring to a subplot, it is cited as "a" instead of the "top-left subplot in Fig. XX".

RC17: Please, expand the figure captions, for example on Figure 2 it would be good to add that the black line is estimated from Eq. 1.

AR17: The figure and table captions were expanded in the revised version of the manuscript to include more specific details. The text for Fig. 2 in the revised manuscript is included below for reference:

Figure 2: Sensitivity analysis for Scenario 1 showing the true and predicted relationships for each ML method. The columns correspond to the predictors and the rows correspond with the percentile value at which the other predictors were set (ex. Subplot **a** varies the macronutrient across its min-max range, while the micronutrient and light are held at their $25^{th}$ percentile values, respectively). The black line shows the true intrinsic relationship calculated from Eq. 1-3. The dashed lines show the predicted apparent relationships for each method (MLR – red; RF – blue; NNE – green). The RF and NNE predicted relationships are the average of the individual predictions for each method. The gray regions around the RF and NNE dashed lines show one standard deviation in the predictions (ex. One standard deviation in the 10 individual NN predictions of the NNE).

RC18: Figure 5 was mentioned as the last but it is placed before Figure 6 and 7.

AR18: The order of the figures has been corrected in the revised manuscript, so each figure is numbered based upon when it is first mentioned in the manuscript.

RC19:        Line 149, 156, 168, 356: please avoid to use the sign " in scientific paper.

AR19: We removed the quotations around the words in the specified lines. Additionally, we have reexamined our usage of other quotation marks throughout the paper and have removed quotations we deemed unnecessary.

RC20:        Line 149: sign " should be before the point .

AR20: We removed the quotations around that particular term.

RC21:        Line 206: it feels that "but" should be replaced by "and".

AR21: We replaced the transition word "but" with the transition word "and".

RC22:     Line 226, 235, 543: word "just" is unnecessary.

AR22: We removed the word "just" from the suggested lines.

RC23:     Line 250: "an ML" should be replaced by "a ML".

AR23: We replaced all instances of "an ML" with "a ML".

RC24:     Line 356: Miss a figure indication.

AR24: We filled in the missing figure indication.

**Updated Author Responses to Comments from Luke Gregor (Referee 2)**

In our previous Author Comments, we occasionally stated phrases such as "(Some change) will be included in the revised manuscript." Where applicable, we have updated our Author Responses below and now reference the actual revised manuscript.

In the following responses, RC stands for Referee Comment and AR stands for Author Response. Referee Comments are in black-colored font and Author Responses are in red-colored font. Unless specified otherwise, references to figure/table numbers and line numbers refer to those of the original manuscript.

RC0: The study by Holder and Gnanadesikan tries to assess if machine learning is able to extract the intrinsic relationship between phytoplankton growth and limiting nutrients and light from observed concentrations of nutrients and light intensity. This topic was investigated with three experiments of increasing complexity asking the following questions (with my brief understanding of the outcomes):

1. Are ML methods able to extract the relationship from observations at all at instantaneous time scales?
   a. Yes, but NNE is better at extracting the relationship than RF despite both achieving fair results
2. If time scales are averaged, can the relationships still be extracted?
   a. Not very well. In most cases the estimated half-saturation is lower than it should be. I.e. even the better of the two ML methods, NNE, is not very accurate.
3. Can the approach work in a more complex model setup where biomass losses are also accounted for?

While I appreciate the question the study is asking and think that this work is important, I found that the manuscript was not very easy to follow (my summaries of the results above might illustrate this). Part of the difficulty may be that the topic is not within my immediate field of expertise, but then I feel there are stylistic changes to be made that will improve the manuscript. I have overall comments in the document below and I linked a **PDF document with comments at the very end of this document** (I used Adobe Reader). I hope these comments help improve the flow of the manuscript.

AR0: We would like to thank Luke Gregor as Referee 2. We have found the comments and suggestions they provided to be very helpful in restructuring this manuscript. In particular, the supplement to these comments has provided some very specific and constructive feedback.

RC1: The title can be improved. To someone who is not familiar with the "intrinsic" and "apparent" terminology, the title is not informative. Something along the lines of : *Can machine learning extract the mechanisms of phytoplankton growth from large-scale observations?*

AR1: We understand how including the terminology in the title can lead to confusion. The title of the manuscript has been changed. The title of the revised manuscript is "Can machine learning extract the mechanisms controlling phytoplankton growth from large-scale observations? – A proof of concept study".

RC2: The use of "intrinsic" and "apparent" relationships this early in the manuscript made it difficult to understand the study as I am not familiar with the terminology.

AR2: The terms "intrinsic" and "apparent" relationships are actually terms that we are defining for the first time in this manuscript. They have not been previously introduced in oceanography literature. Since these terms are used frequently throughout the paper, we find it helpful to introduce them early, including in the abstract.

RC3: I don't have major concerns with the introduction and it builds a good case for why this study is relevant.

AR3: We thank Referee 2 for this kind compliment.

RC4: The questions posed (L72-75) and ideas presented (L100-102) are useful in framing   the study but are not carried clearly through the manuscript. It would be very useful for the reader to have these questions and ideas as a guide for why each experiment was performed. For example, L72-75 from the basis of experiment 1, but these questions are not explicitly answered in the discussion. And lines 100-102 form the basis of the design for experiment 2.

AR4: We have changed some aspects of the introduction based on other Referee comments, which encompass similar feedback as in RC4 above. The revised version of the introduction that highlights the two main questions we are trying to answer is included below for reference:

To investigate when and why the link between intrinsic and apparent relationships break, we try to answer two main questions in this paper:

1. Can ML techniques find the correct underlying intrinsic relationships and, if so, what methods are most skillful in finding them?
2. How do you interpret the apparent relationships that emerge when they diverge from the intrinsic relationships we expect?

In addressing the first question, we first needed to demonstrate that we had a ML method that would correctly extract intrinsic relationships from apparent relationships. We constructed a

simple model in which the biomass is directly proportional to the time-smoothed growth rate. In this scenario, intrinsic and apparent relationships operated on the same time and spatial scale and were only separated by a scaling factor, but the environmental drivers of phytoplankton growth had realistic inter-relationships. Having a better handle on the results from the first question, we were able to move onto the second question where we looked at where the link between intrinsic and apparent relationships diverged. We modified the first scenario so that the apparent relationships use a time-averaged input (similar to what would be used in observations), but the intrinsic relationships operate by smoothing growth rates derived from hourly input. Finally, we conduct a proof-of-concept study with real output from the ESM used to generate the inputs for scenarios 1 and 2, in which the biomass is a nonlinear function of the time-smoothed growth rate.

RC5: There is no overview of the methods. I think this would be useful in addition to an accompanying diagram outlining all the experiments and the use of the machine learning approaches used. It would help the reader understand the flow of the study. BLING is used throughout the study, albeit with different outputs from the model, but it may make sense to introduce the model before the experiment configurations are described.

AR5: We have included a diagram (Table 2 in the revised manuscript) outlining the details of each scenario which include: the predictor variables, the target variable, the equations used to calculate biomass, a description of the source file, and a short description of each scenario.

Because the machine learning approaches are the same for each Scenario, we did not think it would be necessary to include a table or diagram showing this. In the revised manuscript, we have stated more clearly in each of the Scenarios that the same ML methods used in Scenario 1 are also used in Scenarios 2 and 3.

The main reason for including the description of BLING in the third scenario was so readers would not get confused as to which equations and model were being used for each scenario. The revised manuscript keeps the description of BLING in third scenario.

RC6: It would make sense to formalise the following structure for each experiment:

- A brief introduction to the experiment

- HEADING for data

- HEADING for Machine learning parameterisation / application

AR6: In our original author comments, we stated that we would consider using the suggested format for the methods, but we chose to use a different style of structure for the methods section. The revised methods section more clearly states the goal of each Scenario. Additionally, we have included a table outlining the important information for each Scenario (Table 1 in revised manuscript).

We chose not to have a separate heading for the ML parameterizations in each Scenario since the same framework and construction (number of nodes, training parameters, etc.) were the same across all three Scenarios.

RC7: In experiment 2, the authors create hourly data by simulating variability of light conditions. The data are then averaged again to create daily, weekly and monthly data. If I understand correctly, the hourly data is analogous to the data used in experiment 1 - i.e. there is no temporal averaging in the "apparent data". It would be much more methodologically consistent to use the hourly data in experiment 1 and easier for the reader to follow. Either, the authors should implement this, or should make this explicit and state the reason that a separate experiment is needed.

AR7: Yes, the Referee is correct in their understanding that the hourly data is analogous to the data used in Scenario 1 where there was no temporal averaging. We agree that it would be easier for the reader to follow, and we spent several days testing this strategy.

The main issue we ran into was with the size of the hourly dataset. Across all longitudes, latitudes, and hours for a single year, this results in a dataset with 56,214,560 observations. We attempted to randomly sample the dataset with up to 500,000 points to train the machine learning algorithms. Quantities of observations higher than 500,000 were leading to computer crashes because of the computational power required for training the ML algorithms. While it is technically feasible to train random forests and neural networks on this number of observations, this would still require very long spans of time for training each ML method. Since we would like this paper and the methods to be accessible to everyone, we would like our Matlab code to be able to run on a standard laptop. With this in mind, we chose the first BLING scenario since it was already at monthly timescales and the number of observations was significantly less than the amount in an hourly dataset over the course of a year. The number of samples in the monthly dataset of Scenario 1 is only 77,328 compared to the 56 million of the hourly dataset.

RC8: Another question is regarding the model: what is the variability of the nutrients at a daily resolution (native model resolution), and the averaged resolutions (weekly, monthly). Show some violin/box plots for the normalised data.

AR8: A figure with boxplots for the time-averaged datasets of Scenario 2 is included in the revised manuscript as Fig. 6. Additionally, we have included boxplots for Scenarios 1 and 3 in Appendix C of the revised manuscript.

RC9: I still don't fully understand what the predictors and target variables are for each experiment and what is the role of the intrinsic? From what I understand, predictors are always the "apparent" data and biomass is the target. The intrinsic is what describes the relationship between the biomass and the "apparent data". Please make this more clear. Addressing the points above in the structure section will help with this.

AR9: We have clarified the predictor and target variables for each Scenario in the methods section of the revised manuscript. Additionally, we included Table 1 (in the revised manuscript) which sums up the important details of each Scenario.

Regarding the intrinsic and apparent relationships, the intrinsic relationships are those in which the effects of other variables affecting the target variable can be controlled. For example, if one is measuring the effect of macronutrient concentrations on phytoplankton in the lab, it is possible for them to hold concentrations of other variables (light, micronutrient, water temperature, salinity, etc.) at some particular value. Apparent relationships are those for which the effects of other variables affecting the target variable **cannot** be controlled (ex. taking measurements in the field). Another way of saying this is that intrinsic relationships are the underlying relationships governing a system where you can adjust one variable at a time (such as a lab). Apparent relationships are determined by how the intrinsic relationships combine in the environment when variables cannot be adjusted one variable at a time. We have clarified this in the revised manuscript.

RC10: The authors should only NNE results for experiment 2 (figure 4). Is there a reason for this? My presumption is that the intrinsic relationship estimated by RF for micronutrients is poor, thus only NNE is shown. This should be cleared up (unless I missed this).

AR10: The presumption of the Referee is correct. Because the RF performs poorly and is incapable of extrapolating outside the range of the training dataset, we chose to limit further analyses of Scenario 2 to NNEs. This has been clarified in the revised manuscript with the following sentence:

> Because the NNEs showed the closest approximations to the correct shape and magnitude of the curves compared to RF and MLR (Fig. 4), the remaining analysis of Scenario 2 is mainly focused on NNEs.

RC11: From what I understand, the half-saturation constants are the metric for whether the method is able to capture the intrinsic from the apparent. Make this much more clear - also in the abstract

AR11: That is correct. We were using the calculated half-saturation constants as a metric to help identify if the methods were capturing the true relationships. However,

after closer examination we have learned that estimating of the half-saturation constants from predicted curves in the single-variable sensitivity analyses is an unreliable method. This is mostly because the single-variable sensitivity analyses do not allow us to take co-limitations by the other variables into account. We have clarified this in the manuscript.

RC12: The subheadings could be the questions posed in the introduction (see my previous comments on this section). This would help guide the reader

AR12: In our original author responses, we stated that we might use subheadings in the discussion to guide the reader. While we did not use the actual questions posed in the introduction as the subheadings in the revised manuscript, we have largely associated each question with one scenario (first question with Scenario 1 and second question with Scenario 2).

RC13: I think the authors should make the point that given the simplicity of the definition of biomass, one would expect the ML methods to perfectly represent the Michaelis-Menten curves. The authors do correctly state that RF is less likely to estimate accurately as the method is not able to extrapolate. This then increases the importance of showing the distributions of the training and test data set distributions. A further comment: what is the envelope around the estimated curves and why is there a large variability for the NNE at larger values?

AR13: To keep the number of figures in the manuscript to a minimum, we had not included boxplots of each variable in each Scenario. However, we see the use that information can provide. The revised manuscript now includes the distributions of each variable. Scenario 2 is represented in Fig. 6 of the revised manuscript and Scenarios 1 and 3 can be found in Appendix C of the revised manuscript.

The gray regions around the dashed lines for the random forest (RF) and neural network ensemble (NNE) predictions show the standard deviation in the predictions. For example, the NNEs are composed of 10 individual neural networks and each one produces its own predictions. For the sensitivity analysis figures, the dashed lines for NNE show the average prediction of those 10 individual neural networks. Similarly, the gray regions show the range of one standard deviation for those predictions. We have clarified this in the revised manuscript, especially in the descriptions of each figure.

The large variability for the NNEs at the larger values is likely because those particular conditions are outside the range of the dataset on which the NNEs were trained. For example, it is rare that any of the observations would have high macronutrient, high micronutrient, and high irradiance occurring at the exact same time and location. Without any observations in the training subset meeting those types of criteria and the NNE never having seen what those conditions actually produce, the NNE predictions

become less certain. For more information, please see our response to Referee 1 (AR10) above.

RC14: The discussion around scenario/experiment 3 is not clear and I don't feel that there is a take-home message after reading this section.

AR14: The purpose of Scenario 3 is largely to provide a proof-of-concept to how the techniques we demonstrate in Scenarios 1 and 2 can be applied to Earth System Model output. We have clarified this and expanded on the results/discussion in the revised manuscript.

RC15: The captions are not standalone for both figures and tables.

AR15: The revised manuscript includes more detailed descriptions of the tables and figures. The text for Fig. 2 of the revised manuscript is included below as an example:

Figure 2: Sensitivity analysis for Scenario 1 showing the true and predicted relationships for each ML method. The columns correspond to the predictors and the rows correspond with the percentile value at which the other predictors were set (ex. Subplot **a** varies the macronutrient across its min-max range, while the micronutrient and light are held at their 25$^{th}$ percentile values, respectively). The black line shows the true intrinsic relationship calculated from Eq. 1-3. The dashed lines show the predicted apparent relationships for each method (MLR – red; RF – blue; NNE – green). The RF and NNE predicted relationships are the average of the individual predictions for each method. The gray regions around the RF and NNE dashed lines show one standard deviation in the predictions (ex. One standard deviation in the 10 individual NN predictions of the NNE).

RC16: The reader needs to know what the target variable in each table is and there are no units.

AR16: The revised manuscript now includes the target variable and its units in the description of the appropriate tables.

RC17: What is the envelope around the dashed lines.

AR17: The gray regions around the dashed lines for the random forest (RF) and neural network ensemble (NNE) predictions show the standard deviation in the predictions. For example, the NNEs are composed of 10 individual neural networks and each one produces its own predictions. For the sensitivity analysis figures, the dashed lines for NNE show the average prediction of those 10 individual neural networks. Similarly, the gray regions show the range of one standard deviation for those predictions. For additional information, please see our response in AR13.

RC18: Please also note the supplement to this comment:
https://bg.copernicus.org/preprints/bg-2020-262/bg-2020-262-RC2-supplement.pdf

AR18: The additional Referee comments in the supplement are very helpful. These comments are addressed below.

**Updated Author Responses to Supplemental Comments from Luke Gregor (Referee 2)**

The Supplemental PDF attached to Referee 2's comments was a PDF of the original submitted manuscript with specific comments by Referee 2 as highlighted PDF comments.

Any minor grammatical errors (commas, periods, and other punctuation) noted in Referee 2's comments were corrected in the revised manuscript. So the discussion can be focused on the comments, we have not included those grammatical errors in this Author Response. However, we do want to thank Referee 2 for finding grammatical errors that we missed.

In our previous Author Comments, we occasionally stated phrases such as "(Some change) will be included in the revised manuscript." Where applicable, we have updated our Author Responses below and now reference the actual revised manuscript.

For ease of reading the following responses addressing the comments of the Supplemental PDF attached to Referee 2's comments, we have included the referenced paragraph from the original manuscript, along with the associated line numbers in black-colored font. The text sections in the original manuscript that were highlighted by Referee 2 are in orange-colored font. Any line numbers that are referenced refer to the line numbers of the original submitted manuscript. Referee 2's comments then follow the paragraph in green-colored font and our Author Responses follow this in red-colored font.

Acronyms used in this Author Response include OMT (Original Manuscript Text), RHS (Referee Highlighted Section), Referee Comment (RC), and Author Response (AR).

OMT:

Lines 10-30

**Abstract.** Controls on phytoplankton growth are typically determined in two ways: by varying one driver of growth at a time such as nutrient or light in a controlled laboratory setting (intrinsic relationships) or by observing the emergence of relationships in the environment (apparent relationships). However, challenges remain when trying to take the intrinsic relationships found in a lab and scaling them up to the size of ecosystems (i.e., linking intrinsic relationships in the lab to apparent relationships in large ecosystems). We investigated whether machine learning (ML) techniques could help bridge this gap. ML methods have many benefits, including the ability to accurately predict outcomes in complex systems without prior knowledge. Although previous studies have found that ML can find apparent relationships, there has yet to be a systematic study that has examined when and why these apparent relationships will diverge from the underlying intrinsic relationships. To investigate this question, we created three scenarios: one where the intrinsic and apparent relationships operate on the same time and spatial scale, another model where the intrinsic and apparent relationships have different timescales but the

same spatial scale, and finally one in which we apply ML to actual ESM output. Our results demonstrated that when intrinsic and apparent relationships are closely related and operate on the same spatial and temporal timescale, ML is able to extract the intrinsic relationships when only provided information about the apparent relationships. However, when the intrinsic and apparent relationships operated on different timescales (as little separation as hourly to daily), the ML methods underestimated the biomass in the intrinsic relationships. This was largely attributable to the decline in the variation of the measurements; the hourly time series had higher variability than the daily, weekly, and monthly-averaged time series. Although the limitations found by ML were overestimated, they were able to produce more realistic shapes of the actual relationships compared to MLR. Future research may use this type of information to investigate which nutrients affect the biomass most when values of the other nutrients change. From our study, it appears that ML can extract useful information from ESM output and could likely do so for observational datasets as well.

RHS1: Abstract (Line 10)

RC1: General comment: I find that the language used in the abstract might complicate the message.

AR1: Some of the confusion appears to be in the language and terminology used in the beginning of the abstract, which includes the lines mentioned in RC2, RC3, and RC4. The revised manuscript has clarified the language in the abstract to reflect the main points of the paper more accurately. The first portion of the abstract in the revised manuscript is included below for reference:

> A key challenge for biological oceanography is relating the physiological mechanisms controlling phytoplankton growth to the spatial distribution of those phytoplankton. Physiological mechanisms are often isolated by varying one driver of growth, such as nutrient or light, in a controlled laboratory setting producing what we call "intrinsic relationships". We contrast these with the "apparent relationships" which emerge in the environment in climatological data. Although previous studies have found machine learning (ML) can find apparent relationships, there has yet to be a systematic study examining when and why these apparent relationships diverge from the underlying intrinsic relationships found in the lab, and how and why this may depend on the method applied. Here we conduct a proof-of-concept study with three scenarios in which biomass is by construction a function of time-averaged phytoplankton growth rate. In the first scenario, the inputs and outputs of the intrinsic and apparent relationships vary over the same monthly timescales. In the second, the intrinsic relationships relate averages of drivers that vary on hourly timescales to biomass, but the apparent relationships are sought between monthly averages of these inputs and monthly averaged output. In the third scenario we apply ML to the output of an actual Earth System Model (ESM).

RHS2: emergence of relationships in the environment (apparent relationships). (Line 12)

RC2: This could be much more clear and explicit - i.e. observed nutrient concentrations and light intensity.

AR2: Please see our response in AR1.

RHS3: We investigated whether machine learning (ML) techniques could help bridge this gap. (Lines 14-15)

RC3: Be much more specific here. See my general comments what the title should be.

AR3: Please see our response in AR1.

RHS4: prior knowledge. (Line 16)

RC4: prior knowledge needs to be qualified - ML uses data to predict. Perhaps the authors mean "knowledge of the system" as stated later

AR4: The abstract in the revised manuscript no longer describes ML in the way it was presented in the abstract of the original manuscript. However, we have kept the term "knowledge of the system" in the introduction of the revised manuscript.

RHS5: apply ML to actual ESM output. (Line 21)

RC5: To do what?

AR5: The objective was to apply ML to actual ESM output as a proof-of-concept to the kinds of relationships and information one can find. Please also see our response in AR1.

RHS6: intrinsic relationships. (Line 25)

RC6: Why are you estimating intrinsic relationships? Is this predicted with the intrinsic relationship?

AR6: We are trying to understand when and why the intrinsic and apparent relationships diverge. This sentence has been reworded. The new sentence in the revised manuscript reads:

> "When intrinsic and apparent relationships operated on different timescales (as little separation as hourly versus daily), NNEs fed with apparent relationships in time-averaged data produced responses with the right shape but underestimated the biomass."

A key point here is that the intrinsic relationships are what gets coded into the models. If these are incorrect (i.e. the model gets the right answer in a given location due to compensating errors in the intrinsic relationships) it will have the wrong sensitivity to climate change.

RHS7: limitations found by ML were overestimated, (Line 27)

RC7: First mention of limitations - does this refer to limitations of growth?

AR7: By limitations, we are referring to limitations on biomass. For example, the concentration of micronutrient can limit phytoplankton biomass.

RHS8: MLR (Line 28)

RC8: write this out in full

AR8: The revised manuscript does not use the acronym MLR in the manuscript, but instead writes out the full term for "Multiple Linear Regression."

OMT:

Lines 44-56

Limitations on phytoplankton growth are usually characterized in two ways – which we term intrinsic and apparent. Intrinsic relationships are those where the effect of one driver (nutrient/light) at a time is observed, while all others are held constant (often at levels where they are not limiting). An example of such intrinsic relationships is the Michaels-Menten growth rate curves that emerge from laboratory experiments (Eppley and Thomas, 1969). Apparent relationships are those which emerge in the observed environment. An example of apparent relationships is those that emerge from satellite observations, which provide spatial distributions of phytoplankton on timescales (say a month) much longer than the phytoplankton doubling time, which can be compared against monthly distributions of nutrients. A significant challenge that remains is determining how intrinsic relationships found in the laboratory scale up to the apparent relationships observed at the ecosystem scale (i.e., scaling the small to the large). Differences may arise between the two because apparent relationships reflect both intrinsic growth and loss rates, which are near balance over the long monthly timescales usually considered in climatological analyses. Biomass concentrations may thus not reflect growth rates. Differences may also arise because different limitation factors may not vary independently.

RHS9: Limitations on phytoplankton growth are usually characterized in two ways (Line 44)

RC9: This paragraph makes the terminology used in the abstract much clearer

AR9: The terms "intrinsic" and "apparent" relationships are terms that we define for the first time. To the best of our knowledge, these terms have not previously appeared in oceanography literature. Because we use these terms frequently throughout the manuscript, we included them in the abstract.

OMT:

Lines 58-74

Earth System Models (ESMs) have proved valuable in linking intrinsic and apparent relationships. The intrinsic relationships are programmed into ESMs as equations that are run forward in time and the output is typically provided as monthly-averaged fields. The output of these ESMs is then compared against observed fields such as chlorophyll and nutrients and can be analyzed to find apparent relationships between the two. If the ESM output is close to the observations we find in nature, we say that the ESM is performing well. However, as recently pointed out by Löptien and Dietze (2019), ESMs can trade-off biases in physical parameters with biases in biogeochemical parameters (i.e., they can arrive at the same answer for different

reasons). Using two versions of the UVic 2.9 ESM, they showed that they could increase mixing (thus bringing more nutrients to the surface) while simultaneously allowing for this nutrient to be more efficiently cycled – producing similar distributions of surface properties. However, the carbon uptake and oxygen concentrations predicted by the two models diverged under climate change. Similarly, Sarmiento et al. (2004) showed that physical climate models would be expected to produce different spatial distributions of physical biomes due to differences in patterns of upwelling and downwelling, as well as the annual cycle of sea ice. These differences would then be expected to be reflected in differences in biogeochemical cycling, independent of differences in the biological models. These studies highlight the importance of constraining not just individual biogeochemical fields, but also their relationships with each other. What is less clear is: 1. Can robust relationships be found? 2. If so, what methods are most skillful in finding them? 3. How do you interpret the apparent relationships that emerge when they diverge from the intrinsic relationships we expect?

RHS10: relationships (Line 73)

RC10: between what and what?

AR10: Relationships in this sentence refers to relationships in very broad terms between biogeochemical fields and the target variable. The target variable could be phytoplankton biomass, chlorophyll concentrations, or other biogeochemical variables.

The revised version of the manuscript removes these questions from this paragraph and combines them with other sentences in the last paragraph of the introduction to more clearly highlight the main points of the manuscript. The last paragraph of the introduction in the revised manuscript is included below for reference:

> To investigate when and why the link between intrinsic and apparent relationships break, we try to answer two main questions in this paper:
>
> 1. Can ML techniques find the correct underlying intrinsic relationships and, if so, what methods are most skillful in finding them?
> 2. How do you interpret the apparent relationships that emerge when they diverge from the intrinsic relationships we expect?
>
> In addressing the first question, we first needed to demonstrate that we had a ML method that would correctly extract intrinsic relationships from apparent relationships. We constructed a simple model in which the biomass is directly proportional to the time-smoothed growth rate. In this scenario, intrinsic and apparent relationships operated on the same time and spatial scale and were only separated by a scaling factor, but the environmental drivers of phytoplankton growth had realistic inter-relationships. Having a better handle on the results from the first question, we were able to move onto the second question where we looked at where the link between intrinsic and apparent relationships diverged. We modified the first scenario so that the apparent relationships use a time-averaged input (similar to what would be used in observations), but the intrinsic relationships operate by smoothing growth rates derived from hourly input. Finally, we conduct a proof-of-concept study with real output from the ESM used to generate the

inputs for scenarios 1 and 2, in which the biomass is a nonlinear function of the time-smoothed growth rate.

OMT:

Lines 76-81

Recently, researchers have turned to machine learning (ML) to help in uncovering the dynamics of ESMs. ML is capable of fitting a model to a dataset without any prior knowledge of the system and without any of the biases that may come from researchers about what processes are most important. As applied to ESMs, ML has mostly been used to constrain physics parameterizations, such as longwave radiation (Belochitski et al., 2011; Chevallier et al., 1998) and atmospheric convection (Brenowitz and Bretherton, 2018; Gentine et al., 2018; Krasnopolsky et al., 2010, 2013; O'Gorman and Dwyer, 2018; Rasp et al., 2018).

RHS11: As applied to ESMs, ML has mostly been used to constrain physics parameterizations, such as longwave radiation (Belochitski et al., 2011; Chevallier et al., 1998) and atmospheric convection (Brenowitz and Bretherton, 2018; Gentine et al., 2018; Krasnopolsky et al., 2010, 2013; O'Gorman and Dwyer, 2018; Rasp et al., 2018). (Lines 78-81)

RC11: There is also now the study by Kasim et al (preprint). https://www.researchgate.net/publication/338762727_Up_to_two_billion_times_acceleration_of _scientific_simulations_with_deep_neural_architecture_search

AR11: Thank you for bringing this publication to our attention. While we found this study interesting and the results promising, we had some questions about the manuscript. We feel that the peer-reviewed published article will make a worthwhile contribution to the field, but we have not included it in the revised manuscript as we would like to see how it develops.

OMT:

Lines 83-94

With regards to phytoplankton, ML has not been explicitly applied within ESMs but has been used on phytoplankton observations (Bourel et al., 2017; Flombaum et al., 2020; Kruk and Segura, 2012; Mattei et al., 2018; Olden, 2000; Rivero-Calle et al., 2015; Scardi, 1996, 2001; Scardi and Harding, 1999) and has used ESM output as input for an ML model trained on phytoplankton observations (Flombaum et al., 2020). Rivero-Calle et al. (2015) used random forest (RF) to identify the drivers of coccolithophore abundance in the North Atlantic through feature importance measures and partial dependence plots. The authors were able to find an apparent relationship between coccolithophore abundance and environmental levels of $CO_2$, which was consistent with intrinsic relationships between coccolithophore growth rates and ambient $CO_2$ reported from 41 laboratory studies. They also found consistency between the apparent and intrinsic relationships between coccolithophores and temperature. While they were

able to find links between particular apparent relationships found with the RFs and intrinsic relationships between laboratory studies, it remains unclear when and why this link breaks.

RHS12: it remains unclear when and why this link breaks. (Line 93)

RC12: The agreement between these two variables might be due to the scales of variability? Or a consistent reponse between the response of coccolithophores to temperature at a low and resolutions

AR12: Thank you for these suggestions. As we show in this paper, scales of variability can affect the link between intrinsic and apparent relationships.

OMT:

Lines 95-102

ML has been used to examine apparent relationships of phytoplankton in the environment (Flombaum et al., 2020; Rivero-Calle et al., 2015; Scardi, 1996, 2001) and it is reasonable to assume that ML could find intrinsic relationships when provided a new independent dataset from laboratory growth experiments. However, it has yet to be determined under what circumstances the apparent relationships captured by ML are no longer equal to the intrinsic relationships that actually control phytoplankton growth. In this paper, we identify two drivers of such divergence. The first is colimitation that limits the biological responses actually found in the ocean, which causes non-parametric ML methods to produce apparently non-physical results. The second is climatological averaging of the input and output variables, which can distort these relationships in the presence of non-linearity.

RC13: The use of "identify" here makes me think that this is a result from the study. Perhaps "propose"?

AR13: Other reviewers have also pointed to this portion of the introduction as possibly containing results from the paper. To avoid confusion, we have removed the following sentences from the introduction of the revised manuscript:

> "In this paper, we identify two drivers of such divergence. The first is colimitation that limits the biological responses actually found in the ocean, which causes non-parametric ML methods to produce apparently non-physical results. The second is climatological averaging of the input and output variables, which can distort these relationships in the presence of non-linearity."

OMT:

Lines 115-118

In the first scenario, we wanted to determine how well different ML methods could extract intrinsic relationships when only provided information on the apparent relationships and when the intrinsic and apparent relationships were operating on the same timescale. In this scenario, the apparent relationships were simply the result of multiplying the intrinsic relationships between predictors and biomass by a scaling constant.

RHS14: In this scenario, the apparent relationships were simply the result of multiplying the intrinsic relationships between predictors and biomass by a scaling constant. (Lines 117-118)

RC14: I would add that three machine learning methods are used after this sentence.

AR14: In the revised manuscript, we have stated the three ML techniques we are using at the start of the Methods section. Additionally, we include a reference to Table 1 (in the revised manuscript) that includes details about each Scenario.

OMT:

Lines 148-152

The final dataset consisted of three input/predictor variables and one response term with a total of 77,328 "observations." The input variables given to each of three ML methods (Multiple Linear Regression, Random Forests, and Neural Network Ensembles, described in more detail below) were the concentrations (not the limitation terms) for the micronutrient, macronutrient, and light. The response variable was the biomass we calculated from Eq. 1 and 2.

RHS15: (Multiple Linear Regression, Random Forests, and Neural Network Ensembles, described in more detail below) (Lines 149-150)

RC15: I would introduce the use of these methods earlier. i.e. the general concept explained briefly in the opening paragraph.

AR15: Please see our response in AR14.

OMT:

Lines 154-162

The dataset was then randomly split into training and testing subsets, with 60% of the observations going to the training subset and the remainder going to the testing subset. This provided a convenient way to test the generalizability of each ML method by presenting them with "new" observations from the test subset and ensuring the models did not overfit the data. The input and output values for the training subset were then used to train a model for each ML method. Once each method was trained, we provided the trained models with the input values of the testing subset to acquire their respective predictions. These predictions were then compared to the actual output values of the test subset. To assess model performance, we calculated the coefficient of determination ($R^2$), the mean squared error (MSE), and the root mean squared error

(RMSE) between the ML predictions and the actual output values for the training and testing subsets.

RHS16: randomly split into training and testing subsets, (Line 154)

RC16: COMMENT: I'm usually not a fan of random splits (particularly in a simulated environment), as training and testing data would have very similar distributions. But since the goal here is to see if it is possible to do exactly that (can ML capture the relationships), it makes sense.

AR16: Our main purpose for splitting the data into training and testing subsets is to ensure the machine learning methods are not overfitting the data. As you (Referee 2) state in RC16, we want to ensure that the machine learning models capture the relationships.

RHS17: convenient (Line 155)

RC17: Not sure I'd use convenient here. This is standard machine learning practice.

AR17: Because it is a standard machine learning practice, the revised manuscript has replaced the term "convenient" with "standard" in the referenced sentence.

RHS18: "new" observations from the test subset and ensuring the models did not overfit the data. (Line 156-157)

RC18: Would be great to see a plot of the distribution of the training and test data (box plot). as well as the spatial distribution.

AR18: Since the observations in the training and testing subsets were randomly sampled from a large dataset, we felt it was apparent that the subsets contained observations of equal magnitude and spatiotemporal distribution. This was further reinforced in the similar performances of the training/testing subsets for the machine learning methods. However, we would like to remind the Referee and other readers that source files and code are freely available on the Zenodo data repository (https://doi.org/10.5281/zenodo.3932388, Holder and Gnanadesikan, 2020).

Although we did not create boxplots to show the distributions of the training and testing datasets separately, we have included boxplots in the revised manuscript that show the distributions of the full datasets (training and testing, combined; Fig. 6 and Appendix C in the revised manuscript).

Holder, C. D. and Gnanadesikan, A.: Linking intrinsic and apparent relationships between phytoplankton and environmental forcings using machine learning - What are the challenges?, doi:10.5281/zenodo.3932388, 2020.

OMT:

Line 195

$$\overline{L_{Irr}} = \overline{\frac{Irr}{K_{Irr}+Irr}} \neq \frac{\overline{Irr}}{K_{Irr}+\overline{Irr}} \tag{4}$$

RHS19:

$$\overline{L_{Irr}} = \overline{\frac{Irr}{K_{Irr}+Irr}} \neq \frac{\overline{Irr}}{K_{Irr}+\overline{Irr}} \tag{4}$$

(Line 195)

RC19: Maybe add brackets around the fraction with the overbar applied to the entire equation

AR19: To clarify that the overbar applies to the entire fraction term, we changed the highlighted term to include brackets in the revised manuscript.

OMT:

Lines 196-198

(Eq. 4 appears before this in the original manuscript)

[revised manuscript text omitted]

RC24: Good! I was going to comment on this if it was not mentioned in the text.

AR24: Yes, we wanted to make it clear that we were not invalidating multiple linear regression. It is a very useful method!

OMT:

Lines 276-285

RFs are an ensemble ML method utilizing a large number of decision trees to turn "weak learners" into a single "strong learner" by averaging multiple outputs (Breiman, 2001). In general, RFs work by sampling (with replacement) about two-thirds of a dataset and constructing a decision tree. At each split, the random forest takes a random subset of the predictors and examines which variable can be used to split a given set of points into two maximally distinct groups. This use of random predictor subsets helps to ensure the model is not overfitting the data. The process of splitting the data is repeated until an optimal tree is constructed or until the stopping criteria are met, such as a set number of observations in every branch (then called a leaf / final node). The process of constructing a tree is then repeated a specified number of times, which results in a group (i.e., "forest") of decision trees. Random forests can also be used to construct regression trees in which a new set of observations traverse each decision tree with its associated predictor values and the result from each tree is aggregated into an averaged value.

RHS25: two-thirds of a dataset and constructing a decision tree. (Line 278)

RC25: might be good to add that this is commonly referred to as bootstrap aggregation, or bagging in the machine learning world

AR25: We have included this term in the revised manuscript.

RHS26: Random forests can also be used to construct regression trees in which a new set of observations traverse each decision tree with its associated predictor values and the result from each tree is aggregated into an averaged value. (Lines 283-285)

RC26: This sentence is not completely clear. Are you trying to say that the predicted value is the average of all tree's prediction values.

AR26: Yes, that is what we are trying to say. When using random forest for regression (instead of classification), the predicted value is the average of all the individual trees' predictions.

OMT:

Lines 287-293

Here, we used the same parameters for RF in the three scenarios to allow for a direct comparison between the scenarios and to minimize the possible avenues for errors. Each RF scenario was implemented using the TreeBagger function in MATLAB 2019b, where 500 decision trees were constructed with each terminal node resulting in a minimum of five observations per node. An optimization was performed to decide the number of decision trees that minimized the error while still having a relatively short runtime of only several minutes. For reproducible results, the random number generator was set to "twister" with an integer of "123". Any remaining options were left to their default values in the TreeBagger function.

RHS27: minimum of five observations per node. (Line 290)

RC27: How did you decide on 5? This might result in overfitting given the nature and size of the training data set. Is there some sort of hyper-parameter selection process?

AR27: Five observations per node was the default number in the Matlab function used to construct the random forests. This means that at the end of each leaf of a decision tree within the random forest, five observations were being averaged for that single leaf. However, we still would not expect random forests to overfit the data. By construction, random forests generally do not overfit datasets for a couple reasons:

1.  The random way in which data is selected to build individual decision trees.
    a.  When any one decision tree is being constructed, the data is randomly selected with replacement from the available data until it reaches the same number of observations in the dataset. In general, this type of random sampling means that about 2/3 of the observations in a dataset will be captured by this type of sampling. This is then used to construct a decision tree. The process is repeated until the specified number of trees is reached. In our case, this would mean that our training subset is being randomly sampled for the construction of the decision trees. This type of decision tree construction for random forests also means that no decision tree will be trained with every sample, which further decreases the likelihood of overfitting.
2.  The random way in which the predictors are used in the construction of individual

decision trees.

a. When the decision trees are being constructed, only some of the predictors are available each time a split is determined. These predictors are randomly selected. Given those predictors, an error metric is used to determine the best split that will minimize the error. This random way in which predictors are selected decreases the chances of overfitting the data.

Additionally, we only allowed the random forest to be trained on the training subset. The other 40% of the data was in the testing subset and had never been "seen" by the random forests. Since the performance metrics of the training and testing subsets for random forest were very similar, this suggested to us that the relationships had been captured by the random forest in the training subset and those same relationships were present in the testing subset.

OMT:

Lines 304-310

Feed-forward NNs consist of nodes connected by synapses (or weights) and biases with one input layer, (usually) at least one hidden layer, and one output layer. The nodes of the input layer correspond to the input values of the predictor variables, and the hidden and output layer nodes each contain an "activation function." Each node from one layer is connected to all other nodes before and after it. The values from the input layer are transformed by the weights and biases connecting the input layer to the hidden layer, put through the activation function of the hidden layer, modified by the weights and biases connecting the hidden layer to the output layer, and finally entered into the final activation function of the output node.

RHS28: synapses (Line 304)

RC28: I would stick with weights rather than synapses. There has been a move away from comparing NNs with the brain as this gives far too much credit to the capabilities of NNs

AR28: The revised manuscript uses the term "weights" in place of "synapses".

RHS29: Each node from one layer is connected to all other nodes before and after it. (Line 306-307)

RC29: By design FFNNs are fully connected, meaning that each node from one layer is connected to all other nodes in preceding and succeding layers

AR29: Yes, that is correct. We included this line so that readers who may not be experienced in machine learning or neural networks understood the details of FFNNs.

OMT:

Lines 322-330

To minimize the differences between scenarios, we used the same framework for the NNs in each scenario. Each NN consisted of three input nodes (one for each of the predictor variables), 25 nodes in the hidden layer, and one output node. The activation function within the hidden nodes was a hyperbolic tangent sigmoid function and the activation function within the output node used a linear function. The stopping criteria for each NN was set as a validation check such that the training stopped when the error between the predictions and observations increased for six consecutive epochs. An optimization was performed to decide the number of nodes in the hidden layer that 11 minimized the error while maintaining a short training time. Additionally, sensitivity analyses were performed using different activation functions to ensure the choice of activation function had minimal effect on the outcome and apparent relationships found by the NNEs.

RHS30: validation check (Line 326)

RC30: What portion of the data was used for validation?

AR30: We used the default values in the Matlab function for training feedforward neural networks. These default values are 70% in the training set, 15% in the validation set, and 15% in the testing set. The function used in Matlab randomly partitions the data into these categories.

It should be noted that those partition values were only applied to our training subset, not the testing subset. The Matlab function used to train the NNs partitions our training subset into its own training, validation, and test sets. For example, this means that out of the 46,397 observations in our training subset of Scenario 1, 32,477 observations (70%) went to the training set of the NN function, 6,960 observations (15%) went to the validation set of the NN function, and 6,960 observations (15%) went to the test set of the NN function.

RHS31: An optimization was performed to decide the number of nodes in the hidden layer that (Line 327)

RC31: What kind of optimisation? Grid search? What ranges of nodes were explored?

AR31: Please see our responses to Referee 1 (AR8) for these details.

RHS32: ensure the choice of activation function had minimal effect on the outcome and apparent relationships found by the NNEs. (Lines 329-330)

RC32: I would phrase this differently. Perhaps the activation function does make a difference. There are tens of activation functions (https://stats.stackexchange.com/questions/115258/comprehensive-list-of-activation-functions-in-neural-networks-with-pros-cons). It would be more accurate to list the activation functions tested and state that it did not make a difference within these options.

AR32: The revised manuscript lists the activation functions that we tested (Table B3 in the revised manuscript).

OMT:

Lines 332-335

Each NNE scenario used the feedforwardnet function in MATLAB 2019b. Any options not previously specified remained at their default values in the feedforwardnet function. The NNEs contained ten individual NNs for each scenario. For reproducibility, the random number generator was set to "twister," and the random number seed was set to the respective number of its NN (i.e., 1, 2, 3, up to 10).

RHS33: For reproducibility, the random number generator was set to "twister," and the random number seed was set to the respective number of its NN (i.e., 1, 2, 3, up to 10). (Lines 334-335)

RC33: Details like this can be either in supplementary material and/or in the code.

AR33: The revised manuscript removes this sentence from the main text. It has been included in Appendix B2 of the revised manuscript.

OMT:

Lines 337-341

Each variable was scaled between -1 and 1 based on its respective maximum and minimum. This step ensures that no values are too close to the limits of the hyperbolic tangent sigmoid activation function, which would significantly increase the training time of each NN. These scalings were also applied to the RF and MLR methods for consistency between methods and the scaling did not affect the results of either method (results not shown). The results presented in this paper were then transformed back to their original scales to avoid confusion from scaling.

RHS34: scaled between -1 and 1 based on its respective maximum and minimum. (Line 337)

RC34: I'm interested to know why data wasn't scaled with MEAN and STDEV rather? This approach is usually a bit more robust to outliers. Perhaps this is not a problem with model data? i.e. model averages will not have outliers?

AR34: Please see our response in AR35.

RHS35: This step ensures that no values are too close to the limits of the hyperbolic tangent sigmoid activation, (Line 337-338)

RC35: Scaling the data ensures that the gradient of each variable has the same "steepness". https://stats.stackexchange.com/questions/322822/how-normalizing-helps-to-increase-the-speed-of-the-learning

AR35: Yes, it makes sure they have the same "steepness," but it also ensures that the output of the activation function we are using (hyperbolic tangent sigmoid; tanh) are concentrated in a narrow range. For example, if the input to the tanh activation function is between -1 and 1 (inside the red bars of Fig. AR35.1), the range of the output is between about -0.76 and 0.76. In contrast, if the input is outside the range of -1 to 1 (outside the red bars of Fig. AR35.1), the output quickly approaches the extremes of -1 and 1 on the y-axis. If the outputs are toward the extreme

ends, this can cause the NNs to get "stuck" in those extremes during training which affects how much the weights can be adjusted during each epoch (ie. more epochs are needed for training which leads to longer training times).

[Figure]

Figure AR35.1: Tangent sigmoid activation function.

RHS36: These scalings were also applied to the RF and MLR methods for consistency between methods and the scaling did not affect the results of either method (results not shown). (Lines 339-340)

RC36: Rephrase: Scaling of the input variables is not necessary for RF and MLR, but was still applied for each of the methods for the sake of consistency with NNE.

AR36: The revised manuscript has rephrased the highlighted sentence.

RHS37: paper were then transformed back to their original scales to avoid confusion from scaling. (Line 341)

RC37: Scaling the output/target variable is not usually done. Though this would make no difference. I think you can thus leave this sentence out.

AR37: We received questions in past submissions when we did not specify that the values were transformed back to their original values. For clarity, we have kept this sentence in the revised manuscript.

OMT:

Lines 345-349

In Scenario 1, the RF and NNE both outperformed the MLR as demonstrated by higher $R_2$ values, lower MSE, and lower RMSE (Table 1). The decreased performance of the MLR is not inherently surprising, given the non-linearity of the underlying model, but it does demonstrate that the range of nutrients and light produced as inputs by ESM2Mc is capable of producing a non-linear response. Additionally, each method showed similar performances between the training and testing subsets suggesting adequate capture of the model dynamics in both subsets.

RHS38: Additionally, each method showed similar performances between the training and testing subsets suggesting adequate capture of the model dynamics in both subsets. (Lines 348-349)

RC38: I have a feeling that this might be due to the random shuffling of testing and training data; i.e. the training dataset is almost perfectly representative of the test dataset

AR38: That was our main intention that the training and testing subsets be representative of one another and of the complete dataset. The highlighted sentence provides support to that intention.

OMT:

Lines 367-373

When we computed an "effective" half-saturation for the nutrient curves in the top row of Fig. 2, we got values for $K_N$ that were far lower than the actual ones specified in the model (Table 4). The "effective" half-saturation of when other predictors are held at their 25th percentile for the micro- and macronutrient were underestimated by one and two orders of magnitude, respectively. It was only at the higher percentiles that the micronutrient "effective" half-saturation was adequately captured when the macronutrient was not limiting. Furthermore, the "effective" half-saturation of the macronutrient was not captured even when the other variables were held at their 75th percentiles because the 75th percentile of the micronutrient still limited growth.

RHS39: "effective" (Line 367)

RC39: why is this in quotations - maybe pseudo/quasi is a better word here?

AR39: These quotations were removed in the revised manuscript. Additionally, we removed other unnecessary quotation marks in the revised manuscript.

OMT:

Lines 424-436

Despite the fact that it agreed well with the observations, the RF prediction deviated from the true response to a given variable when other variables are held at higher percentiles (Fig. 2). This can likely be explained by the range of the training subset and how RFs acquire their predictions. When presented with predictor information, RFs rely on the information contained within their training data. If they are presented with predictor information that goes outside the range of the dataspace of the training set, RFs will provide a prediction based on the range of the training set. When performing the sensitivity analysis, the values of the predictors in the higher percentiles were probably outside the range of the training subset. For example, the bottom left plot of Fig. 2 shows how RF deviates from the true response as the concentration of the macronutrient increases – actually decreasing as nutrient increases despite the fact that such a result is not programmed into the underlying model. Although there may be observations in the training subset where the light and micronutrient are at their 75$_{th}$ percentile values when the macronutrient is low, there likely are not any observations where high levels of the macronutrient, micronutrient, and light are co-occurring. Without any observations meeting that criteria, the RF provided the highest prediction it could based on the training information. We discuss this point in more detail below.

RHS40: it (Line 424)

RC40: It would be better to put "the RF prediction" first and then "it"

AR40: Significant changes have been made to the results and discussion section of the revised manuscript.

OMT:

Lines 462-474

When comparing the apparent relationships of the time-averaged datasets with those of the hourly intrinsic relationships, the methods almost always underestimated the true response to light and nutrient (Fig. 3 and 4). This result is not entirely unexpected. The averaging of the hourly values into daily, weekly, and monthly timescales quickly leads to a loss of variability, especially for light (Fig. 5). In fact, the variability was lost in the daily time averaging with the longer timescales showing only small differences in the possible range of values (Fig. 5). The loss of variability means that the light limitation computed from the averaged light is systematically higher than the averaged light limitation. To match the observed biomass, the asymptotic biomass at high light has to be systematically lower (see Appendix A for the mathematical proof). Differences were much smaller for nutrients as they varied much less over the course of a month in our dataset. Our results emphasize that when comparing apparent relationships in the environment to intrinsic relationships from the laboratory, it is essential to take into account which timescales of variability averaging has removed. Insofar as most variability is at hourly time scales, daily-, weekly-, and monthly-averaged data will produce very

similar apparent relationships (Fig. 4). But if there was a strong week-to-week variability in some predictor, this may not be the case.

RHS41: has (Line 472)

RC41: was / has been

AR41: Significant changes have been made to the results and discussion section of the revised manuscript.

OMT:

Lines 495-501

The large increases in biomass in the micronutrient plots and hindrance of biomass in the light and macronutrient plots suggest that the system is limited by the concentration of micronutrient (Fig. 7). The biomass remained low even when macronutrient and light were at favorable levels because even when at the $75_{th}$ percentile value, the micronutrient was still limiting (Fig. 8). Conceptually this makes sense since the micronutrient limitation in the BLING model hinders growth, but also limits the efficiency of light-harvesting (Galbraith et al., 2010). Additionally, the computation of the "effective" half-saturation constants demonstrates that the half-saturation constant for light drops sharply as nutrients drop (Table 4).

RHS42: computation of the "effective" (Line 500)

RC42: Could be replaced by "estimated"

AR42: Yes, "estimated" seems to be the word more in line with our intention. The term "effective" has been replaced in the revised manuscript.

OMT:

Lines 504-507

Our main objective in this manuscript was to use ML to determine under what conditions intrinsic and apparent relationships between phytoplankton are no longer equal, to identify whether such divergence depends on the ML method or how the input data is handled, and to understand how such divergence is related to underlying biological dynamics.

RHS43: how the input data is handled, (Line 506)

RC43: Be more specific here, bring in the time aspect.

AR43: Significant changes have been made to the conclusions section of the revised manuscript.

OMT:

Lines 538-548

Both RFs and NNEs performed well when the predictions they were asked to make were within the range of the training data. However, the sensitivity analyses illustrated the impact of RFs inability to extrapolate outside that range and that RF's suggested systematic decreases in biomass at high values of a limiting variable. Nonetheless, RFs were able to capture the same relationships as the NNEs when the sensitivity analysis was querying environments within the range of the training data. It seems that as long as RFs are presented with information across the range of the dataset, RFs will perform just as well as NNEs in a sensitivity analysis. This strengthens the conclusions of Rivero-Calle et al. (2015) in that physiologically reasonable relationships between forcing variables and biomass found using RF are reliable so long as the forcing variables (in this case $pCO_2$ and temperature) vary over their entire range independently of other variables (nutrients and light). However, when variation in $pCO_2$ is related to variation in nutrients and light (i.e., in the seasonal climatology where $pCO_2$ is high in the winter, light is low, and nutrients are high) RFs are unable to extract a clear signal of $pCO_2$ limitation.

RHS44: This strengthens the conclusions of Rivero-Calle et al. (2015) in that physiologically reasonable relationships between forcing variables and biomass found using RF are reliable so long as the forcing variables (in this case pCO2 and temperature) vary over their entire range independently of other variables (nutrients and light). However, when variation in pCO2 is related to variation in nutrients and light (i.e., in the seasonal climatology where pCO2 is high in the winter, light is low, and nutrients are high) RFs are unable to extract a clear signal of pCO2 limitation. (Lines 543-548)

RC44: This is new literature in the conclusion. I would try to bring this into the discussion rather.

AR44: Significant changes have been made to the conclusions section of the revised manuscript.

OMT:

Lines 562-573

ML techniques have several benefits that could make them useful for biological oceanographers and ecosystem modelers. Many ML methods (including the two presented here) do not require any prior knowledge of a system to construct a model. Additionally, new methods are continually being developed for viewing the dynamics of the ML models. Given these advantages, ML could provide a compact form for representing relationships between ecosystem parameters such as biomass and primary productivity and their environmental drivers (nutrients and light) in observational data and complex models. Preliminary work indicates that we can use NNEs in particular to: 1. Compare model relationships with those derived from observational datasets, rather than simply using spatial patterns of errors. 2. Evaluate whether differences between models reflect important differences in biological parameters or whether they are due to differences in the physical circulation. We would expect that two different physical models run with the same biological scheme would produce the same relationships. 3. Evaluating whether global warming really would be expected to drive ecosystems outside their historical parameter range. We will report on these results in a future manuscript.

RHS45: dynamics (Line 564)

RC45: What do you mean by dynamics? Do you mean feature importances as an example?

AR45: Significant changes have been made to the conclusions section of the revised manuscript.

RHS46: compact form (Line 565)

RC46: unclear what is meant by this?

AR46: Significant changes have been made to the conclusions section of the revised manuscript.

RHS47: relationships (Line 568)

RC47: relationships of what?

AR47: Significant changes have been made to the conclusions section of the revised manuscript.

RHS48: Evaluate whether differences between models reflect important differences in biological parameters or whether they are due to differences in the physical circulation. (Lines 569-570)

RC48: Also, as pointed out by the author in the introduction, it may also differentiate whether models are similar for the right reason.

Would one not have to be careful of the implementation of the "intrinsic" equations and half-saturation constants of the model?

AR48: Significant changes have been made to the conclusions section of the revised manuscript.

OMT:

Line 758

Table 3: Scenario 3 comparison of MLR, RF, and NNE method performance for the training and testing sets.

RHS49: Scenario 3 comparison of MLR, RF, and NNE method performance for the training and testing sets. (Line 758)

RC49: Be specific about what the target variable is (biomass). Further, there are no units

AR49: The revised manuscript now includes the target biomass variable, along with its units, in the captions for the Tables.

OMT:

Line 763

(References Table 4 in the original manuscript)

RHS50: -2.11 x $10^4$ (This is the value in Table 4 associated with the 25th percentile of the Micronutrient for Scenario 3; Line 763)

RC50: Typo?

AR50: That is not a typo. One result we found was limitations in the variables can affect the estimate of the half-saturation. The issue is that there are not any observations where the macronutrient is at the 25th percentile or below when the micronutrient is limiting. The micronutrient is more or less uncorrelated with biology in this range and any half-saturations derived from it will be poor estimates. This clarification has been made in the revised manuscript.

RHS51: 1.85 (This is the value in Table 4 associated with the 25th percentile of the Light variable for Scenario 3; Line 763)

RC51: Should these light half-saturation estimates be the same as those above? If so, why is this so far off? If not, then make this clear!

AR51: In BLING, we use the Geider et al. model for light limitation. In this model, the chlorophyll to carbon ratio $\theta$ adjusts with the light; it becomes lower as light gets higher and higher as light gets lower. Since $Irr_k \propto \frac{Lim_{nut}}{\theta}$, this means that the ratio $Irr/Irr_k$ ends up being a lot more constant than one might expect – essentially it only drops to zero when the plankton cannot make any more chlorophyll. Please note that in Scenarios 1 and 2 we assumed that $Irr_k$ was independent of $Irr$. At very low values when nutrients are highly limiting, $Irr_k$ is very small; while at higher values of nutrients, it is larger.

OMT:

Lines 791-792

Figure 6: Scatter plots from the BLING model (a: surface biomass vs. temperature-normalized growth rate; b: mean nutrient limitation vs. monthly-averaged nutrients; c: mean light limitation vs. monthly-averaged Irr, Irr$_k$).

RHS52: Figure 6: Scatter plots from the BLING model (a: surface biomass vs. temperature-normalized growth rate; b: mean nutrient limitation vs. monthly-averaged nutrients; c: mean light limitation vs. monthly-averaged Irr, Irr$_k$). (Lines 791-792)

RC52: should the dependent (predicted) variables not be on the y-axes?

AR52: As it is currently constructed, the horizontal axis represents an "input" constructed from monthly mean variables (light, nutrient), while the vertical axis represents a target computed by the model. We have clarified this in the revised manuscript.

OMT:

Lines 795-798

Figure 7: Sensitivity analysis for Scenario 3 with the columns corresponding to the predictors and the rows corresponding with the percentile value at which the other predictors were set. The gray circles show the observations from the BLING model and the dashed lines show the predicted apparent relationships for each method.

RHS53: Figure 7: Sensitivity analysis for Scenario 3 with the columns corresponding to the predictors and the rows corresponding with the percentile value at which the other predictors were set. The gray circles show the observations from the BLING model and the dashed lines show the predicted apparent relationships for each method.

RC53: I have two comments here:

1) It would be more useful to have a 2D histogram in this data representation. Represent this data as a 2D contour where the colormap is scaled logarithmically. Using a grey colormap will then allow you to still plot the dashed lines in colour.

2) What are the actual curves for the intrinsic relationship?

AR53: Addressing comment 1: Yes, that is a good suggestion. We have implemented this in the revised manuscript.

Addressing comment 2: Since Scenario 3 was being used as a proof-of-concept, we did not include the intrinsic relationships. The reason for this was we were attempting to demonstrate the type of information one could gain from having access to Earth System Model output only, but not necessarily access to the computational resources to run the Earth System Model code itself.

RHS54: (The Referee comment is posted next to the Legend in Figure 7; Line 794)

RC54: Are the different models necessary here? If RF is susceptible to "missing data" why should it be used in an even more complex scenario? The same applies for MLR - it is clearly not complex enough to capture this signal.

AR54: The revised manuscript now only includes sensitivity curves for the NNE in this figure.

RHS55: (The Referee comment highlights the label "75th percentile" in Figure 7; Line 794)

RC55: Is it still useful to show the different percentiles at this point? Is this point not proven in experiment 2.

AR55: The different percentiles still affect the result. For example, in Figure 4 (in the original manuscript) the predicted biomass values increase with higher percentiles. The values on the y-axis change between the subplots.

RHS56: (The Referee comment highlights the "P" in the units for Biomass in Figure 7; Line 794)

RC56: What is P?

AR56: Here, P stands for phosphorus. This is analogous to the macronutrient term.

OMT:

Lines 801-802

Figure 8: A 3-D scatter plot showing the concentrations from Scenario 3 for the macronutrient, micronutrient, and light with the color of the data points corresponding to the biomass concentrations.

RHS57: Figure 8: A 3-D scatter plot showing the concentrations from Scenario 3 for the macronutrient, micronutrient, and light with the color of the data points corresponding to the biomass concentrations. (Lines 801-802)

RC57: I don't think this is a very informative plot. It is difficult to see what is really going on with the majority of the data being obscured by other data. In addition to this, it is quite strange to have the x-axis (macro) going from large to small, rather than small to large

AR57: The main purpose of this figure was to show that some of the relationships found by the sensitivity analysis were reflected in the data. Specifically, that increases in the micronutrient led to large increases in biomass.

It was necessary to plot it with the macronutrient going from large to small, otherwise the area we were wanting to focus on would be obscured by the other data.

We have removed this plot in the revised manuscript.

**Updated Author Responses to Comments from Anonymous Referee 3**

After the initial discussion had already closed, the Associate Editor was contacted by the third referee since they were unable to post the comments to the Interactive Discussion on the Biogeosciences website. We welcome this additional feedback and address these comments below.

In our previous Author Comments, we occasionally stated phrases such as "(Some change) will be included in the revised manuscript." Where applicable, we have updated our Author Responses below and now reference the actual revised manuscript.

In the following responses RC stands for Referee Comment and AR stands for Author Response. Referee Comments are in black-colored font and Author Responses are in red-colored font. Unless specified otherwise, references to figure/table numbers and line numbers refer to those of the original manuscript.

RC0: I was initially excited to dig into this manuscript after skimming the abstract and seeing what it was about, but after doing so I must admit I did not find it to be especially compelling. I am very sorry to be so harsh here, but it is possible that I misunderstand the authors' objectives, and I am certainly open to changing my mind about this manuscript.

AR0: We want to thank Anonymous Referee #3 for the comments and suggestions they have provided. We have found them to be helpful in revising the manuscript.

RC1: I see three results here:

AR1: We address the three statements in the following author responses AR1.1, AR1.2, and AR1.3.

RC1.1: The ML methods reproduce the output ok (high R^2s, Figure 1). Ok, yes, fine, but this is totally boring & expected since you're feeding them 60% of the data, which are presumably randomly chosen and therefore give fantastic coverage of the dataset. This is just a sanity check result, basically, but I count it for completeness.

AR1.1: The main purpose of splitting the data into training and testing datasets was to ensure a general fit to the data and to ensure each method had enough information to capture the underlying dynamics of the system. Given that this is a standard practice in machine learning, we included it here for completeness.

RC1.2: The effective half-saturation estimates are sometimes good, sometimes bad. This is interesting but it seems pretty all over the place why it's sometimes good & sometimes bad, so it's hard to draw too much from it besides that, as the authors discuss, one should take effective half-sats with caution and many grains of salt, likely because of colimitation. However, if the authors are attempting to claim that ML methods can't do a good job at this, the authors have to do a lot more work to show this; the results are based on pretty much all the default Matlab

settings, there aren't sensitivity results for these, there are other methods that might do better, etc. This is not a rigorous negative result, in other words.

AR1.2: Our original intent was to use the estimated half-saturation coefficients as a way to quantify the accuracy of the ML predictions. However, we realized this was not a reliable method and the revised manuscript now reflects this. In place of the quantitative method of estimating the half-saturation coefficients, we used more qualitative methods to visualize co-limitation patterns and variable interactions in the revised manuscript.

In terms of the default Matlab settings, we have included sensitivity analyses for the ML methods in Appendix B of the revised manuscript. In these analyses, we tested the number of decision trees in random forests, the number of nodes in a single layer NN, the addition of a second hidden layer in NNs, and different activation functions in NNs.

RC1.3: In a highly idealized setting, random forests don't get the intrinsic relationships right, while neural networks do. Beyond this, it pretty much falls apart. Again I find the lack of rigor in giving these methods and others a fair chance insufficient. Also, why not show the partial dependence plots in addition to what the authors have shown? Even if you like yours better, the PDPs can be illuminating.

AR1.3: Although we could show the partial dependence plots, we chose not to explicitly include them as figures in the revised manuscript so we could keep the length of the manuscript to a more reasonable length

RC1.4: Moreover, I guess my main issue with this whole paper is that the authors really seem to be conflating apples and oranges when it comes to the equations at hand. Intrinsic relationships are typically growth rate vs. (concentration, temperature, light level, whatever). Apparent relationships are concentrations versus these. The Flombaum et al PDP for abundance vs. temperature shouldn't look like the Eppley curve of growth rate vs. temperature. The authors make strict assumptions about loss rates (without explicitly deriving them) to try to match these up but this involves extra assumptions that are totally unrealistic including steady state and no feedbacks between trophic levels. And then when you integrate the intrinsic relationships over time and you include e.g. fluctuations in light, predator-prey cycling… of course the relationships are different. So the paper basically is just showing that except in extremely idealized contexts, and only then after making some strict assumptions to put things in the same units, intrinsic and apparent relationships between drivers and populations aren't the same, when you try to find the intrinsic relationships with a couple of ML methods using their default settings. I don't find this result interesting enough to warrant publication as it is.

AR1.4: On the long space and time scales associated with climatological model output, one *can* often see relationships between biomass and the drivers of growth rate. We have revised the discussion to make it more clear why this is the case. The reviewer is correct that these will result in causing the biomass to depend on more things than just the instantaneous growth rate. However, it is still useful to have some sense of how the intrinsic and apparent relationships might diverge in the specific case where we know what the functional relationship is- and in particular which methods would be expected to yield biased results.

We tackle the question the reviewer asks in a followup paper, where we compare BLING (which diagnoses biomass from the smoothed growth rate) with TOPAZ (which allows biomass to be transported and grazed).

Minor comments:

RC2: The title could use some work, it is long and confusing.

AR2: The title of the revised manuscript has been changed to be more reflective of the main points we are trying to make in the paper. The new title is "Can machine learning extract the mechanisms controlling phytoplankton growth from large-scale observations? – A proof of concept study."

RC3: Abstract: 'as little separation as hourly to daily' sounds like you mean as low as an 1h separation between timescales – I'd use 'versus' rather than 'to' to be clearer. You don't define MLR in the abstract, and it's also not clear what you mean by 'the limitations found by ML were overestimated' – by whom?

AR3: We have replaced the term "to" with "versus" in the referenced sentence.

We have removed the acronym of MLR and replaced it with its meaning of Multiple Linear Regression.

Where we stated "limitations found by ML were overestimated," we were referring to the estimated half-saturation coefficients. We believe the abstract in the revised manuscript clarifies this.

RC4: Reporting MSE & RMSE is totally redundant.

AR4: We have removed MSE from the relevant tables in the revised manuscript.

RC5: Equation 1: Ok, fine, you can write the equation this way, but at least cite an example of someone using the starting differential equations so we know better where you're coming from, if you're not going to derive it yourself? Also see above; this is not an intrinsic relationship in my mind, because you have to make additional assumptions about other things, namely zooplankton dynamics, that have nothing to do with any given intrinsic relationship.

AR5: The revised manuscript now includes the following sentence:

> "While simplistic, this is actually the steady-state solution of a simple phytoplankton-zooplankton system when grazing scales as the product of phytoplankton and zooplankton concentrations, and zooplankton mortality is quadratic in the zooplankton concentration."

A proof of this is offered below:

$$\frac{\partial P}{\partial t} = \mu * P - \lambda * \left(\frac{P}{Z_*}\right) * Z$$

$$\frac{\partial Z}{\partial t} = \epsilon\lambda * \left(\frac{P}{Z_*}\right) * Z - \gamma * \left(\frac{Z}{Z_*}\right) * Z$$

where $\mu$ is a growth rate, $\lambda$ a grazing rate for zooplankton on phytoplankton, $\gamma$ a rate of carnivory, $Z_*$ is a biomass scale, and $\epsilon$ an assimilation efficiency. Insofar as the rates of biomass transfer through such a system are slow in comparison to the rates at which biomass accumulates, setting the rate of changes in the first equation to be approximately zero enables us to solve for the approximate steady state

$$Z \approx \frac{\mu}{\lambda} * Z_*$$

and in the second

$$P \approx \frac{\gamma}{\epsilon\lambda} * Z = \frac{\mu}{\lambda} * \left(\frac{\gamma Z_*}{\epsilon\lambda}\right)$$

so that both zooplankton and phytoplankton concentrations are linear in the growth rate.

RC6: I don't understand why the authors switched the light portion of the equations from the different scenarios. Yes, you justify it & all as a Taylor expansion, but why not just use the same dependencies?

AR6: We responded to this comment by changing the dependencies in Scenarios 1 and 2 so that they mimic BLING.

---

## Referee Report (RR1)

**Review of Holder and Gnanadesikan (round 2)**

The authors have taken great care to implement the many suggestions made by the reviewers in the first round of revision. Thank you for implementing these. As a result, I find that the manuscript is much easier to follow and the presentation of the results (figures and tables) is clear – a fantastic improvement. I would also like to commend the authors for including results that may not appear to be a "success". I have added my comments below. They are only technical corrections. Suggested insertions are shown as italics.

**L241**: hours-days → *hours to days*
**L401**: "essentially captured all of it" could be strengthened with a quantitative addition of *($R^2 > 0.99$)*.
**L487**: capitalized "No" should be de-capitalized
**Fig3, 7, 8**: Would it be possible to show the same y-scale for the subplots? A suggestion. If the authors feel that this dilutes the message they are trying to convey (the shape of the curve), then do not change.
**Fig7**: A comment that might be useful for future experiments. The underestimation of biomass is the largest for irradiance. Could using the daytime-equivalent irradiance improve estimates? (i.e., do not include nighttime for averaging)

---

## Referee Report (RR2)

**Comments on revised version.**

Authors made significant improvements of their manuscript. I am appreciated that the authors took into account my previous comments. The manuscript is more completed now and shows meaningful results.

The title of the manuscript is more appropriate. Also, the introduction and conclusion are well structured, and it is much easier to understand the main goal of this work and its results. Authors conclude that the NNEs can provide an important information on the link between intrinsic and apparent relationships that provide more qualitative information than quantitative. It was not obvious in previous version of the manuscript.

The additional table 1 is very useful, as well as the Appendix B.

The results on figure 7 are very important and interesting. I want to congratulate the authors on this idea. Also new interaction plots help to enhance the main results and give the article completeness.

Thank you for the explanation of why you choose the $25^{th}$, $50^{th}$ and $75^{th}$ percentiles in your work to test the effect of limitations.

Questions and comments:

In the Eq.9 what do authors mean by min_s and max_s? Is it -1 and 1? Please clarify.

It can be better to present the maps of differences on Fig. 1. For example, keep Figure 1a like it is and show the differences on Fig.1 b, c and d between true values and ML methods.

On Fig. 2, 4 and 5 it would be useful to add the corresponding colours on the grey areas around curves.

Fig. 8, 9 and 12 miss a colorbar.

I did not notice any mention of the Appendix C in the text.

---

## Author Response (AR2)

Dear Editor,

We would like to thank the Reviewers for their constructive feedback in this second round of revisions. We have addressed each Reviewer's comments in our responses below.

In this second round of revisions, we have made the following changes:

- Included error plots in Fig. 1
- Changed the gray standard deviation regions around the curves in the sensitivity analyses to be the same color as the curves they represent
- Included mention of Appendix C in the main text
- Made minor corrections (RC1-3) suggested by Referee 2 (Luke Gregor)

We greatly appreciate your time, feedback, and consideration of our manuscript.

Thank you,

Christopher Holder

**Author Responses to Comments from Anonymous Referee 1 (Round 2)**

As with our previous Author Responses in Round 1, we continue the use abbreviations and colors for ease of reading. In the following responses RC stands for Referee Comment and AR stands for Author Response. Referee Comments are in black-colored font and Author Responses are in red-colored font. Unless specified otherwise, references to figure/table numbers and line numbers refer to those of the revised manuscript (Round 1). When we reference the updated revised manuscript that will be submitted along with these author responses, we refer to it as revised manuscript (Round 2).

RC0: Authors made significant improvements of their manuscript. I am appreciated that the authors took into account my previous comments. The manuscript is more completed now and shows meaningful results.

The title of the manuscript is more appropriate. Also, the introduction and conclusion are well structured, and it is much easier to understand the main goal of this work and its results. Authors conclude that the NNEs can provide an important information on the link between intrinsic and apparent relationships that provide more qualitative information than quantitative. It was not obvious in previous version of the manuscript.

The additional table 1 is very useful, as well as the Appendix B.

The results on figure 7 are very important and interesting. I want to congratulate the authors on this idea. Also new interaction plots help to enhance the main results and give the article completeness.

Thank you for the explanation of why you choose the $25^{th}$, $50^{th}$ and $75^{th}$ percentiles in your work to test the effect of limitations.

AR0: We want to thank Referee 1 for providing additional comments and feedback in this second round of review.

RC1: In the Eq.9 what do authors mean by min_s and max_s? Is it -1 and 1? Please clarify.

AR1: In the revised manuscript (Round 2), we have clarified that the value of min_s is -1 and the value of max_s is 1.

RC2: It can be better to present the maps of differences on Fig. 1. For example, keep Figure 1a like it is and show the differences on Fig.1 b, c and d between true values and ML methods.

AR2: We do agree with your comment that maps of differences can be informative and have incorporated them as part of Fig. 1. Rather than replace Fig. 1 b, c, and d with contour plots showing the errors, we included the error plots below Fig. 1 b, c, d for each ML method. It occurred to us that although it was clear from the error plots that the NNE and RF had the lowest errors, it still seemed like the original prediction contour plots also helped to demonstrate that these two methods did well at reproducing the original patterns. So rather than remove the original prediction contour plots, we added $Log_{10}$ Absolute Error plots beneath the prediction contour plots.

We chose to use the $Log_{10}$ Absolute Error metric so all the error plots could use the same color scale. The error values of the RF and NNE compared to the errors of the MLR differed by several orders of magnitude.

Additionally, we revised the text to include mention of the newly included plots.

RC3: On Fig. 2, 4 and 5 it would be useful to add the corresponding colours on the grey areas around curves.

RC3: We updated the grey areas around the curves in Fig. 2, 4, and 5 to be colored according to the color of the line they represent. For example, the grey standard deviation region around the green NNE line in Fig. 2 is now light green instead of grey. We also did this for the grey standard deviation regions around the curves in Fig. 11.

RC4: Fig. 8, 9 and 12 miss a colorbar.

AR4: For the interaction plots in Fig. 8 and 12, the color of the plots corresponds to the biomass concentration. Since the z-axis is already the biomass concentration, we chose not to include a colorbar because we thought it would be redundant. The main purpose of allowing the surface plots to be colored from blue to yellow was to introduce contrast into the figures so we could see and interpret them more effectively.

We did not include colorbars for the contour plots in Fig. 9 and 12 because each contour plot is normalized according to its own probability density function. The plots would have been much less informative if all the contour plots were on the same colorbar scale, since the scale for one contour plot had values of $10^4$ and another contour plot had values up to $10^{15}$. Additionally, the main purpose of the contour plots was to show the regions with the highest density of observations, so a colorbar did not seem completely necessary and we feared it might distract from the overall figure. However, in lieu of a colorbar we did make sure to state the meaning of the colors in the description of the appropriate figures (blue for few observations up to red for many observations).

RC5: I did not notice any mention of the Appendix C in the text.

AR5: We have included mentions in the main text of the figures in Appendix C in the revised manuscript (Round 2).

**Author Responses to Comments from Referee 2 – Luke Gregor (Round 2)**

As we did above with our Author Responses to Referee 1, we will continue the use abbreviations and colors for ease of reading. For details about the abbreviations and colors, along with how we reference the manuscript in our responses, please see the red text directly underneath the heading "Author Responses to Comments from Anonymous Referee 1 (Round 2)."

RC0: The authors have taken great care to implement the many suggestions made by the reviewers in the first round of revision. Thank you for implementing these. As a result, I find that the manuscript is much easier to follow and the presentation of the results (figures and tables) is clear – a fantastic improvement. I would also like to commend the authors for including results that may not appear to be a "success". I have added my comments below. They are only technical corrections. Suggested insertions are shown as italics.

AR0: We want to thank Luke Gregor (Referee 2) for their additional feedback and comments.

RC1: **L241**: hours-days → *hours to days*

AR1: We replaced the text as suggested.

RC2: **L401**: "essentially captured all of it" could be strengthened with a quantitative addition of

*($R^2 > 0.99$).*

AR2: We revised the text as suggested.

RC3: **L487**: capitalized "No" should be de-capitalized

AR3: We revised the text as suggested.

RC4: **Fig 3, 7, 8**: Would it be possible to show the same y-scale for the subplots? A suggestion. If the authors feel that this dilutes the message they are trying to convey (the shape of the curve), then do not change.

AR4: Ideally, we would have all the subplots on the same y-scale. However, as you already correctly pointed out, this makes it difficult to discern the shape of the curve in many of the subplots. Additionally, since the color scale corresponds to the biomass concentration (which is also on the z-axis), the colors of the surface plots become affected as well. These two aspects combined make it difficult to interpret the plots when they are on the same y-scale. For example, when we tried to have the subplot of Fig. 3 n on the same y-scale as the subplot of Fig. 3 m, the surface of Fig. 3 n appeared as a mostly flat blue plane since its biomass values were much lower than the biomass values in Fig. 3 m.

RC5: **Fig7**: A comment that might be useful for future experiments. The underestimation of biomass is the largest for irradiance. Could using the daytime-equivalent irradiance improve estimates? (i.e., do not include nighttime for averaging)

AR5: That is an excellent suggestion and one that we will consider for future experiments. It is reasonable to assume that using the daytime-equivalent would increase the average since all the zero values from nighttime would not be considered in the averaging.

**Author Responses to Comments from Anonymous Referee 3 (Round 2)**

Referee 3 selected the option "accepted as is" for the revised manuscript (Round 1), so no additional comments or feedback were provided by Referee 3. We want to thank Referee 3 for taking the time to review our manuscript and for their helpful feedback and comments from the first round of revisions.